# Do We Really Need to Approach the Entire Pareto Front in Many-Objective Bayesian Optimisation?

## Abstract

Many-objective optimisation, a subset of multi-objective optimisation, involves optimisation problems with more than three objectives. As the number of objectives increases, the number of solutions needed to adequately represent the entire Pareto front typically grows substantially. This makes it challenging, if not infeasible, to design a search algorithm capable of effectively exploring the entire Pareto front. This difficulty is particularly acute in the Bayesian optimisation paradigm, where sample efficiency is critical and only a limited number of solutions (often a few hundred) are evaluated. Moreover, after the optimisation process, the decision-maker eventually selects just one solution for deployment, regardless of how many high-quality, diverse solutions are available. In light of this, we argue an idea that under a limited evaluation budget, it may be more useful to focus on finding a single solution of the highest possible quality for the decision-maker, rather than aiming to approximate the entire Pareto front as existing many-/multi-objective Bayesian optimisation methods typically do. Bearing this idea in mind, this paper proposes a single point-based multi-objective search framework (SPMO) that aims to improve the quality of solutions along a direction that leads to a good tradeoff between objectives. Within SPMO, we present a simple acquisition function, called expected single-point improvement (ESPI), working under both noiseless and noisy scenarios. We show that ESPI can be optimised effectively with gradient-based methods via the sample average approximation (SAA) approach and theoretically prove its convergence guarantees under the SAA. We also empirically demonstrate that the proposed SPMO is computationally tractable and outperforms state-of-the-arts on a wide range of benchmark and real-world problems.

## 1 Introduction

Multi-objective optimisation problems (MOPs) (Emmerich & Deutz, 2018; Zheng & Wang, 2024) involve scenarios where multiple objectives need to be optimised simultaneously. Unlike single-objective optimisation problems which typically have a single optimal solution, in MOPs there is a set of optimal solutions known as Pareto optimal solutions. The corresponding points in the objective space form what is known as the Pareto front. In general, a multi-objective optimisation algorithm aims to generate a set of solutions that well approximate the Pareto front, from which the decision-maker chooses a solution to deploy based on their preferences.

In modern applications, optimisation is becoming increasingly complex, with a growing number of requirements and objectives that need to be considered at the same time (Lin et al., 2025). Taking the car cab design as an example, there are up to nine objectives to be optimised (Deb & Jain, 2013), including cabin space, fuel efficiency, acceleration time, and road noise at various speeds. This has given rise to a new research topic – many-objective optimisation, focusing on MOPs involving more than three objectives (Li et al., 2015).

At the same time, many real-world multi-/many-objective optimisation problems are black-box and costly in terms of solution evaluation. This is evident across a range of fields, including chemistry (Park et al., 2018; Shields et al., 2021; Dunlap et al., 2023), materials science (Liang et al., 2021; Low et al., 2024; Peng et al., 2025), and transportation (Deb & Jain, 2013; Jain & Deb, 2013a;

Cheaitou & Cariou, 2019; Deb et al., 2009). For example, in vehicle design optimisation, it can take about 20 hours to evaluate a vehicle design (Youn et al., 2004; Daulton et al., 2021). To tackle such problems, multi-objective Bayesian optimisation (MOBO) is a very effective approach (Garnett, 2023), along with other alternatives (e.g., surrogate-assisted evolutionary algorithms (Jin, 2011; Liang et al., 2024)). Over the past decades, a variety of effective MOBO methods have emerged, including scalarisation-based methods (Knowles, 2006; Paria et al., 2020; Lin et al., 2022) which convert a multi-objective problem into a number of single-objective problems, and Pareto-based methods which consider Pareto dominance relations over objectives (Daulton et al., 2020; Emmerich et al., 2006; Tu et al., 2022). Most works aim to find a good approximation of the entire Pareto front.

However, with the increase of the objective number, the number of solutions needed to adequately represent the problem's Pareto front typically grows substantially. For example, for a 10-objective problem, it normally needs 220 points (i.e., $\binom{12}{3}$) even if only three divisions on each objective are considered (Das & Dennis, 1998). This difficulty is especially pronounced in Bayesian optimisation, where often only a few hundred solutions can be generated and evaluated. With such a limited budget, it is highly unlikely for an optimisation algorithm to reach or even be close to the Pareto front. On top of that, after the optimisation process, the decision-maker eventually selects just one solution to deploy, regardless of how many high-quality, diverse solutions are provided.

Given the above, this paper argues an idea that under a limited evaluation budget, it may be more useful to focus on finding a single solution of the highest possible quality for the decision-maker, rather than aiming to approximate the entire Pareto front. That is, we do not care about diversifying solutions to represent the entire Pareto front, but focus on improving the quality of a single solution. Figure 1 illustrates this idea in a bi-objective case. As can be seen from the figure, in contrast to aiming for a set of diversified solutions which existing MOBO methods typically do (Li et al., 2025; Lin et al., 2022), our method aims for a single solution with better convergence (i.e., closer to the Pareto front). Although it may yield a worse *hypervolume* (HV) value (Zitzler & Thiele, 1999) compared to the diverse solution set obtained by existing methods, it may be more likely to be chosen by the decision-maker as it achieves a more favourable trade-off among the objectives.

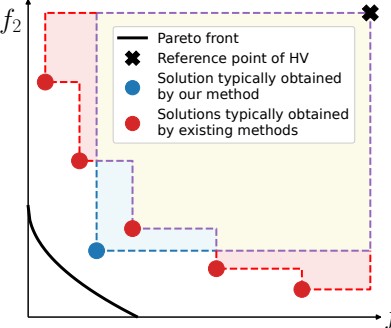

Figure 1: An illustration of our idea in a bi-objective case, in comparison with existing methods that aim to search for the entire Pareto front. The red points represent (non-dominated) solutions that existing methods may obtain, and the blue point represents what our method aims for. It can be seen that the red points are much more diversified, hence, as a whole, having a better hypervolume (HV) value (Zitzler & Thiele, 1999). However, the blue point has better convergence (i.e., closer to the Pareto front) than any single red point, which may be more likely to be preferred by the decision-maker.

Bearing this idea in mind, this paper proposes a single point-based multi-objective search framework, called SPMO. The contributions of this work can be summarised as follows.

- We propose a novel MOBO search framework that does not aim to approximate the entire Pareto front, but rather focuses on improving the quality of solutions along a single direction that leads to a good tradeoff between objectives.

- Within SPMO, we present a simple acquisition function, called Expected Single-Point Improvement (ESPI). We show that ESPI can be optimised effectively with gradient-based methods via the sample average approximation (SAA) approach from Balandat et al. (2020) and also theoretically prove its theoretical convergence guarantees under the SAA.

- We consider both noiseless and noisy cases, resulting in two versions of the proposed ESPI.

- We verify SPMO through an extensive experimental study, including in comparison with various state-of-the-arts, under both sequential and batch optimisation settings, through sensitivity analysis, with different metrics embedded, and on a range of benchmark and real-world problems.

## 2 BACKGROUND AND RELATED WORK

### 2.1 BACKGROUND

**Many-Objective Optimisation.** Many-objective optimisation, a subset of multi-objective optimisation, refers to an optimisation scenario having more than three objectives to be considered. Without loss of generality, this paper considers the problem of minimising a vector-valued function: $\boldsymbol{f}(\boldsymbol{x}) : \mathcal{X} \to \mathbb{R}^m$, where $\boldsymbol{x} \in \mathcal{X}$ ($\mathcal{X} \subset \mathbb{R}^d$) and $m$ is the number of objectives. In multi-/many-objective optimisation, a solution $\boldsymbol{x}_1$ is said to dominate $\boldsymbol{x}_2$, denoted by $\boldsymbol{x}_1 \prec \boldsymbol{x}_2$, if $\forall i \in \{1, ..., m\}$, $f_i(\boldsymbol{x}_1) \leq f_i(\boldsymbol{x}_2)$ and $\exists j \in \{1, ..., m\}$, $f_j(\boldsymbol{x}_1) < f_j(\boldsymbol{x}_2)$. If a solution $\boldsymbol{x}_1 \in \mathcal{X}$ is not dominated by any other solution, then $\boldsymbol{x}_1$ is said to be Pareto optimal. The collection of Pareto optimal solutions of a problem is called the Pareto set, and its mapping to the objective space is called Pareto front.

**Bayesian Optimisation (BO).** BO is a sample-efficient global optimisation approach that builds a probabilistic surrogate, typically a Gaussian process (GP), and uses an acquisition function $\alpha(\boldsymbol{x}) : \mathcal{X} \to \mathbb{R}$ to decide which points to evaluate. In this work, we model each objective with an independent Gaussian process $f_i \sim \mathcal{GP}(m_i(\boldsymbol{x}), k_i(\boldsymbol{x}, \boldsymbol{x}'))$, where $m_i(\boldsymbol{x}) : \mathcal{X} \to \mathbb{R}$ is the $i$th mean function, and $k_i(\cdot, \cdot) : \mathcal{X} \times \mathcal{X} \to \mathbb{R}$ is the $i$th covariance function. Given $n$ observed points $\mathcal{D}^n = \{(\boldsymbol{x}^t, \boldsymbol{y}^t)\}_{t=1}^n$ where $\boldsymbol{y}^t = \boldsymbol{f}(\boldsymbol{x}^t) + \boldsymbol{\zeta}^t$ and the noise $\boldsymbol{\zeta}^t \sim \mathcal{N}(0, \mathrm{diag}(\boldsymbol{\sigma}_\zeta^2))$, the posterior distribution of the $i$th objective at a new location $\boldsymbol{x}$ is a Gaussian distribution: $p(f_i(\boldsymbol{x})|\mathcal{D}^n) \sim \mathcal{N}(\mu_i(\boldsymbol{x}), \sigma_i^2(\boldsymbol{x}))$ where $\mu_i(\boldsymbol{x})$ and $\sigma_i^2(\boldsymbol{x})$ are the mean and variance at $\boldsymbol{x}$, respectively. Detailed expressions of the mean and variance are given in Appendix A.

### 2.2 RELATED WORK

Over the past decades, various MOBO methods have been proposed (Konakovic Lukovic et al., 2020; Daulton et al., 2022a). They can be loosely divided into scalarisation-based and Pareto-based methods. In scalarisation-based methods (Knowles, 2006; Paria et al., 2020), a multi-objective problem is converted into a number of single-objective problems (Chugh, 2020). Hence, one can leverage acquisition functions (Lai & Robbins, 1985) from single-objective BO to decide which point to evaluate. For instance, using random augmented Tchebycheff scalarisations (Miettinen, 1999), ParEGO (Knowles, 2006) and TS-TCH (Paria et al., 2020) optimise expected improvement (EI) (Jones et al., 1998) and Thompson sampling (TS) (Thompson, 1933), respectively. In contrast, Pareto-based methods consider Pareto dominance relations over objectives (Emmerich et al., 2006; Tu et al., 2022). A popular idea is to use HV as maximising the HV value is equivalent to finding the entire Pareto front (Shang et al., 2020). Expected hypervolume improvement (EHVI) is commonly used in MOBO (Couckuyt et al., 2014; Daulton et al., 2020; 2021). Another idea is to leverage information theory to guide exploration toward regions likely contributing to the Pareto front. For instance, joint entropy search (JES) (Tu et al., 2022) selects points that maximise the joint information gain for optimal inputs (i.e., the approximated Pareto set) and outputs (i.e., the approximated Pareto front). All of the above methods aim to approach the entire Pareto front.

It is worth noting that, similar to our approach, a few studies do not attempt to approximate the entire Pareto front. Some methods instead target a specific region of the front, such as the central area (Gaudrie et al., 2018; 2020; Binois et al., 2020). Another line of work incorporates decision-maker preferences by dynamically adjusting the target region based on elicited or updated preferences during optimisation (Abdolshah et al., 2019; Astudillo & Frazier, 2020; Ozaki et al., 2024; Ip et al., 2025). In contrast, our method assumes no prior knowledge of decision-maker preferences and seeks to identify a high-quality trade-off solution across objectives. A more detailed discussion of related work can be found in Appendix C.

## 3 THE PROPOSED METHOD

In this section, we first give the proposed MOBO framework. We then present the considered acquisition function (called ESPI), which is based on a simple distance-based metric. We note that analytically solving ESPI is not feasible, and thus consider its Monte Carlo approximation. Lastly, we consider ESPI under noisy cases, namely noisy ESPI (NESPI), and also its MC approximation.

**Single Point-based Multi-Objective (SPMO) Framework.** Algorithm 1 gives the procedure of the proposed SPMO framework. As can be seen, SPMO is very similar to a standard MOBO algorithm, except for the step of maximising the acquisition function based on a single-point quality metric (line

---

**Algorithm 1:** Single Point-based Multi-Objective (SPMO) Framework

**Input:** $\boldsymbol{f}$: Expensive black-box problem with $m$ objectives; $T$: Maximum number of evaluations;
  $g$: Metric that measures the quality of a single point; $\alpha$: Acquisition function;
  $\mathcal{D}^{n_0} := \{(\boldsymbol{x}^t, \boldsymbol{y}^t)\}_{t=1}^{n_0}$: Initial observed points.

1 **for** $n = n_0 + 1 : T$ **do**
2    $GPs \leftarrow \text{Train } \mathcal{GP}s(\mathcal{D}^n)$              `// Train m Gaussian process models`
3    $\boldsymbol{x}^n \leftarrow \arg\max_{\boldsymbol{x} \in \mathcal{X}} \alpha\big(g(\boldsymbol{x}, GPs)\big)$    `// Maximise the acquisition function based`
                                         `on the single-point quality metric g(·)`
4    $\boldsymbol{y}^n \leftarrow \boldsymbol{f}(\boldsymbol{x}^n) + \boldsymbol{\zeta}^n$                     `// Evaluate the solution xⁿ`
5    $\mathcal{D}^n \leftarrow \mathcal{D}^{n-1} \cup \{(\boldsymbol{x}^n, \boldsymbol{y}^n)\}$           `// Augment the observed solution`

**Output:** $\mathcal{D}^T$: Observed solutions.

---

3). In principle, any metric that can reflect the quality of a solution in achieving a good trade-off between objectives can be adopted. This includes distance-based metrics and scalarisation-based metrics, such as the weighted sum or augmented Tchebycheff (Miettinen, 1999) with a fixed weight vector (e.g., $(\frac{1}{m}, \ldots, \frac{1}{m}) \in \mathbb{R}^m$ in the $m$-objective case). Here, we consider a simple distance-based metric, and we will compare different metrics in our experiments (Section 6; details in Appendix F.5).

**Single-Point Improvement (SPI).** We consider the distance of solutions to a utopian point: $g(\boldsymbol{f}(\boldsymbol{x}), \boldsymbol{z}^*) = \|\boldsymbol{f}(\boldsymbol{x}) - \boldsymbol{z}^*\| = \sqrt{\sum_{i=1}^m \big(f_i(\boldsymbol{x}) - z_i^*\big)^2}$, where $m$ denotes the number of objectives, and $\boldsymbol{z}^* = (z_1^*, z_2^*, \ldots, z_m^*)$ is a utopian point, i.e., $z_i^* \leq \min_{\boldsymbol{x} \in \mathcal{X}} f_i(\boldsymbol{x})$. In many real-world cases, the utopian value of an objective can be loosely estimated, for example, by assuming idealised conditions such as zero cost, time or error (Branke et al., 2008). In our experimental evaluation, we perform a sensitivity analysis on the choice of the utopian point, and it shows that substantially different settings can yield consistent results.

We first consider noiseless cases, i.e., $\bar{\boldsymbol{y}}^t = \boldsymbol{f}(\bar{\boldsymbol{x}}^t)$. Let $\bar{\mathcal{D}}^n = \{(\bar{\boldsymbol{x}}^t, \bar{\boldsymbol{y}}^t\}_{t=1}^n$ be $n$ observed points and $\bar{X}^n = \{\bar{\boldsymbol{x}}^t\}_{t=1}^n$ be the set of all the observed decision vectors in $\bar{\mathcal{D}}^n$. For any point $\boldsymbol{x} \in \mathcal{X}$, we define the single-point improvement (SPI) as: $I_{SP}(\boldsymbol{f}(\boldsymbol{x})|g^*, \boldsymbol{z}^*, \bar{\mathcal{D}}^n) = \max\Big(0, \ g^* - \|\boldsymbol{f}(\boldsymbol{x}) - \boldsymbol{z}^*\|\Big)$, where $g^* = \min_{\boldsymbol{x} \in \bar{X}^n} g(\boldsymbol{f}(\boldsymbol{x}), \boldsymbol{z}^*)$.

**Expected Single-Point Improvement (ESPI).** We now present ESPI to account for the posterior distribution $p(\boldsymbol{f}|\bar{\mathcal{D}}^n)$. Suppose that we independently model each objective $f_i$ as a Gaussian process based on $\bar{\mathcal{D}}^n$. Then, the posterior of each $f_i$ at a new location $\boldsymbol{x}$ is a Gaussian random variable, i.e., $p(f_i(\boldsymbol{x})|\bar{\mathcal{D}}^n) \sim \mathcal{N}\big(\mu_i(\boldsymbol{x}), \sigma_i^2(\boldsymbol{x})\big)$, in which $f_1, \ldots, f_m$ are mutually independent Gaussians. Let $\eta_i := f_i - z_i^*$. Then we obtain $\eta_i \sim \mathcal{N}\big(\mu_i(\boldsymbol{x}) - z_i^*, \sigma_i^2(\boldsymbol{x})\big)$ which is a Gaussian as well. The proposed ESPI is defined as:

$$\alpha_{\text{ESPI}}(\boldsymbol{x}) = \mathbb{E}\big[I_{SP}(\boldsymbol{f}(\boldsymbol{x})|g^*, \boldsymbol{z}^*, \bar{\mathcal{D}}^n)\big] = \mathbb{E}_{p(\boldsymbol{\eta})}\big[\max(0, \ g^* - \|\boldsymbol{\eta}\|)\big] \tag{1}$$

An illustration of ESPI in a bi-objective case is given in Figure 5 of Appendix B for aiding understanding. Note that the integral in Eq. 1 cannot be solved analytically as it involves the distribution of $\|\boldsymbol{\eta}\|$, whose PDF and CDF have no closed-form expressions and are typically computed via numerical methods (Imhof, 1961; Ruben, 1962; Das, 2025). Hence, we use the MC integration with samples from the posterior $\tilde{\boldsymbol{f}}_t(\boldsymbol{x}) \sim p(\boldsymbol{f}(\boldsymbol{x})|\bar{\mathcal{D}}^n)$ for $t = 1, \ldots, N$ to estimate Eq. 1:

$$\alpha_{\text{ESPI}}(\boldsymbol{x}) \approx \hat{\alpha}_{\text{ESPI}}(\boldsymbol{x}) = \frac{1}{N} \sum_{t=1}^N I_{SP}(\tilde{\boldsymbol{f}}_t(\boldsymbol{x})|g^*, \boldsymbol{z}^*, \bar{\mathcal{D}}^n) \tag{2}$$

**Noisy Expected Single-Point Improvement (NESPI).** In the real world, it is not uncommon to encounter an optimisation problem with noises: $\boldsymbol{y}^t = \boldsymbol{f}(\boldsymbol{x}^t) + \boldsymbol{\zeta}^t$, where $\boldsymbol{\zeta}^t \sim \mathcal{N}(0, \text{diag}(\boldsymbol{\sigma}_\zeta^2))$. In noisy cases, simply using the observed best distance $g^*$ may adversely affect the optimisation performance. Here, we present an extension of ESPI, i.e., noisy ESPI (NESPI). Let $\mathcal{D}^n = \{(\boldsymbol{x}^t, \boldsymbol{y}^t)\}_{t=1}^n$ be $n$ observed points and $X^n = \{\boldsymbol{x}^t\}_{t=1}^n$ be the set of all the observed decision vectors in $\mathcal{D}^n$. By considering the uncertainty in the function values at $X^n$, the proposed NESPI is defined as:

$$\alpha_{\text{NESPI}}(\boldsymbol{x}) = \int \alpha_{\text{ESPI}}(\boldsymbol{x}|\hat{g}^*) p(\boldsymbol{f}|\mathcal{D}^n) d\boldsymbol{f} \tag{3}$$

where $\hat{g}^*$ denotes the smallest distance to the utopian point over $\boldsymbol{f}(X^n)$. Note that in noiseless cases, NESPI is equivalent to ESPI. Additionally, ESPI and NESPI can be naturally extended to the parallel (batch) setting by using sequential greedy approximation (Balandat et al., 2020).

Like in the noiseless case, the integral in Eq. 3 is also analytically intractable but can be approximated using the MC integration. Let $\tilde{\boldsymbol{f}}_t(\boldsymbol{x}) \sim p(\boldsymbol{f}(\boldsymbol{x})|\mathcal{D}^n)$ for $t = 1, \ldots, N$ be samples from the posterior, and let $\hat{g}^* = \min_{\boldsymbol{x} \in X^n} \|\tilde{\boldsymbol{f}}_t(\boldsymbol{x}) - \boldsymbol{z}^*\|$ be the smallest distance to utopian point over the previously evaluated points under the sampled function $\tilde{\boldsymbol{f}}_t(\boldsymbol{x})$. Then, $\alpha_{\text{NESPI}} \approx \frac{1}{N} \sum_{t=1}^{N} \alpha_{\text{ESPI}}(\boldsymbol{x}|\hat{g}^*, \boldsymbol{z}^*, \mathcal{D}^n)$. Using the MC integration, the inner expectation in $\alpha_{\text{NESPI}}$ can be computed simultaneously using samples from the joint posterior $\tilde{\boldsymbol{f}}_t(\boldsymbol{x}, X^n) \sim p(f(\boldsymbol{x}, X^n)|\mathcal{D}^n)$ over $\boldsymbol{x}$ and $X^n$:

$$\alpha_{\text{NESPI}}(\boldsymbol{x}) \approx \hat{\alpha}_{\text{NESPI}}(\boldsymbol{x}) = \frac{1}{N} \sum_{t=1}^{N} I_{SP}(\tilde{\boldsymbol{f}}_t|\hat{g}^*, \boldsymbol{z}^*, \mathcal{D}^n) \tag{4}$$

## 4    OPTIMISING ESPI AND NESPI

Having presented the MC estimators of ESPI and NESPI, we are now ready to optimise them.

**Differentiability.**    The MC estimators of ESPI and NESPI ($\hat{\alpha}_{\text{ESPI}}(\boldsymbol{x})$ in Eq. 2 and $\hat{\alpha}_{\text{NESPI}}(\boldsymbol{x})$ in Eq. 4) are differentiable with respect to $\boldsymbol{x}$. We are able to automatically compute exact gradients of the MC estimators of ESPI and NESPI ($\nabla_{\boldsymbol{x}} \hat{\alpha}_{\text{ESPI}}(\boldsymbol{x})$ and $\nabla_{\boldsymbol{x}} \hat{\alpha}_{\text{NESPI}}(\boldsymbol{x})$) by leveraging the auto-differentiation in modern computational frameworks. This facilitates efficient gradient-based optimisation of ESPI and NESPI.

**SAA Convergence Results.**    The sample average approximation (SAA) approach Kleywegt et al. (2002), which addresses stochastic optimisation problems by using the MC simulation, has gained increasing popularity and has become a standard technique in BO for optimising MC-based acquisition functions Balandat et al. (2020). By fixing the base samples, the SAA yields a deterministic acquisition function which enables using (quasi-) higher-order optimisation algorithms to obtain fast convergence rates for acquisition optimisation. We now give the theoretical convergence guarantees of ESPI under the SAA.

**Theorem 1.** *Suppose that $\mathcal{X}$ is compact and $\boldsymbol{f}$ has a multi-output GP prior whose mean and covariance functions are continuously differentiable. Let $\alpha_{\text{ESPI}}^* := \max_{\boldsymbol{x} \in \mathcal{X}} \alpha_{\text{ESPI}}(\boldsymbol{x})$ denote the maximum of ESPI, $S^* := \arg\max_{\boldsymbol{x} \in \mathcal{X}} \alpha_{\text{ESPI}}(\boldsymbol{x})$ denote the set of maximisers of $\alpha_{\text{ESPI}}$, $\hat{\alpha}_{\text{ESPI}}^N(\boldsymbol{x})$ denote the deterministic function via the base samples $\{\epsilon^t\}_{t=1}^N \sim \mathcal{N}(0, I_m)$. Suppose $\hat{\boldsymbol{x}}_N^* \in \arg\max_{\boldsymbol{x} \in \mathcal{X}} \hat{\alpha}_{\text{ESPI}}^N(\boldsymbol{x})$, then*

*(1) $\hat{\alpha}_{\text{ESPI}}^N(\hat{\boldsymbol{x}}_N^*) \to \alpha_{\text{ESPI}}^*$ a.s.*

*(2) $d(\hat{\boldsymbol{x}}_N^*, S^*) \to 0$ a.s., where $d(\hat{\boldsymbol{x}}_N^*, S^*) := \inf_{\boldsymbol{x} \in S^*} \|\hat{\boldsymbol{x}}_N^* - \boldsymbol{x}\|$.*

The proof of the theorem is given in Appendix D.1. This theorem indicates that one is able to optimise the MC estimator of the acquisition function ESPI to obtain a solution that converges almost surely to the optimal solution of the original function. For the noise case NESPI, due to space limitation we here skip the theorem of the theoretical convergence guarantees under the SAA. It (Theorem 2), along with the proof, can be found at Appendix D.2.

## 5    EXPERIMENTAL DESIGN

**Compared Methods.**    To evaluate the proposed SPMO, we consider six MOBO methods. They include one baseline method (Sobol (Sobol, 1967)), four well-established methods that aim to approximate the entire Pareto front, and one method that aims at the trade-off region of the Pareto front. The four methods consist of two scalarisation-based methods, ParEGO (Knowles, 2006) (along with its noisy variant NParEGO (Daulton et al., 2021)) and TS-TCH (Paria et al., 2020), and two Pareto-based methods, EHVI (Daulton et al., 2020) (along with its noisy variant NEHVI (Daulton et al., 2021)), and JES (Tu et al., 2022). For the method that does not aim at the entire Pareto front, we consider C-EHVI (Gaudrie et al., 2018; 2020). C-EHVI prefers the central region of the Pareto front, and we would like to see if it is competitive against our method in identifying a well-balanced

solution. For all EI-based methods, i.e., ParEGO, NParEGO, EHVI, NEHVI, C-EHVI and SPMO, we use the log version as suggested by Ament et al. (2023). Note that we only consider NESPI in this work, as it is equivalent to ESPI under noiseless cases.[1]

**Benchmarks and Real-World Problems.** For benchmark problems, we first choose two most widely scalable functions, DTLZ1 and DTLZ2 (Deb et al., 2005). However, their Pareto fronts are rather homogeneous, i.e., with a simplex shape. We then include their inverted versions, i.e., inverted DTLZ1 and inverted DTLZ2 (Deb & Jain, 2013). These problems do not include the one with convex Pareto fronts nor different objective scales. We thus add convex DTLZ2 and scaled DTLZ2 (Jain & Deb, 2013a). We also give the results of other DTLZ problems which have different features (e.g., degenerate and disconnected Pareto fronts) (Deb et al., 2005; Cheng et al., 2017) in Appendix F.1. Each problem is considered with 3, 5 and 10 objectives, following the practice in Deb & Jain (2013); Jain & Deb (2013a); Li et al. (2014), and tested under both noiseless and noisy cases. We also consider two well-studied expensive real-world problems (Tanabe & Ishibuchi, 2020), i.e., car side impact design (Jain & Deb, 2013a) and car cab design (Deb & Jain, 2013). The former is a four-objective problem without noise. The latter, which has nine objectives, involves four stochastic variables (out of the total seven variables) that introduce noise into the optimisation process, thus a natural optimisation problem with noise. For the other problems without noise, to make their noise cases, we use the additive zero-mean Gaussian noise with a standard deviation of 0.1, as suggested in Hernandez-Lobato et al. (2016); Jiang & Li (2025). The details of the problem formulations are given in Appendix E.3.

**Performance Metrics.** The proposed method aims to find a single trade-off solution between objectives which has a high chance of being favoured by the decision-maker. We thus first consider two single-point-based metrics, i.e., the distance-based metric used in the proposed method (reported as log-distance for better visualisation) and the HV-based metric, which measures the HV contribution of a solution. We expect that our method performs well on the distance-based metric since we directly optimise it. Notably, we are not certain whether our method performs best on the single-point HV since we do not directly optimise it. On the other hand, since most of the compared methods aim to achieve a good approximation of the entire Pareto front, we also consider the HV of the whole (nondominated) solution set obtained. It is expected that our method performs poorly, compared to other methods since we only optimise one point, rather than maximising the HV of the whole set (see Figure 1). For the reference point of the two HV metrics, we followed the practice in Ishibuchi et al. (2018); Balandat et al. (2020); Chugh (2020); Daulton et al. (2020) (see detailed settings in Appendix E.2).

**Budget and Statistical Validation.** For all the methods, we allow a maximum of 200 evaluations, following the practice in Daulton et al. (2020; 2021); Konakovic Lukovic et al. (2020). To enable statistical comparisons, each optimisation was repeated 30 times. We use the Wilcoxon rank-sum test (Wilcoxon, 1992) at a significance level of $\alpha = 0.05$ and Holm-Bonferroni correction (Holm, 1979) to see if our method differs significantly from each peer method.

# 6 EXPERIMENTAL RESULTS

We first report the results under noiseless cases, then under noisy cases and under batch settings. Next, we perform the sensitivity analysis of the utopian point used. Afterwards, we compare the proposed framework working with different single-point metrics, e.g., the distance-based, weighted sum-based and Tchebycheff-based. Lastly, we give the acquisition optimisation wall time for all the algorithms.

**Noiseless Cases.** We begin our evaluation by considering the distance metric, for which we expect good performance obtained by our SPMO. Table 1 shows the results of SPMO and the other methods on the seven benchmark (with five objectives) and real-world problems. Unsurprisingly, as can be seen from the table, our method significantly outperforms the other methods on all the problems. In addition, to understand the anytime performance, Figure 2 presents the trajectories of the distance-based metric obtained by the six methods. As seen, SPMO demonstrates a clear advantage over the other methods, obtaining a better convergence rate from the very beginning.

---

[1]Although NEHVI is equivalent to EHVI under noiseless cases, the wall time of NEHVI is much higher than EHVI (see Table 23). Hence we consider EHVI and NEHVI under noiseless and noisy cases, respectively.

Table 1: Results of the distance-based metric (log distance) obtained by the SPMO and the six peer methods on the benchmark problems with 5 objectives and the car side impact design problem on 30 runs. The method with the best mean is highlighted in bold. The symbols "+", "∼" and "−" indicate that the method is statistically worse than, equivalent to and better than our SPMO, respectively.

| Method | DTLZ1 Mean (Std) | DTLZ2 Mean (Std) | Inverted DTLZ1 Mean (Std) | Inverted DTLZ2 Mean (Std) | Convex DTLZ2 Mean (Std) | Scaled DTLZ2 Mean (Std) | Car side impact Mean (Std) | Sum up +/∼/− |
|---|---|---|---|---|---|---|---|---|
| Sobol | $3.7e+0\,(3.0e{-}1)^{+}$ | $2.4e{-}1\,(4.6e{-}2)^{+}$ | $4.8e+0\,(3.1e{-}1)^{+}$ | $6.0e{-}1\,(5.7e{-}2)^{+}$ | $-3.1e{-}1\,(2.5e{-}1)^{+}$ | $2.3e{-}1\,(5.1e{-}2)^{+}$ | $-1.9e{-}1\,(2.5e{-}2)^{+}$ | **7/ 0/ 0** |
| ParEGO | $3.4e+0\,(1.9e{-}1)^{+}$ | $6.7e{-}2\,(8.1e{-}2)^{+}$ | $3.2e+0\,(5.4e{-}1)^{+}$ | $2.5e{-}1\,(1.3e{-}2)^{+}$ | $-1.7e+0\,(2.0e{-}1)^{+}$ | $1.3e{-}1\,(8.9e{-}2)^{+}$ | $-3.3e{-}1\,(6.0e{-}3)^{+}$ | **7/ 0/ 0** |
| TS-TCH | $3.9e+0\,(2.2e{-}1)^{+}$ | $2.0e{-}1\,(4.1e{-}2)^{+}$ | $4.8e+0\,(3.2e{-}1)^{+}$ | $4.6e{-}1\,(1.2e{-}2)^{+}$ | $-7.1e{-}1\,(2.1e{-}1)^{+}$ | $2.6e{-}1\,(3.8e{-}2)^{+}$ | $-2.7e{-}1\,(1.5e{-}2)^{+}$ | **7/ 0/ 0** |
| EHVI | $3.5e+0\,(9.6e{-}2)^{+}$ | $9.3e{-}3\,(3.5e{-}3)^{+}$ | $4.0e+0\,(4.4e{-}1)^{+}$ | $2.3e{-}1\,(5.5e{-}3)^{+}$ | $-1.4e+0\,(2.1e{-}1)^{+}$ | $3.1e{-}1\,(6.0e{-}2)^{+}$ | $-3.3e{-}1\,(1.0e{-}2)^{+}$ | **7/ 0/ 0** |
| C-EHVI | $3.6e+0\,(1.4e{-}1)^{+}$ | $3.3e{-}3\,(3.8e{-}3)^{+}$ | $3.9e+0\,(4.6e{-}1)^{+}$ | $2.5e{-}1\,(1.8e{-}2)^{+}$ | $-3.5e{-}1\,(2.9e{-}1)^{+}$ | $7.6e{-}2\,(8.0e{-}2)^{+}$ | $-3.3e{-}1\,(1.0e{-}2)^{+}$ | **7/ 0/ 0** |
| JES | $3.4e+0\,(1.3e{-}1)^{+}$ | $1.1e{-}1\,(8.0e{-}2)^{+}$ | $4.5e+0\,(1.7e{-}1)^{+}$ | $2.6e{-}1\,(2.0e{-}2)^{+}$ | $-1.0e+0\,(4.5e{-}1)^{+}$ | $8.8e{-}2\,(9.3e{-}2)^{+}$ | $-3.3e{-}1\,(8.3e{-}3)^{+}$ | **7/ 0/ 0** |
| **SPMO** | **3.1e+0 (3.0e−1)** | **9.0e−4 (8.3e−4)** | **2.9e+0 (4.9e−1)** | **2.1e−1 (5.0e−5)** | **-2.1e+0 (7.7e−3)** | **1.7e−4 (1.2e−4)** | **-3.4e−1 (2.3e−6)** | |

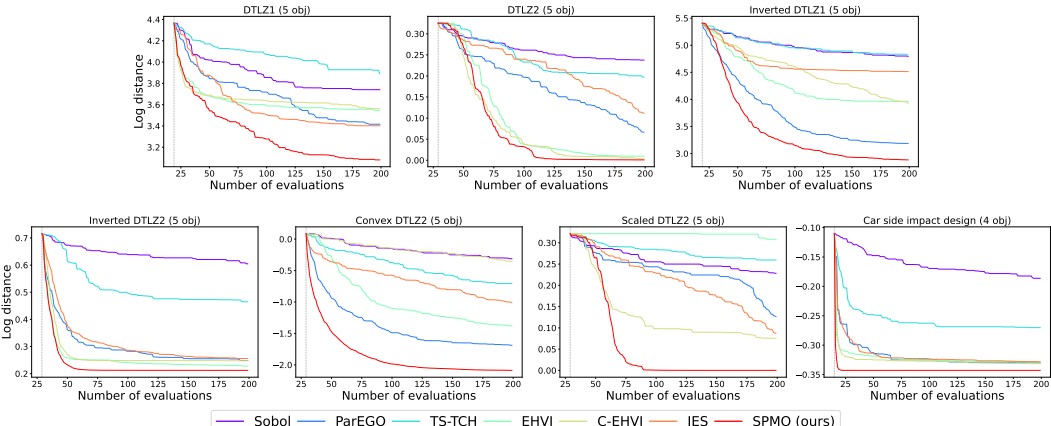

Figure 2: Trajectories of the distance-based metric (log distance) obtained by the seven methods on the benchmark problems with 5 objectives and the car side impact design problem. Each coloured line represents the mean metric value on 30 independent runs (after the initial Sobol samples, represented by the dashed grey line).

We now compare the methods by the HV metric of a single solution. Table 2 shows the results of the best solution (in terms of its HV value) obtained by SPMO and the five peer methods. As can be seen, SPMO is still very competitive, performing best on all the problems except DTLZ2 where EHVI obtains the best HV. To get a sense of what such a best-HV solution looks like, we use a spider chart to plot the solution on the five-objective inverted DTLZ1 problem in Figure 3. As seen, the solution of SPMO has the largest area, and it actually Pareto dominates the solutions of the other methods (i.e., better or at least equal on all five objectives).

Lastly, we consider the HV results of all the solutions obtained by the compared methods (Table 3). Interestingly, although SPMO does not aim to approximate the entire Pareto front, it still gets fairly good results—outperforming the other methods on at least 3 out of the 7 problems. One explanation for this is that within very tight budget, searching for improving convergence of solutions may play a bigger part than searching for improving diversity, thus contributing more to the HV value.

Due to the space limitation, we here only show results of the benchmark problems with five objectives. Results on 3- and 10-objective problems can be found in Appendix F.1. A general pattern is that as the number of objectives increases, the advantages of SPMO become more pronounced. On the 3-objective problems, the differences between SPMO and the peer methods in terms of the two single-point metrics are relatively small, whereas on the 10-objective problems, the gaps in both metrics become substantially larger (see Figures 6 and 7 in the Appendix). Regarding the HV of all evaluated solutions, SPMO statistically outperforms the peer methods on at least 2 and 5 out of the 6 problems on the 3-objective and 10-objective cases, respectively.

**Noisy Cases.** We compare the proposed SPMO with the peer methods on the seven noisy benchmark and real-world problems. Due to the space limitation, the results (the distance-based metric, two HV metrics, and convergence trajectories) are given in Appendix F.2. Like in the noiseless setting, SPMO significantly outperforms the peer methods on all the problems with respect to the single-point

Table 2: The HV of the best solution (in terms of its HV value) obtained by SPMO and the peer methods on the benchmark problems with 5 objectives and the car side impact design problem on 30 runs. The method with the best mean is highlighted in bold. The symbols "+", "∼" and "−" indicate that the method is statistically worse than, equivalent to and better than our SPMO, respectively.

| Method | DTLZ1 Mean (Std) | DTLZ2 Mean (Std) | Inverted DTLZ1 Mean (Std) | Inverted DTLZ2 Mean (Std) | Convex DTLZ2 Mean (Std) | Scaled DTLZ2 Mean (Std) | Car side impact Mean (Std) | Sum up +/∼/− |
|---|---|---|---|---|---|---|---|---|
| Sobol | 8.7e+12 (3.7e+11)$^+$ | 3.5e−2 (1.5e−2)$^+$ | 5.1e+12 (1.1e+12)$^+$ | 8.7e−4 (1.5e−3)$^+$ | 3.8e−1 (1.7e−1)$^+$ | 3.8e−2 (2.1e−2)$^+$ | 2.6e−1 (1.8e−2)$^+$ | **7/ 0/ 0** |
| ParEGO | 9.1e+12 (1.8e+11)$^+$ | 1.3e−1 (5.6e−2)$^+$ | 8.8e+12 (7.0e+11)$^+$ | 4.0e−2 (2.9e−3)$^+$ | 1.1e+0 (6.2e−2)$^+$ | 8.7e−2 (5.3e−2)$^+$ | 3.6e−1 (2.6e−3)$^+$ | **7/ 0/ 0** |
| TS-TCH | 8.4e+12 (3.8e+11)$^+$ | 3.2e−2 (1.3e−2)$^+$ | 4.9e+12 (1.2e+12)$^+$ | 7.1e−3 (1.1e−3)$^+$ | 6.5e−1 (1.4e−1)$^+$ | 2.8e−2 (1.5e−2)$^+$ | 3.1e−1 (9.3e−3)$^+$ | **7/ 0/ 0** |
| EHVI | 9.0e+12 (9.5e+10)$^+$ | **1.9e−1 (7.2e−3)$^\sim$** | 7.5e+12 (9.7e+11)$^+$ | 4.5e−2 (1.4e−3)$^+$ | 1.0e+0 (9.7e−2)$^+$ | 1.5e−2 (1.4e−2)$^+$ | 3.6e−1 (6.8e−3)$^+$ | **6/ 1/ 0** |
| C-EHVI | 9.0e+12 (1.0e+11)$^+$ | 1.6e−1 (1.9e−2)$^+$ | 7.7e+12 (7.8e+11)$^+$ | 4.1e−2 (4.1e−3)$^+$ | 4.3e−1 (2.1e−1)$^+$ | 1.2e−1 (5.1e−2)$^+$ | 3.6e−1 (4.4e−3)$^\sim$ | **6/ 1/ 0** |
| JES | 9.0e+12 (1.2e+11)$^+$ | 1.0e−1 (3.9e−2)$^+$ | 6.2e+12 (5.0e+11)$^+$ | 3.9e−2 (4.5e−3)$^+$ | 8.5e−1 (2.4e−1)$^+$ | 1.1e−1 (5.5e−2)$^+$ | 3.6e−1 (5.1e−3)$^+$ | **7/ 0/ 0** |
| SPMO | **9.3e+12 (2.8e+11)** | 1.9e−1 (1.4e−2) | **9.2e+12 (4.8e+11)** | **4.9e−2 (1.2e−5)** | **1.3e+0 (5.0e−3)** | **1.6e−1 (1.6e−2)** | **3.6e−1 (3.2e−3)** | |

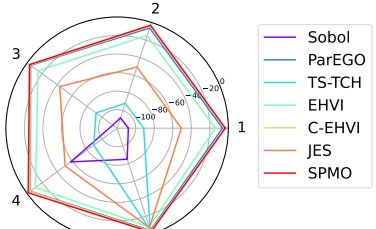

Figure 3: Spider chart of the best solution (in terms of its HV) obtained by the seven methods on the inverted DTLZ1 problem with 5 objectives in a typical run. Each axis in the spider chart represents one objective. Here, the objective values are multiplied by −1 in this minimisation problem, such that a solution with a larger area indicates better quality.

Table 3: The HV of all the solutions obtained by the seven methods on the seven benchmark (with five objectives) and real-world problems on 30 independent runs. The method with the best mean HV is highlighted in bold. The symbols "+", "∼", and "−" indicate that a method is statistically worse than, equivalent to, and better than SPMO, respectively.

| Method | DTLZ1 Mean (Std) | DTLZ2 Mean (Std) | Inverted DTLZ1 Mean (Std) | Inverted DTLZ2 Mean (Std) | Convex DTLZ2 Mean (Std) | Scaled DTLZ2 Mean (Std) | Car side impact Mean (Std) | Sum up +/∼/− |
|---|---|---|---|---|---|---|---|---|
| Sobol | 1.0e+13 (3.1e+10)$^+$ | 7.9e−2 (2.5e−2)$^+$ | 5.6e+12 (8.7e+11)$^+$ | 9.3e−4 (1.5e−3)$^+$ | 5.8e−1 (2.1e−1)$^+$ | 8.6e−2 (3.1e−2)$^+$ | 5.2e−1 (1.0e−2)$^+$ | **7/ 0/ 0** |
| ParEGO | 1.0e+13 (3.8e+10)$^-$ | 3.6e−1 (2.1e−1)$^+$ | 9.1e+12 (6.2e+11)$^+$ | 1.4e−1 (1.0e−2)$^-$ | **1.5e+0 (2.8e−2)$^-$** | 2.2e−1 (1.6e−1)$^+$ | 7.4e−1 (1.4e−2)$^-$ | **3/ 0/ 4** |
| TS-TCH | 1.0e+13 (2.6e+10)$^+$ | 5.9e−2 (2.7e−2)$^+$ | 5.4e+12 (1.0e+12)$^+$ | 2.0e−2 (3.6e−3)$^+$ | 1.0e+0 (1.5e−1)$^+$ | 5.1e−2 (2.7e−2)$^+$ | 6.5e−1 (1.1e−2)$^-$ | **6/ 0/ 1** |
| EHVI | **1.0e+13 (3.3e+8)$^-$** | **8.3e−1 (6.2e−2)$^-$** | 8.2e+12 (6.6e+11)$^+$ | **2.1e−1 (1.9e−3)$^-$** | 1.5e+0 (6.9e−2)$^+$ | 2.0e−2 (1.7e−2)$^+$ | 7.4e−1 (9.6e−3)$^-$ | **3/ 0/ 4** |
| C-EHVI | 1.0e+13 (1.6e+11)$^\sim$ | 3.6e−1 (8.5e−2)$^+$ | 8.0e+12 (6.9e+11)$^+$ | 7.2e−2 (9.2e−3)$^-$ | 5.8e−1 (2.7e−1)$^+$ | 2.1e−1 (1.1e−1)$^+$ | 5.9e−1 ( 3.0e−2)$^-$ | **4/ 1/ 2** |
| JES | 1.0e+13 (9.0e+9)$^-$ | 2.8e−1 (1.5e−1)$^+$ | 8.6e+12 (3.5e+11)$^+$ | 1.4e−1 (9.4e−3)$^-$ | 1.3e+0 (2.6e−1)$^+$ | **3.1e−1 (2.0e−1)$^\sim$** | 7.4e−1 (1.3e−2)$^-$ | **3/ 1/ 3** |
| SPMO | 1.0e+13 (8.1e+10) | 5.0e−1 (8.2e−2) | **9.6e+12 (2.9e+11)** | 6.4e−2 (2.8e−3) | 1.5e+0 (3.2e−2) | 3.0e−1 (1.1e−1) | 5.3e−1 (3.1e−2) | |

metrics (distance and HV). Regarding the HV of all the (nondominated) solutions obtained, SPMO achieves the best performance on the majority of the problems. Figure 4 shows the spider chart of the best solution (with respect to its HV) obtained by each algorithm in a typical run on the 9-objective car cab design problem. The violin plot of their HV values in 30 independent runs is given in the left panel for reference. As can be seen, the solution of SPMO is the best or close to the best on most of the objectives (except the first objective), thus having a clearly larger area.

**Batch Setting.** Previously, we considered the case in the sequential setting, where solutions are evaluated sequentially. Now we want to see if the proposed method works in the batch setting. Here, the batch size $q$ is set to 5, a commonly used value (Lin et al., 2022). As can be seen in Tables 14–16 and Figures 13–15 (Appendix F.3), similar to the results in the sequential setting, SPMO generally performs best. On the single-point metrics, it achieves the smallest distance on all the problems and the highest HV on 5 out of the 6 problems (except on DTLZ2). As for the HV of all evaluated solutions, SPMO obtains the best HV on two problems and takes the second or third places on the remaining ones. This indicates that prioritising convergence is also very useful for many-objective problems in batch settings.

**Sensitivity Analysis.** A parameter needed in the proposed method is the utopian point. In our experiment, we set it to be the vector consisting of the best value on each objective (i.e., the problem's ideal point). However, in real life, the ideal point is usually unknown before the optimisation. Hence, we would like to investigate how much different utopian points affect the performance. In this context, we consider three different settings. The first one is slightly better than the ideal point, i.e., with a difference of $0.01$, the second is fairly better than the ideal point (i.e. $0.1$), and the last one is significantly better than the ideal point (i.e. $1.0$). The results are given in Appendix F.4. As can be seen, interestingly, SPMO with the three lower utopian points with different levels performs better

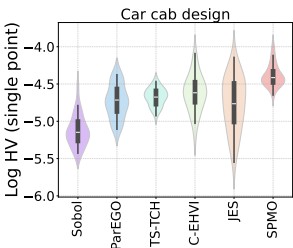 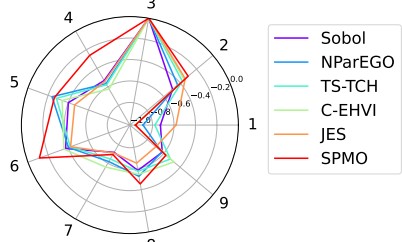

Figure 4: *Left:* Violin plots of the HV values of the best solution (with respect to its HV) obtained by all the methods on the car cab design problem in 30 independent runs. *Right:* The objective values (normalised and multiplied by $-1$) of the best solution (with the highest HV) obtained by each method on the cab design problem in a typical run.

than or at least equivalently to SPMO with our current setting. This indicates that 1) using the ideal point in the proposed method may not be the best choice (though it performs better than the other MOBO methods), and 2) SPMO's performance is robust to the choice of the utopian point and a liberal estimate is sufficient to achieve good results, e.g., considering zero cost in real-world cases.

**Comparison of Single-Point Metrics within SPMO.** In the proposed SPMO framework, we employ a distance metric (i.e., the distance of a solution to the utopian point), denoted as $\text{SPMO}_{dist}$. However, different metrics can be adopted provided that they can reflect the quality of a solution in achieving a good trade-off between objectives. We now consider two other well-known metrics, weighted sum and Tchebycheff scalarisation (with the same weights $(\frac{1}{m}, \ldots, \frac{1}{m})$), denoted by $\text{SPMO}_{ws}$ and $\text{SPMO}_{Tch}$, respectively. We compare these three versions of SPMO. The results (given in Appendix F.5) show that $\text{SPMO}_{dist}$ performs in general better than $\text{SPMO}_{ws}$ and $\text{SPMO}_{Tch}$. It obtains the best result on at least 4 out of the 6 problems on the HV of the best solution. As for the HV of all the solutions, $\text{SPMO}_{dist}$ performs best on DTLZ1 and its variants, but worse than $\text{SPMO}_{Tch}$ on DTLZ2 and its variants (except convex DTLZ2). A possible explanation is that $\text{SPMO}_{Tch}$ has slower convergence and can be better in exploring different solutions, thus better on relatively easy-to-converge problems.

**Acquisition Optimisation Wall Time.** Lastly, we present the wall time for optimising the acquisition function (i.e., determining a solution to be evaluated). The results (Appendix F.6) show that our method is among the fastest algorithms. When the number of objectives is 3 or 5, the time of all the methods is acceptable with a maximum of 98 seconds. As the number of objectives increases to 10, hypervolume-based methods (i.e., EHVI and NEHVI) become very expensive (taking about half an hour and more than 3 hours, respectively).[2] The proposed SPMO method shows high computational efficiency, achieving the lowest time requirement in four out of the six instances.

## 7 CONCLUSION

This work presented a multi-objective BO framework that aims to find a single trade-off solution of the highest possible quality with respect to multiple objectives, rather than seeking to explore their entire Pareto front. We theoretically proved the convergence guarantees under the SAA and empirically verified the proposed framework through extensive experiments, including on noiseless/noisy and sequential/batch cases, by sensitivity analysis, with different metrics for the acquisition function, and on a range of benchmark and real-world problems. A noticeable limitation of the proposed framework is that it focuses on finding a single trade-off point, thus failing to capture the information about the entire Pareto front; it thus may be less useful for certain applications where such information is valuable (e.g., the Pareto front's ranges and nadir points). A detailed discussion of its applicability is provided in Appendix G. However, interestingly, the proposed framework showed its competitiveness against existing state-of-the-arts with respect to even the quality of the whole solution set (through HV, see Table 3). Future work includes studying and enhancing the scalability of the proposed methods (i.e., in higher-dimensional search space) and extending their applicability to other scenarios, e.g., multi-fidelity optimisation (see Appendix H for more details).

---

[2]Wall time is measured based on the initial samples. Due to the exponentially increasing computational complexity with objectives, EHVI and NEHVI are fully evaluated only for problems with objectives $m \leq 5$.

ETHICS STATEMENT

This work complies with the ICLR Code of Ethics. No human subjects, personal data, or sensitive information are involved, and no risks of harm are anticipated.

REPRODUCIBILITY STATEMENT

Implementation details and experimental settings are provided in Appendix E. The data and code are available at an anonymised repository for reproducibility: `https://anonymous.4open.science/r/SPMO-D550`.

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

# Appendix to:

# Do We Really Need to Approach the Entire Pareto Front in Many-Objective Bayesian Optimisation?

## A  MULTI-OBJECTIVE BAYESIAN OPTIMISATION (MOBO)

MOBO consists of two main steps, i.e., training $m$ Gaussian process models based on the observed solutions and optimising an acquisition function $\alpha(\boldsymbol{x}) : \mathcal{X} \to \mathbb{R}$ to select a solution for evaluation. In this work, we model each objective with an independent Gaussian process $f_i \sim \mathcal{GP}(m_i(\boldsymbol{x}), k_i(\boldsymbol{x}, \boldsymbol{x}'))$, where $m_i(\boldsymbol{x}) : \mathcal{X} \to \mathbb{R}$ is the $i$th mean function, and $k_i(\cdot, \cdot) : \mathcal{X} \times \mathcal{X} \to \mathbb{R}$ is the $i$th covariance function. We use the notation $K(\mathrm{A}, \mathrm{B})$ to represent the covariance matrix at all pairs of solutions in set A and in set B. Given $n$ observed solutions $\mathcal{D}^n = \{(\boldsymbol{x}^t, \boldsymbol{y}^t)\}_{t=1}^n$ where $\boldsymbol{y}^t = \boldsymbol{f}(\boldsymbol{x}^t) + \boldsymbol{\zeta}^t$ and the noise $\boldsymbol{\zeta}^t \sim \mathcal{N}(0, \mathrm{diag}(\boldsymbol{\sigma}_\zeta^2))$, the posterior distribution of $i$th objective at a new location $\boldsymbol{x}$ is a Gaussian distribution:

$$p(f_i(\boldsymbol{x})|\mathcal{D}^n) \sim \mathcal{N}(\mu_i(\boldsymbol{x}), \sigma_i^2(\boldsymbol{x})) \tag{5}$$

$$\mu_i(\boldsymbol{x}) = K(\boldsymbol{x}, X^n)(K(X^n, X^n) + \sigma_{\zeta_i}^2 \mathbf{I})^{-1} Y_i \tag{6}$$

$$\sigma_i^2(\boldsymbol{x}) = K(\boldsymbol{x}, \boldsymbol{x}) - K(\boldsymbol{x}, X^n)((K(X^n, X^n) + \sigma_{\zeta_i}^2 \mathbf{I})^{-1} K(X^n, \boldsymbol{x}) \tag{7}$$

where $\mu_i(\boldsymbol{x})$ and $\sigma_i^2(\boldsymbol{x})$ are the mean and variance at $\boldsymbol{x}$, respectively; $X^n = (\boldsymbol{x}^1, \dots, \boldsymbol{x}^n) \in \mathbb{R}^{n \times d}$ and $Y_i^n = (y_i^1, \dots, y_i^n) \in \mathbb{R}^n$ are the matrix of evaluated solutions and the corresponding vector of $y$ values, respectively; $\sigma_{\zeta_i}^2$ is the variance of the observation noise $\zeta_i \sim \mathcal{N}(0, \sigma_{\zeta_i}^2)$, and corresponds to the $i$-th diagonal entry of the noise covariance matrix $\mathrm{diag}(\boldsymbol{\sigma}_\zeta^2)$;

## B  ILLUSTRATIVE EXAMPLE OF ESPI

To help understand the proposed ESPI, an illustration of ESPI in a bi-objective case is shown in Figure 5. In this example, the utopian point is denoted by the black star, while the red dots represent the nondominated solutions from the current dataset. A new candidate point, whose objective values are yet to be observed, is shown as the blue dot. The red dotted line indicates the shortest Euclidean distance $g^*$ from the utopian point to the existing nondominated set. In contrast, the blue dotted line represents the distance from the new point to the utopian point. When a new point lies within the shaded blue region and is closer to the utopian point, the improvement $I_{SP}(\cdot)$ is higher.

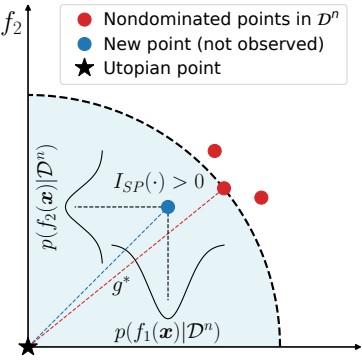

Figure 5: Illustration of the proposed expected single-point improvement (ESPI) in a bi-objective space. The figure shows the utopian point (black star), nondominated points in the current dataset $\mathcal{D}^n$ (red dots), and a new candidate point whose true objective values are not yet observed (blue dot). The dashed arc represents the current best Euclidean distance from the utopian point to the nondominated set, denoted as $g^*$ (red dotted line). The blue dotted line represents the distance from the new candidate point to the utopian point $\boldsymbol{z}^*$, which may improve upon the current best distance $g^*$. When a new point lies within the shaded blue region and is closer to the utopian point, the improvement $I_{SP}(\cdot)$ is higher.

## C    EXTENDED RELATED WORK

Over the last two decades, a variety of BO methods have been proposed to tackle expensive multi-objective optimisation problems (Jeong & Obayashi, 2005; Bautista, 2009; Svenson, 2011; Svenson & Santner, 2016; Zuluaga et al., 2016; Zhan et al., 2017; Picheny et al., 2019; Belakaria et al., 2020b; Malkomes et al., 2021). Most of them aim to identify a good approximation of the entire Pareto front (Keane, 2006; Svenson, 2011; Parr, 2013; Zuluaga et al., 2013; Ahmadianshalchi et al., 2024). To do so, some studies convert a multi-objective problem into multiple single-objective problems by a scalarisation function (e.g., random augmented Tchebycheff scalarisation). They then optimise acquisition functions from single-objective BO to determine the next evaluation point(s) (Knowles, 2006; Zhao & Zhang, 2023). In these studies, different acquisition functions are employed, such as expected improvement (EI) (Jones et al., 1998) in Knowles (2006); Zhang et al. (2010); Namura et al. (2017); Chugh (2020), Thompson sampling (TS) (Thompson, 1933) in Paria et al. (2020); Zhang & Golovin (2020), and upper confidence bound (UCB) (Lai & Robbins, 1985) in Paria et al. (2020); Zhang & Golovin (2020).

The remaining studies directly optimise multi-objective problems by considering the definition of optimality in multi-objective optimisation, i.e., the Pareto dominance relation. A representative approach is to use HV since maximising the HV value is equivalent to finding the entire Pareto front (Ponweiser et al., 2008; Couckuyt et al., 2014; Daulton et al., 2020; 2021). Along this line, expected hypervolume improvement (EHVI) is widely considered (Emmerich et al., 2006; Daulton et al., 2023; Qing et al., 2023; Yang et al., 2019b;a; Deng et al., 2025) as it is a natural extension of the EI for multi-objective optimisation. Another idea is to leverage information theory to guide exploration toward regions likely contributing to the Pareto front. Such methods focus on improving the posterior of optimal inputs (i.e., the approximated Pareto set) (Garrido-Merchán et al., 2023; Garrido-Merchán & Hernández-Lobato, 2019; Hernandez-Lobato et al., 2016), optimal outputs (i.e., the approximated Pareto front) (Belakaria et al., 2019; 2021; Suzuki et al., 2020), or both of them (Tu et al., 2022).

That said, there do exist a few studies that do not aim to identify the entire Pareto front. Among them, some attempt to use decision-maker preferences to guide the search towards specific region(s) (Abdolshah et al., 2019; Astudillo & Frazier, 2020; Ozaki et al., 2024; Ip et al., 2025). Such methods typically adjust the target region by eliciting or updating decision-maker preferences during optimisation. Another attempt is to directly target certain region(s) of the Pareto front, without an assumption that decision-maker preferences can be available or elicited (Gaudrie et al., 2018; 2020; Binois et al., 2020). For example, Gaudrie et al. (2018; 2020) propose the Centred Expected Hypervolume Improvement (C-EHVI) by dynamically adjusting the reference point using the Kalai-Smorodinsky equilibrium (also used in Binois et al. (2020)) to approach the central part of the Pareto front. Such work is more relevant to our study, and we have thus included C-EHVI in our experimental comparison.

## D    THEORETICAL RESULTS

### D.1    PROOF OF THEOREM 1.

We consider the setting from Balandat et al. (2020, Section D.5). Let $\epsilon^t \sim \mathcal{N}(0, I_m)$.[3] Using the reparameterisation trick, we can write the posterior at $\boldsymbol{x}$ as

$$\bar{\boldsymbol{f}}_{\boldsymbol{t}}(\boldsymbol{x}, \epsilon^t) = \bar{\mu}(\boldsymbol{x}) + \bar{L}(\boldsymbol{x})\epsilon^t$$

where $\bar{\mu}(\boldsymbol{x}) : \mathbb{R}^d \to \mathbb{R}^m$ is the multi-output GP's posterior mean; $\bar{L}(\boldsymbol{x}) \in \mathbb{R}^{m \times m}$ is a root decomposition (often a Cholesky decomposition) of the multi-output GP's posterior covariance $\bar{K} \in \mathbb{R}^{m \times m}$; and $\epsilon^t \in \mathbb{R}^m$. Let

$$\bar{A}(\boldsymbol{x}, \epsilon^t) = \max\left(0, g^* - \|\bar{\boldsymbol{f}}_{\boldsymbol{t}}(\boldsymbol{x}, \epsilon^t) - \boldsymbol{z}^*\|\right)$$

where $g^* = \min_{\boldsymbol{x} \in \bar{X}^n} g(\boldsymbol{f}(\boldsymbol{x}), \boldsymbol{z}^*)$ and $\boldsymbol{z}^* = (z_1^*, z_2^*, \ldots, z_m^*)$ is the utopian point. Following (Balandat et al., 2020, Theorem 3), we need to show that there exists an integrable function $\ell : \mathbb{R}^m \mapsto$

---

[3]Theorem 1 can be extended to handle non-iid base samples from a family of quasi-Monte Carlo methods as in Balandat et al. (2020).

$\mathbb{R}$ such that for almost every $\epsilon^t$ and all $\boldsymbol{x}, \boldsymbol{y} \in \mathcal{X} \subset \mathbb{R}^d$,

$$|\bar{A}(\boldsymbol{x}, \epsilon^t) - \bar{A}(\boldsymbol{y}, \epsilon^t)| \le \ell(\epsilon^t)\|\boldsymbol{x} - \boldsymbol{y}\|. \tag{8}$$

Let

$$\bar{A}(\boldsymbol{x}, \epsilon^t) = \max\left(0,\ g^* - \|\bar{\boldsymbol{f}_t}(\boldsymbol{x}, \epsilon^t) - \boldsymbol{z}^*\|\right)$$

$$= \frac{1}{2}\left(g^* - \|\bar{\boldsymbol{f}_t}(\boldsymbol{x}, \epsilon^t) - \boldsymbol{z}^*\| + |g^* - \|\bar{\boldsymbol{f}_t}(\boldsymbol{x}, \epsilon^t) - \boldsymbol{z}^*\||\right).$$

Hence we obtain,

$$\left|\bar{A}(\boldsymbol{x}, \epsilon^t) - \bar{A}(\boldsymbol{y}, \epsilon^t)\right|$$

$$= \left|\frac{1}{2}\left(\|\bar{\boldsymbol{f}_t}(\boldsymbol{y}, \epsilon^t) - \boldsymbol{z}^*\| - \|\bar{\boldsymbol{f}_t}(\boldsymbol{x}, \epsilon^t) - \boldsymbol{z}^*\|\right) + \frac{1}{2}\left(|g^* - \|\bar{\boldsymbol{f}_t}(\boldsymbol{x}, \epsilon^t) - \boldsymbol{z}^*\|| - |g^* - \|\bar{\boldsymbol{f}_t}(\boldsymbol{y}, \epsilon^t) - \boldsymbol{z}^*\||\right)\right|.$$

Let $I_1 := \|\bar{\boldsymbol{f}_t}(\boldsymbol{y}, \epsilon^t) - \boldsymbol{z}^*\| - \|\bar{\boldsymbol{f}_t}(\boldsymbol{x}, \epsilon^t) - \boldsymbol{z}^*\|$ and $I_2 := |g^* - \|\bar{\boldsymbol{f}_t}(\boldsymbol{x}, \epsilon^t) - \boldsymbol{z}^*\|| - |g^* - \|\bar{\boldsymbol{f}_t}(\boldsymbol{y}, \epsilon^t) - \boldsymbol{z}^*\||$. Hence

$$\left|\bar{A}(\boldsymbol{x}, \epsilon^t) - \bar{A}(\boldsymbol{y}, \epsilon^t)\right| \le \frac{1}{2}|I_1| + \frac{1}{2}|I_2|.$$

We observe that

$$|I_1| = \left|\|\bar{\boldsymbol{f}_t}(\boldsymbol{y}, \epsilon^t) - \boldsymbol{z}^*\| - \|\bar{\boldsymbol{f}_t}(\boldsymbol{x}, \epsilon^t) - \boldsymbol{z}^*\|\right|$$

$$\le \|\bar{\boldsymbol{f}_t}(\boldsymbol{y}, \epsilon^t) - \bar{\boldsymbol{f}_t}(\boldsymbol{x}, \epsilon^t)\|$$

$$= \|\bar{\mu}(\boldsymbol{y}) + \bar{L}(\boldsymbol{y})\epsilon^t - (\bar{\mu}(\boldsymbol{x}) + \bar{L}(\boldsymbol{x})\epsilon^t)\|$$

$$\le \|\bar{\mu}(\boldsymbol{y}) - \bar{\mu}(\boldsymbol{x})\| + \|(\bar{L}(\boldsymbol{y}) - \bar{L}(\boldsymbol{x}))\epsilon^t\|.$$

Since $\mathcal{X}$ is compact, and $\bar{\mu}$ and $\bar{L}$ have uniformly bounded gradients, they are Lipschitz. There exist $C_{\mu_1}, C_{L_1} < \infty$ such that

$$|I_1| \le \|\bar{\mu}(\boldsymbol{y}) - \bar{\mu}(\boldsymbol{x})\| + \|(\bar{L}(\boldsymbol{y}) - \bar{L}(\boldsymbol{x}))\epsilon^t\|$$

$$\le \ell_{I_1}(\epsilon^t)\|\boldsymbol{x} - \boldsymbol{y}\|$$

where $\ell_{I_1}(\epsilon^t) := C_{\mu_1} + C_{L_1}\|\epsilon^t\|$. Furthermore,

$$|I_2| = \left||g^* - \|\bar{\boldsymbol{f}_t}(\boldsymbol{x}, \epsilon^t) - \boldsymbol{z}^*\|| - |g^* - \|\bar{\boldsymbol{f}_t}(\boldsymbol{y}, \epsilon^t) - \boldsymbol{z}^*\||\right|$$

$$\le \left|\|\bar{\boldsymbol{f}_t}(\boldsymbol{x}, \epsilon^t) - \boldsymbol{z}^*\| - \|\bar{\boldsymbol{f}_t}(\boldsymbol{y}, \epsilon^t) - \boldsymbol{z}^*\|\right|$$

$$\le \|\bar{\boldsymbol{f}_t}(\boldsymbol{x}, \epsilon^t) - \bar{\boldsymbol{f}_t}(\boldsymbol{y}, \epsilon^t)\|$$

$$= \|\bar{\mu}(\boldsymbol{x}) + \bar{L}(\boldsymbol{x})\epsilon^t - (\bar{\mu}(\boldsymbol{y}) + \bar{L}(\boldsymbol{y})\epsilon^t)\|$$

$$\le \|\bar{\mu}(\boldsymbol{x}) - \bar{\mu}(\boldsymbol{y})\| + \|(\bar{L}(\boldsymbol{x}) - \bar{L}(\boldsymbol{y}))\epsilon^t\|.$$

Since $\mathcal{X}$ is compact, and $\bar{\mu}$ and $\bar{L}$ have uniformly bounded gradients, they are Lipschitz. There exist $C_{\mu_2}, C_{L_2} < \infty$ such that

$$|I_2| \le \ell_{I_2}(\epsilon^t)\|\boldsymbol{x} - \boldsymbol{y}\|$$

where $\ell_{I_2}(\epsilon^t) := C_{\mu_2} + C_{L_2}\|\epsilon^t\|$. Hence

$$\left|\bar{A}(\boldsymbol{x}, \epsilon^t) - \bar{A}(\boldsymbol{y}, \epsilon^t)\right| \le \frac{1}{2}|I_1| + \frac{1}{2}|I_2|$$

$$\le \frac{1}{2}\ell_{I_1}(\epsilon^t)\|\boldsymbol{x} - \boldsymbol{y}\| + \frac{1}{2}\ell_{I_2}(\epsilon^t)\|\boldsymbol{x} - \boldsymbol{y}\|$$

$$= \frac{1}{2}(\ell_{I_1}(\epsilon^t) + \ell_{I_2}(\epsilon^t))\|\boldsymbol{x} - \boldsymbol{y}\|.$$

Hence

$$\left|\bar{A}(\boldsymbol{x}, \epsilon^t) - \bar{A}(\boldsymbol{y}, \epsilon^t)\right| \le \ell(\epsilon^t)\|\boldsymbol{x} - \boldsymbol{y}\|$$

where $\ell(\epsilon^t) := (C_{\mu_1} + C_{\mu_2}) + (C_{L_1} + C_{L_2})\|\epsilon^t\|$. Note that $\ell(\epsilon^t)$ is integrable because all absolute moments exist for the Gaussian distribution. Since this satisfies the criteria for Theorem 3 in Balandat et al. (2020), the theorem holds for ESPI.

## D.2 THEOREM 2 AND ITS PROOF

**Theorem 2.** *Suppose that $\mathcal{X}$ is compact and that $\boldsymbol{f}$ has a multi-output GP prior with continuously differentiable mean and covariance functions. Let $X^n = \{\boldsymbol{x}^t\}_{t=1}^n$ be the set of observed decision vectors in $\mathcal{D}^n$, $\alpha_{\text{NESPI}}^* := \max_{\boldsymbol{x}\in\mathcal{X}} \alpha_{\text{NESPI}}(\boldsymbol{x})$ denote the maximum of NESPI, $S^* := \arg\max_{\boldsymbol{x}\in\mathcal{X}} \alpha_{\text{NESPI}}(\boldsymbol{x})$ denote the set of maximisers of $\alpha_{\text{NESPI}}$, $\hat{\alpha}_{\text{NESPI}}^N(\boldsymbol{x})$ denote the deterministic acquisition function via the base samples $\{\epsilon^t\}_{t=1}^N \sim \mathcal{N}(0, I_{(n+1)m})$. Suppose that $\hat{\boldsymbol{x}}_N^* \in \arg\max_{\boldsymbol{x}\in\mathcal{X}} \hat{\alpha}_{\text{NESPI}}^N(\boldsymbol{x})$, then*

*(1) $\hat{\alpha}_{\text{NESPI}}^N(\hat{\boldsymbol{x}}_N^*) \to \alpha_{\text{NESPI}}^*$ a.s.,*

*(2) $d(\hat{\boldsymbol{x}}_N^*, S^*) \to 0$ a.s., where $d(\hat{\boldsymbol{x}}_N^*, S^*) := \inf_{\boldsymbol{x}\in S^*} \|\hat{\boldsymbol{x}}_N^* - \boldsymbol{x}\|$.*

*Proof of Theorem 2.* Let $X^n := [(\boldsymbol{x}^1)^T, \ldots, (\boldsymbol{x}^n)^T]^T \in \mathbb{R}^{nd}$ be the $n$ observed points; $\boldsymbol{x}^{n+1} \in \mathcal{X} \subset \mathbb{R}^d$ be a candidate point; and $\epsilon^t \in \mathbb{R}^{(n+1)m}$ with $\epsilon^t \in \mathcal{N}(0, I_{(n+1)m})$. Let $\tilde{\boldsymbol{f}}_t(X^n, \boldsymbol{x}^{n+1}) := [\tilde{\boldsymbol{f}}_t(\boldsymbol{x}^1), \ldots, \tilde{\boldsymbol{f}}_t(\boldsymbol{x}^n), \tilde{\boldsymbol{f}}_t(\boldsymbol{x}^{n+1})]$ denote the $t^{\text{th}}$ sample of the corresponding objectives, so that we can write the posterior via the reparameterisation trick as:

$$\boldsymbol{f}_t(X^n, \boldsymbol{x}^{n+1}, \epsilon^t) = \mu(X^n, \boldsymbol{x}^{n+1}) + L(X^n, \boldsymbol{x}^{n+1})\epsilon^t$$

where $\mu(X^n, \boldsymbol{x}^{n+1}) : \mathbb{R}^{(n+1)d} \to \mathbb{R}^{(n+1)m}$ is the multi-output GP's posterior mean; $L(X^n, \boldsymbol{x}^{n+1}) \in \mathbb{R}^{(n+1)m \times (n+1)m}$ is a root decomposition of the multi-output GP's posterior covariance $K \in \mathbb{R}^{(n+1)m \times (n+1)m}$. Let $\boldsymbol{f}^{(m)}(\boldsymbol{x}_i, \epsilon^t) := S_i\Big(\mu(X^n, \boldsymbol{x}^{n+1}) + L(X^n, \boldsymbol{x}^{n+1})\epsilon^t\Big)$, where $\boldsymbol{f}^{(m)}(\boldsymbol{x}_i, \epsilon^t) \in \mathbb{R}^m$ represents the posterior at point $\boldsymbol{x}_i$ and $S_i \in \mathbb{R}^{m \times (n+1)m}, i = 1, \ldots, n+1$ is the selector matrix used to extract the corresponding element for the $\boldsymbol{x}_i$. Let

$$A(\boldsymbol{x}^{n+1}, \epsilon^t; X^n) = \max\Big(0, \hat{g}_t^*(X^n) - \|\boldsymbol{f}^{(m)}(\boldsymbol{x}^{n+1}, \epsilon^t) - \boldsymbol{z}^*\|\Big).$$

Let $\hat{g}_t^*(X^n) := \min_{\boldsymbol{x}_{obs}\in\{\boldsymbol{x}_i\}_{i=1}^n} \|\boldsymbol{f}_t^{(m)}(\boldsymbol{x}_{obs}, \epsilon^t) - \boldsymbol{z}^*\|$, hence we obtain:

$$A(\boldsymbol{x}^{n+1}, \epsilon^t; X^n) = \max\Big(0, \min_{\boldsymbol{x}_{obs}\in\{\boldsymbol{x}_i\}_{i=1}^n} \|\boldsymbol{f}^{(m)}(\boldsymbol{x}_{obs}, \epsilon^t) - \boldsymbol{z}^*\| - \|\boldsymbol{f}^{(m)}(\boldsymbol{x}^{n+1}, \epsilon^t) - \boldsymbol{z}^*\|\Big).$$

Following Balandat et al. (2020, Theorem 3), we need to show that there exists an integrable function $\ell : \mathbb{R}^m \mapsto \mathbb{R}$ such that for almost every $\epsilon^t$ and all $\boldsymbol{x}^{n+1}, \boldsymbol{y}^{n+1} \in \mathcal{X} \subset \mathbb{R}^d$,

$$|A(\boldsymbol{x}^{n+1}, \epsilon^t; X^n) - A(\boldsymbol{y}^{n+1}, \epsilon^t; X^n)| \le \ell(\epsilon^t)\|\boldsymbol{x}^{n+1} - \boldsymbol{y}^{n+1}\|. \tag{9}$$

Let

$$A(\boldsymbol{x}^{n+1}, \epsilon^t; X^n) = \max\Big(0, \hat{g}_t^*(X^n) - \|\boldsymbol{f}^{(m)}(\boldsymbol{x}^{n+1}, \epsilon^t) - \boldsymbol{z}^*\|\Big)$$

$$= \frac{1}{2}\Big(\hat{g}_t^*(X^n) - \|\boldsymbol{f}^{(m)}(\boldsymbol{x}^{n+1}, \epsilon^t) - \boldsymbol{z}^*\| + \big|\hat{g}_t^*(X^n) - \|\boldsymbol{f}^{(m)}(\boldsymbol{x}^{n+1}, \epsilon^t) - \boldsymbol{z}^*\|\big|\Big).$$

Hence we obtain

$$\big|A(\boldsymbol{x}^{n+1}, \epsilon^t; X^n) - A(\boldsymbol{y}^{n+1}, \epsilon^t; X^n)\big|$$

$$= \Big|\frac{1}{2}\Big(\|\boldsymbol{f}^{(m)}(\boldsymbol{y}^{n+1}, \epsilon^t) - \boldsymbol{z}^*\| - \|\boldsymbol{f}^{(m)}(\boldsymbol{x}^{n+1}, \epsilon^t) - \boldsymbol{z}^*\|\Big)$$

$$+ \frac{1}{2}\Big(\big|\hat{g}_t^*(X^n) - \|\boldsymbol{f}^{(m)}(\boldsymbol{x}^{n+1}, \epsilon^t) - \boldsymbol{z}^*\|\big| - \big|\hat{g}_t^*(X^n) - \|\boldsymbol{f}^{(m)}(\boldsymbol{y}^{n+1}, \epsilon^t) - \boldsymbol{z}^*\|\big|\Big)\Big|.$$

Let $I_1' := \|\boldsymbol{f}^{(m)}(\boldsymbol{y}^{n+1}, \epsilon^t) - \boldsymbol{z}^*\| - \|\boldsymbol{f}^{(m)}(\boldsymbol{x}^{n+1}, \epsilon^t) - \boldsymbol{z}^*\|$ and $I_2' := \big|\hat{g}_t^*(X^n) - \|\boldsymbol{f}^{(m)}(\boldsymbol{x}^{n+1}, \epsilon^t) - \boldsymbol{z}^*\|\big| - \big|\hat{g}_t^*(X^n) - \|\boldsymbol{f}^{(m)}(\boldsymbol{y}^{n+1}, \epsilon^t) - \boldsymbol{z}^*\|\big|$. Hence

$$\big|A(\boldsymbol{x}^{n+1}, \epsilon^t; X^n) - A(\boldsymbol{y}^{n+1}, \epsilon^t; X^n)\big| \le \frac{1}{2}|I_1'| + \frac{1}{2}|I_2'|.$$

We observe that

$$
\begin{aligned}
|I_1'| &= \left| \|\boldsymbol{f}^{(m)}(\boldsymbol{y}^{n+1}, \epsilon^t) - \boldsymbol{z}^*\| - \|\boldsymbol{f}^{(m)}(\boldsymbol{x}^{n+1}, \epsilon^t) - \boldsymbol{z}^*\| \right| \\
&\leq \|\boldsymbol{f}^{(m)}(\boldsymbol{y}^{n+1}, \epsilon^t) - \boldsymbol{f}^{(m)}(\boldsymbol{x}^{n+1}, \epsilon^t)\| \\
&= \|\mu^{(m)}(\boldsymbol{y}^{n+1}) + L^{(m)}(\boldsymbol{y}^{n+1})\epsilon^t - (\mu^{(m)}(\boldsymbol{x}^{n+1}) + L^{(m)}(\boldsymbol{x}^{n+1})\epsilon^t)\| \\
&\leq \|\mu^{(m)}(\boldsymbol{y}^{n+1}) - \mu^{(m)}(\boldsymbol{x}^{n+1})\| + \|(L^{(m)}(\boldsymbol{y}^{n+1}) - L^{(m)}(\boldsymbol{x}^{n+1}))\epsilon^t\|
\end{aligned}
$$

Since $\mathcal{X}$ is compact, and $\mu^{(m)}$ and $L^{(m)}$ have uniformly bounded gradients, they are Lipschitz. There exist $C_{\mu_1}', C_{L_1}' < \infty$ such that

$$
\begin{aligned}
|I_1'| &\leq \|\mu^{(m)}(\boldsymbol{y}^{n+1}) - \mu^{(m)}(\boldsymbol{x}^{n+1})\| + \|(L^{(m)}(\boldsymbol{y}^{n+1}) - L^{(m)}(\boldsymbol{x}^{n+1}))\epsilon^t\| \\
&\leq \ell_{I_1'}(\epsilon^t)\|\boldsymbol{x}^{n+1} - \boldsymbol{y}^{n+1}\|.
\end{aligned}
$$

where $\ell_{I_1'}(\epsilon^t) := C_{\mu_1}' + C_{L_1}'\|\epsilon^t\|$. Furthermore,

$$
\begin{aligned}
|I_2'| &= \left| \left| \hat{g}_t^*(X^n) - \|\boldsymbol{f}^{(m)}(\boldsymbol{x}^{n+1}, \epsilon^t) - \boldsymbol{z}^*\| \right| - \left| \hat{g}_t^*(X^n) - \|\boldsymbol{f}^{(m)}(\boldsymbol{y}^{n+1}, \epsilon^t) - \boldsymbol{z}^*\| \right| \right| \\
&\leq \left| \|\boldsymbol{f}^{(m)}(\boldsymbol{x}^{n+1}, \epsilon^t) - \boldsymbol{z}^*\| - \|\boldsymbol{f}^{(m)}(\boldsymbol{y}^{n+1}, \epsilon^t) - \boldsymbol{z}^*\| \right| \\
&\leq \|\boldsymbol{f}^{(m)}(\boldsymbol{x}^{n+1}, \epsilon^t) - \boldsymbol{f}^{(m)}(\boldsymbol{y}^{n+1}, \epsilon^t)\| \\
&= \|\mu^{(m)}(\boldsymbol{x}^{n+1}) + L^{(m)}(\boldsymbol{x}^{n+1})\epsilon^t - (\mu^{(m)}(\boldsymbol{y}^{n+1}) + L^{(m)}(\boldsymbol{y}^{n+1})\epsilon^t)\| \\
&\leq \|\mu^{(m)}(\boldsymbol{x}^{n+1}) - \mu^{(m)}(\boldsymbol{y}^{n+1})\| + \|(L^{(m)}(\boldsymbol{x}^{n+1}) - L^{(m)}(\boldsymbol{y}^{n+1}))\epsilon^t\|
\end{aligned}
$$

Since $\mathcal{X}$ is compact, and $\mu^{(m)}$ and $L^{(m)}$ have uniformly bounded gradients, they are Lipschitz. There exist $C_{\mu_2}', C_{L_2}' < \infty$ such that

$$
|I_2'| \leq \ell_{I_2'}(\epsilon^t)\|\boldsymbol{x}^{n+1} - \boldsymbol{y}^{n+1}\|
$$

where $\ell_{I_2'}(\epsilon^t) := C_{\mu_2}' + C_{L_2}'\|\epsilon^t\|$. Hence

$$
\begin{aligned}
\left| A(\boldsymbol{x}^{n+1}, \epsilon^t) - A(\boldsymbol{y}^{n+1}, \epsilon^t) \right| &\leq \frac{1}{2}|I_1'| + \frac{1}{2}|I_2'| \\
&\leq \frac{1}{2}\ell_{I_1'}(\epsilon^t)\|\boldsymbol{x}^{n+1} - \boldsymbol{y}^{n+1}\| + \frac{1}{2}\ell_{I_2'}(\epsilon^t)\|\boldsymbol{x}^{n+1} - \boldsymbol{y}^{n+1}\| \\
&= \frac{1}{2}(\ell_{I_1'}(\epsilon^t) + \ell_{I_2'}(\epsilon^t))\|\boldsymbol{x}^{n+1} - \boldsymbol{y}^{n+1}\|
\end{aligned}
$$

Hence

$$
\left| A(\boldsymbol{x}^{n+1}, \epsilon^t) - A(\boldsymbol{y}^{n+1}, \epsilon^t) \right| \leq \ell(\epsilon^t)\|\boldsymbol{x}^{n+1} - \boldsymbol{y}^{n+1}\|
$$

where $\ell(\epsilon^t) := (C_{\mu_1}' + C_{\mu_2}') + (C_{L_1}' + C_{L_2}')\|\epsilon^t\|$. Note that $\ell(\epsilon^t)$ is integrable because all absolute moments exist for the Gaussian distribution. Since this satisfies the criteria for Theorem 3 in Balandat et al. (2020), the theorem holds for NESPI. It is worth noting that Theorems 1 and 2 readily extend to the batch setting.

It can be shown that the gradient of $\hat{\alpha}_{\text{ESPI}}(\boldsymbol{x})$ or $\hat{\alpha}_{\text{NESPI}}(\boldsymbol{x})$ is an unbiased estimator of the true gradient of $\alpha_{\text{ESPI}}$ or $\alpha_{\text{NESPI}}$, though it is not necessary for the SAA approach Daulton et al. (2021).

# E    EXPERIMENT SETTINGS

## E.1    IMPLEMENTATION DETAILS

All experiments were conducted using Python 3.12, with all methods developed on the open-source Python framework BoTorch (Balandat et al., 2020), which builds on GPyTorch (Gardner et al., 2018) for Gaussian process modelling and PyTorch (Paszke et al., 2019) for automatic differentiation. The computational studies were performed on a Red Hat Enterprise Linux 8.8 system, operating on a 64-bit x86 CPU architecture. The computing cluster utilised Intel Xeon Platinum 8360Y processors running at 2.40 GHz.

The code for ParEGO, NParEGO, TS-TCH, EHVI, NEHVI, and JES is available at `https://github.com/pytorch/botorch`. Specifically, we implement ParEGO and NParEGO based on the implementation provided by BoTorch.[4] We implement TS-TCH based on the implementation provided by BoTorch.[5] We implement EHVI and NEHVI based on the implementation provided by BoTorch.[6] We implement JES based on the implementation provided by BoTorch.[7]

### E.2 METHOD DETAILS

For all the methods, we set the same $2(d+1)$ points from a scrambled Sobol sequence and allow a maximum of 200 evaluations, following the practice in Daulton et al. (2020; 2021); Konakovic Lukovic et al. (2020). All the methods use $N = 128$ Monte Carlo samples.

For ParEGO (Knowles, 2006) and its noisy variant NParEGO (Daulton et al., 2021), we employ random scalarisations, whereby a weight vector $\boldsymbol{w} \in \mathbb{R}^m$ is generated from the unit simplex. The augmented Tchebycheff scalarisation function is applied, defined as $g(\boldsymbol{y}) = \max_i(w_i y_i) + \alpha \sum_i (w_i y_i)$. Log expected improvement and noisy log expected improvement are used as the acquisition functions in ParEGO and NParEGO, respectively, as recommended in Ament et al. (2023). In the batch setting, $q$ distinct weight vectors are sampled, and the acquisition function is optimised sequentially for each.

Table 4: Reference points of the six benchmark problems used for hypervolume computation in EHVI and NEHVI, as well as for performance evaluation. Note that the referents points of the two real-world problems, i.e., car side impact design and car cab design, are set to $(1.1, ..., 1.1) \in \mathbb{R}^m$ in the normalised objective space following the practice in Tanabe & Ishibuchi (2020) the utopian and nadir points are available at `https://github.com/ryojitanabe/reproblems/tree/master/ideal_nadir_points`).

| Problem | Reference point | Suggested by |
|---|---|---|
| DTLZ1 | $(400.0, ..., 400.0) \in \mathbb{R}^m$ | Balandat et al. (2020); Chugh (2020) |
| DTLZ2 | $(1.1, \ldots, 1.1) \in \mathbb{R}^m$ | Balandat et al. (2020); Daulton et al. (2020); Ishibuchi et al. (2018) |
| Inverted DTLZ1 | $(400.0, \ldots, 400.0) \in \mathbb{R}^m$ | Chugh (2020); Ishibuchi et al. (2018) |
| Inverted DTLZ2 | $(1.1, \ldots, 1.1) \in \mathbb{R}^m$ | Ishibuchi et al. (2018) |
| Convex DTLZ2 | $(1.1, \ldots, 1.1) \in \mathbb{R}^m$ | Ishibuchi et al. (2018) |
| Scaled DTLZ2 | $(1.1 * 2^0, \ldots, 1.1 * 2^m) \in \mathbb{R}^m$ | Ishibuchi et al. (2018) |
| DTLZ3 | $(10000.0, \ldots, 10000.0) \in \mathbb{R}^m$ | Balandat et al. (2020) |
| DTLZ4 | $(1.1, \ldots, 1.1) \in \mathbb{R}^m$ | Balandat et al. (2020) |
| DTLZ5 | $(10.0, \ldots, 10.0) \in \mathbb{R}^m$ | Balandat et al. (2020) |
| DTLZ6 | $(10.0, \ldots, 10.0) \in \mathbb{R}^m$ | Balandat et al. (2020) |
| DTLZ7 | $(15.0, \ldots, 15.0) \in \mathbb{R}^m$ | Balandat et al. (2020) |

For TS-TCH (Paria et al., 2020), similarly to ParEGO, we employ random scalarisations by using the augmented Tchebycheff scalarisation function. After converting to single-objective optimisation problem, Thompson sampling is used as the acquisition function. We draw a sample from the joint posterior over a discrete set of $1000d$ points sampled from a scrambled Sobol sequence, suggested by Daulton et al. (2020). In the batch setting, $q$ distinct weight vectors are sampled, and the acquisition function is optimised sequentially for each.

For EHVI (Daulton et al., 2020) and its noisy variant NEHVI (Daulton et al., 2021), the reference point is predefined, following the practice in Daulton et al. (2020); Yang et al. (2019b). Table 4 lists the reference points used for each problem in the experimental evaluation. The logarithmic variants of both acquisition functions are employed, as recommended in Ament et al. (2023). In the batch setting, the sequential greedy optimisation strategy is adopted. It is worth noting that in this study, EHVI and NEHVI are evaluated only on problems with 3 and 5 objectives, as the acquisition optimisation wall time becomes prohibitively high when the number of objectives increases to 10 (see Table 23).

For C-EHVI (Gaudrie et al., 2018; 2020), the Kalai-Smorodinsky equilibrium is used to determine the reference point of HV. The logarithmic variant of the acquisition functions is employed, as recommended in Ament et al. (2023).

---

[4]EI, NEI, and the BoTorch multi-objective tutorial.

[5]TS and BoTorch multi-objective tutorial

[6]EHVI, NEHVI, and BoTorch multi-objective tutorial

[7]Botorch JES implementation

For joint entropy search (JES), we use $S = 10$ Monte Carlo samples and $p = 10$ number of Pareto optimal points, according to Tu et al. (2022). In the batch setting, the sequential greedy optimisation strategy is adopted.

For all the problems, we normalise the input variables and standardise the objective values before training Gaussian processes. We assume an independent surrogate model for each objective, using a constant mean function and a Matérn 5/2 ARD kernel. We optimise all acquisition functions by using the L-BFGS-B, with up to 200 iterations.

### E.3    PROBLEM DETAILS

The details of the benchmark problems and real-world problems are given in the following.

**DTLZ1.**   DTLZ1 is a scalable benchmark problem from Deb et al. (2005), which is defined as:

$$f_1(\boldsymbol{x}) = \frac{1}{2}x_1 x_2 \cdots x_{m-1}(1 + g(\boldsymbol{x}_m)),$$

$$f_2(\boldsymbol{x}) = \frac{1}{2}x_1 x_2 \cdots (1 - x_{m-1})(1 + g(\boldsymbol{x}_m)),$$

$$\vdots$$

$$f_{m-1}(\boldsymbol{x}) = \frac{1}{2}x_1(1 - x_2)(1 + g(\boldsymbol{x}_m)),$$

$$f_m(\boldsymbol{x}) = \frac{1}{2}(1 - x_1)(1 + g(\boldsymbol{x}_m)),$$

$$\text{s.t.} \quad 0 \leq x_i \leq 1, \quad \text{for } i = 1, 2, \ldots, d.$$

where $g(\boldsymbol{x}_m) = 100 \left(|\boldsymbol{x}_m| + \sum_{x_i \in \boldsymbol{x}_m} (x_i - 0.5)^2 - \cos(20\pi(x_i - 0.5))\right)$ and $\boldsymbol{x}_m$ is the last $d - m + 1$ variables.

**DTLZ2.**   DTLZ2 is a scalable benchmark problem from Deb et al. (2005), which is defined as:

$$f_1(\boldsymbol{x}) = (1 + g(\boldsymbol{x}_m)) \cos(x_1\pi/2) \cdots \cos(x_{m-2}\pi/2) \cos(x_{m-1}\pi/2),$$
$$f_2(\boldsymbol{x}) = (1 + g(\boldsymbol{x}_m)) \cos(x_1\pi/2) \cdots \cos(x_{m-2}\pi/2) \sin(x_{m-1}\pi/2),$$
$$f_3(\boldsymbol{x}) = (1 + g(\boldsymbol{x}_m)) \cos(x_1\pi/2) \cdots \sin(x_{m-2}\pi/2),$$

$$\vdots$$

$$f_m(\boldsymbol{x}) = (1 + g(\boldsymbol{x}_m)) \sin(x_1\pi/2),$$
$$\text{s.t.} \quad 0 \leq x_i \leq 1, \quad \text{for } i = 1, 2, \ldots, d.$$

where $g(\boldsymbol{x}_m) = \sum_{x_i \in \boldsymbol{x}_m} (x_i - 0.5)^2$ and $\boldsymbol{x}_m$ is the last $d - m + 1$ variables.

**Inverted DTLZ1.**   Inverted DTLZ1 is a variant of DTLZ1 (Jain & Deb, 2013a), which is defined as:

$$f_i(\boldsymbol{x}) = 0.5 \cdot (1 + g(\boldsymbol{x}_m)) - f_i^{\text{DTLZ1}}(\boldsymbol{x}), \quad i = 1, \ldots, m$$

where $g(\boldsymbol{x}_m)$ is the same function as used in DTLZ1, $f_i^{\text{DTLZ1}}(\boldsymbol{x})$ denotes the $i$th objective of the original DTLZ1 formulation.

**Inverted DTLZ2.**   Inverted DTLZ2 is a variant of DTLZ2 Jain & Deb (2013b), which is defined as:

$$f_i(\boldsymbol{x}) = 1 + g(\boldsymbol{x}_m) - f_i^{\text{DTLZ2}}(\boldsymbol{x}), \quad i = 1, \ldots, m$$

where $g(\boldsymbol{x}_m)$ is the same function as used in DTLZ2, $f_i^{\text{DTLZ2}}(\boldsymbol{x})$ denotes the $i$th objective of the original DTLZ2 formulation.

**Convex DTLZ2.**   Convex DTLZ2 is a variant of DTLZ2 (Deb & Jain, 2013), which is defined as:

$$f_i(\boldsymbol{x}) = (f_i^{\text{DTLZ2}}(\boldsymbol{x}))^4, \quad i = 1, \ldots, m - 1$$
$$f_m(\boldsymbol{x}) = (f_m^{\text{DTLZ2}}(\boldsymbol{x}))^2$$

where $f_i^{\text{DTLZ2}}(\boldsymbol{x})$ denotes the $i$th objective of the original DTLZ2 formulation. This problem convert the original concave problem to convex problem.

**Scaled DTLZ2.** Scaled DTLZ2 is a variant of DTLZ2, which is defined as:

$$f_i(\boldsymbol{x}) = 2^{i-1} \cdot f_i^{\text{DTLZ2}}(\boldsymbol{x}), \quad i = 1, \dots, m$$

where $f_i^{\text{DTLZ2}}(\boldsymbol{x})$ denotes the $i$th objective of the original DTLZ2 formulation. This benchmark problem is used to see whether an algorithm can deal with problems with different scales of different objectives.

**DTLZ3–DTLZ7.** DTLZ3–DTLZ7 are scalable multi-objective benchmark problems. Their mathematical formulations are provided in Deb et al. (2005).

According to the original paper (Deb et al., 2005), the dimensionality $d$ of DTLZ1 and its variant (i.e., inverted DTLZ1) is $m + 4$, the dimensionality $d$ of DTLZ2-6 and their variants (i.e., inverted DTLZ2, convex DTLZ2, and scaled DTLZ2) is $m + 9$, and the dimensionality $d$ of DTLZ7 is $m + 19$.

**Car Side Impact Design.** The car side-impact problem aims to minimise vehicle weight while satisfying safety constraints related to occupant injury and structural response (Jain & Deb, 2013a). It involves $m = 4$ objectives with $d = 7$ variables, which are based on a surrogate model that is fit to data collected from a simulator. The mathematical formulations are given as follows:

$$
\begin{aligned}
f_1(\boldsymbol{x}) &= 1.98 + 4.9x_1 + 6.67x_2 + 6.98x_3 + 4.01x_4 + 1.78x_5 + 10^{-5}x_6 + 2.73x_7 \\
f_2(\boldsymbol{x}) &= 4.72 - 0.5x_4 - 0.19x_2x_3 \\
f_3(\boldsymbol{x}) &= 0.5\left(V_{\text{MBP}}(\boldsymbol{x}) + V_{\text{FD}}(\boldsymbol{x})\right) \\
f_4(\boldsymbol{x}) &= -\sum_{i=1}^{10} \max\left(g_i(\boldsymbol{x}), 0\right)
\end{aligned}
$$

where the constraint functions $g_i(\boldsymbol{x})$ are defined as:

$$
\begin{aligned}
g_1(\boldsymbol{x}) &= 1 - 1.16 + 0.3717x_2x_4 + 0.0092928x_3 \\
g_2(\boldsymbol{x}) &= 0.32 - 0.261 + 0.0159x_1x_2 + 0.06486x_1 + 0.019x_2x_7 - 0.0144x_3x_5 - 0.0154464x_6 \\
g_3(\boldsymbol{x}) &= 0.32 - 0.214 - 0.00817x_5 + 0.045195x_1 + 0.0135168x_1 - 0.03099x_2x_6 \\
&\quad + 0.018x_2x_7 - 0.007176x_3 - 0.023223x_3 + 0.00364x_5x_6 + 0.018x_2^2 \\
g_4(\boldsymbol{x}) &= 0.32 - 0.74 + 0.61x_2 + 0.031296x_3 + 0.031872x_7 - 0.227x_2^2 \\
g_5(\boldsymbol{x}) &= 32 - 28.98 - 3.818x_3 + 4.2x_1x_2 - 1.27296x_6 + 2.68065x_7 \\
g_6(\boldsymbol{x}) &= 32 - 33.86 - 2.95x_3 + 5.057x_1x_2 + 3.795x_2 + 3.4431x_7 - 1.45728 \\
g_7(\boldsymbol{x}) &= 32 - 46.36 + 9.9x_2 + 4.4505x_1 \\
g_8(\boldsymbol{x}) &= 4 - f_2(\boldsymbol{x}) \\
g_9(\boldsymbol{x}) &= 9.9 - V_{\text{MBP}}(\boldsymbol{x}) \\
g_{10}(\boldsymbol{x}) &= 15.7 - V_{\text{FD}}(\boldsymbol{x})
\end{aligned}
$$

with volume terms defined as:

$$
\begin{aligned}
V_{\text{MBP}}(\boldsymbol{x}) &= 10.58 - 0.674x_1x_2 - 0.67275x_2 \\
V_{\text{FD}}(\boldsymbol{x}) &= 16.45 - 0.489x_3x_7 - 0.8435x_6x_7.
\end{aligned}
$$

The search space is defined as:

$$
\begin{aligned}
x_1 &\in [0.5, 1.5], \\
x_2 &\in [0.45, 1.35], \\
x_3, x_4 &\in [0.5, 1.5], \\
x_5 &\in [0.875, 2.625], \\
x_6, x_7 &\in [0.4, 1.2].
\end{aligned}
$$

**Car Cab Design.** This vehicle performance optimisation problem involves $m = 9$ objectives with $d = 7$ variables, relating to aspects such as car roominess, fuel economy, acceleration time, and road noise at various speeds Deb & Jain (2013). The problem includes 7 decision variables and 4 stochastic variables, which are based on a surrogate model that is fit to data collected from a simulator, defined as:

$$f_1(\boldsymbol{x}) = 1.98 + 4.9x_1 + 6.67x_2 + 6.98x_3 + 4.01x_4 + 1.75x_5 + 10^{-5}x_6 + 2.73x_7$$

$$f_2(\boldsymbol{x}) = [1.16 - 0.3717x_2x_4 - 0.00931x_2x_{10} - 0.484x_3x_9 + 0.01343x_6x_{10}]_+$$

$$f_3(\boldsymbol{x}) = \left[\frac{1}{0.32}\left(\begin{array}{l}0.261 - 0.0159x_1x_2 - 0.188x_1x_8 - 0.019x_2x_7 + 0.0144x_3x_5 + 0.8757x_5x_{10} \\ + 0.08045x_6x_9 + 0.00139x_8x_{11} + 0.00001575x_{10}x_{11}\end{array}\right)\right]_+$$

$$f_4(\boldsymbol{x}) = \left[\frac{1}{0.32}\left(\begin{array}{l}0.214 + 0.00817x_5 - 0.131x_1x_8 - 0.0704x_1x_9 + 0.03099x_2x_6 - 0.018x_2x_7 \\ + 0.0208x_3x_8 + 0.121x_3x_9 - 0.00364x_5x_6 + 0.0007715x_5x_{10} \\ - 0.0005354x_6x_{10} + 0.00121x_8x_{11} + 0.00184x_9x_{10} - 0.018x_2^2\end{array}\right)\right]_+$$

$$f_5(\boldsymbol{x}) = \left[\frac{0.74 - 0.61x_2 - 0.163x_3x_8 + 0.001232x_3x_{10} - 0.166x_7x_9 + 0.227x_2^2}{0.32}\right]_+$$

$$f_6(\boldsymbol{x}) = \left[\frac{1}{32}\cdot\frac{1}{3}\left(\begin{array}{l}28.98 + 3.818x_3 - 4.2x_1x_2 + 0.0207x_5x_{10} + 6.63x_6x_9 - 7.77x_7x_8 + 0.32x_9x_{10} \\ + 33.86 + 2.95x_3 + 0.1792x_{10} - 5.057x_1x_2 - 11x_2x_8 - 0.0215x_5x_{10} - 9.98x_7x_8 \\ + 22x_8x_9 + 46.36 - 9.9x_2 - 12.9x_1x_8 + 0.1107x_3x_{10}\end{array}\right)\right]_+$$

$$f_7(\boldsymbol{x}) = \left[\frac{4.72 - 0.5x_4 - 0.19x_2x_3 - 0.0122x_4x_{10} + 0.009325x_6x_{10} + 0.000191x_{11}^2}{4.0}\right]_+$$

$$f_8(\boldsymbol{x}) = \left[\frac{10.58 - 0.674x_1x_2 - 1.95x_2x_8 + 0.02054x_3x_{10} - 0.0198x_4x_{10} + 0.028x_6x_{10}}{9.9}\right]_+$$

$$f_9(\boldsymbol{x}) = \left[\frac{16.45 - 0.489x_3x_7 - 0.843x_5x_6 + 0.0432x_9x_{10} - 0.0556x_9x_{11} - 0.000786x_{11}^2}{15.7}\right]_+$$

where $[\cdot]_+$ denotes $\max(0, \cdot)$ and the search space is defined as:

$$x_1 \in [0.5, \ 1.5],$$
$$x_2 \in [0.45, \ 1.35],$$
$$x_3, \ x_4 \in [0.5, \ 1.5],$$
$$x_5 \in [0.875, \ 2.625],$$
$$x_6, \ x_7 \in [0.4, \ 1.2].$$

The four stochastic variables are defined as:

$$x_8 \sim \mathcal{N}(0.345, \ 0.006^2),$$
$$x_9 \sim \mathcal{N}(0.192, \ 0.006^2),$$
$$x_{10}, \ x_{11} \sim \mathcal{N}(0, \ 10^2).$$

# F  ADDITIONAL EXPERIMENTAL RESULTS

## F.1  NOISELESS CASES

In this section, we present the results on the six noiseless problems (i.e., DTLZ1 and DTLZ2 along with their four variants) with 3 and 10 objectives. Tables 5, 6 and 7 show the distance-based metric (log distance), the HV of the best solution (in terms of its HV value) and the HV of all evaluated solutions obtained by the SPMO and the peer methods on the six noiseless problems, respectively. Figures 6, 7 and 8 present the violin plots, illustrating the distributions of the corresponding results reported in Tables 5, 6, and 7, respectively. In addition, Figure 9 presents the trajectories of the distance metric obtained by each method on the noiseless problems with 3 and 10 objectives. We also give the results of the problems DTLZ3–DLTZ7 with 5 objectives, shown in Tables 8, 9 and 10.

Table 5: Results of the distance-based metric (log distance) obtained by the SPMO and the peer methods on the noiseless problems with 3 objectives (**top**) and 10 objectives (**bottom**) on 30 independent runs. The method with the best mean is highlighted in bold. The symbols "+", "∼" and "−" indicate that the method is statistically worse than, equivalent to and better than our SPMO, respectively.

| Method | DTLZ1 (3) Mean (Std) | DTLZ2 (3) Mean (Std) | Inverted DTLZ1 (3) Mean (Std) | Inverted DTLZ2 (3) Mean (Std) | Convex DTLZ2 (3) Mean (Std) | Scaled DTLZ2 (3) Mean (Std) | Sum up +/∼/− |
|---|---|---|---|---|---|---|---|
| Sobol | 3.8e+0 (3.3e−1)$^+$ | 2.3e−1 (5.1e−2)$^+$ | 4.4e+0 (4.0e−1)$^+$ | 5.7e−2 (5.3e−2)$^+$ | −7.7e−2 (1.7e−1)$^+$ | 2.2e−1 (4.8e−2)$^+$ | 6/ 0/ 0 |
| ParEGO | 3.7e+0 (2.0e−1)$^+$ | 4.7e−3 (3.1e−3)$^+$ | 3.5e+0 (5.7e−1)$^+$ | −3.0e−1 (7.7e−3)$^+$ | −9.3e−1 (1.2e−1)$^+$ | 5.2e−3 (4.7e−3)$^+$ | 6/ 0/ 0 |
| TS-TCH | 3.9e+0 (2.6e−1)$^+$ | 1.4e−1 (7.1e−2)$^+$ | 4.3e+0 (3.6e−1)$^+$ | −1.1e−1 (2.4e−2)$^+$ | −3.4e−1 (1.3e−1)$^+$ | 1.5e−1 (3.6e−2)$^+$ | 6/ 0/ 0 |
| EHVI | 3.6e+0 (1.4e−1)$^+$ | 4.6e−3 (2.5e−3)$^+$ | 4.0e+0 (2.1e−1)$^+$ | −3.0e−1 (4.3e−3)$^+$ | −9.1e−1 (1.1e−1)$^+$ | 1.8e−2 (6.0e−2)$^+$ | 6/ 0/ 0 |
| C-EHVI | 3.6e+0 (1.2e−1)$^+$ | 1.6e−3 (1.7e−3)$^+$ | 4.0e+0 (2.6e−1)$^+$ | −2.8e−1 (2.9e−2)$^+$ | −4.5e−1 (2.4e−1)$^+$ | 3.2e−3 (5.5e−3)$^+$ | 6/ 0/ 0 |
| JES | 3.5e+0 (1.1e−1)$^+$ | 5.5e−3 (4.2e−3)$^+$ | 4.1e+0 (1.7e−1)$^+$ | −3.0e−1 (4.8e−3)$^+$ | −9.1e−1 (9.0e−2)$^+$ | 9.6e−3 (8.7e−3)$^+$ | 6/ 0/ 0 |
| SPMO | **3.3e+0 (2.9e−1)** | **2.2e−4 (1.2e−4)** | **2.8e+0 (6.6e−1)** | **−3.1e−1 (1.3e−5)** | **−1.2e+0 (1.4e−2)** | **3.8e−5 (1.6e−5)** | |

| Method | DTLZ1 (10) Mean (Std) | DTLZ2 (10) Mean (Std) | Inverted DTLZ1 (10) Mean (Std) | Inverted DTLZ2 (10) Mean (Std) | Convex DTLZ2 (10) Mean (Std) | Scaled DTLZ2 (10) Mean (Std) | Sum up +/∼/− |
|---|---|---|---|---|---|---|---|
| Sobol | 3.8e+0 (2.4e−1)$^+$ | 2.4e−1 (4.1e−2)$^+$ | 5.2e+0 (2.5e−1)$^+$ | 1.2e+0 (5.4e−2)$^+$ | −4.5e−1 (2.2e−1)$^+$ | 2.4e−1 (5.1e−2)$^+$ | 6/ 0/ 0 |
| ParEGO | 3.7e+0 (2.6e−1)$^+$ | 1.2e−1 (8.3e−2)$^+$ | 3.5e+0 (4.7e−1)$^∼$ | 8.6e−1 (1.9e−2)$^+$ | −1.8e+0 (2.3e−1)$^+$ | 1.1e−1 (6.3e−2)$^+$ | 5/ 1/ 0 |
| TS-TCH | 3.8e+0 (3.9e−1)$^+$ | 2.1e−1 (2.3e−2)$^+$ | 5.3e+0 (3.7e−1)$^+$ | 1.0e+0 (1.4e−2)$^+$ | −6.7e−1 (2.5e−1)$^+$ | 2.1e−1 (3.6e−2)$^+$ | 6/ 0/ 0 |
| C-EHVI | 3.7e+0 (1.5e−1)$^+$ | 6.4e−3 (5.1e−3)$^+$ | 3.9e+0 (5.1e−1)$^+$ | 8.4e−1 (1.6e−2)$^+$ | −5.5e−1 (3.2e−1)$^+$ | 5.5e−2 (7.5e−2)$^+$ | 6/ 0/ 0 |
| JES | 3.4e+0 (2.1e−1)$^+$ | 1.5e−1 (6.6e−2)$^+$ | 4.5e+0 (5.3e−1)$^+$ | 8.6e−1 (2.0e−2)$^+$ | −1.7e+0 (3.0e−1)$^+$ | 1.2e−1 (8.5e−2)$^+$ | 6/ 0/ 0 |
| SPMO | **2.8e+0 (5.5e−1)** | **1.3e−3 (2.6e−3)** | **3.4e+0 (5.1e−1)** | **7.8e−1 (2.4e−2)** | **−3.2e+0 (1.5e−1)** | **1.7e−2 (3.7e−2)** | |

Table 6: The HV of the best solution (in terms of its HV value) obtained by the proposed SPMO and the peer methods on the noiseless problems with 3 objectives (**top**) and 10 objectives (**bottom**) on 30 independent runs. The method with the best mean is highlighted in bold. The symbols "+", "∼" and "−" indicate that the method is statistically worse than, equivalent to and better than our SPMO, respectively.

| Method | DTLZ1 (3) Mean (Std) | DTLZ2 (3) Mean (Std) | Inverted DTLZ1 (3) Mean (Std) | Inverted DTLZ2 (3) Mean (Std) | Convex DTLZ2 (3) Mean (Std) | Scaled DTLZ2 (3) Mean (Std) | Sum up +/∼/− |
|---|---|---|---|---|---|---|---|
| Sobol | 5.3e+7 (3.3e+6)$^+$ | 3.3e−2 (2.1e−2)$^+$ | 4.4e+7 (5.7e+6)$^+$ | 1.1e−1 (2.5e−2)$^+$ | 1.9e−1 (1.1e−1)$^+$ | 3.7e−2 (1.8e−2)$^+$ | 6/ 0/ 0 |
| ParEGO | 5.6e+7 (1.5e+6)$^+$ | 1.6e−1 (7.0e−3)$^+$ | 5.5e+7 (4.2e+6)$^+$ | 3.1e−1 (3.7e−3)$^+$ | 7.2e−1 (5.1e−2)$^+$ | **1.6e−1 (9.5e−3)**$^−$ | 5/ 0/ 1 |
| TS-TCH | 5.3e+7 (3.1e+6)$^+$ | 5.0e−2 (3.1e−2)$^+$ | 4.6e+7 (5.2e+6)$^+$ | 2.0e−1 (1.2e−2)$^+$ | 3.7e−1 (8.4e−2)$^+$ | 4.8e−2 (2.5e−2)$^+$ | 6/ 0/ 0 |
| EHVI | 5.7e+7 (7.7e+5)$^+$ | 1.6e−1 (4.6e−3)$^+$ | 5.1e+7 (2.1e+6)$^+$ | 3.1e−1 (2.2e−3)$^+$ | 6.8e−1 (4.6e−2)$^+$ | 1.3e−1 (3.4e−2)$^−$ | 5/ 0/ 1 |
| C-EHVI | 5.7e+7 (8.3e+5)$^+$ | 1.3e−1 (1.3e−2)$^+$ | 5.1e+7 (2.4e+6)$^+$ | 3.0e−1 (1.4e−2)$^+$ | 4.5e−1 (1.4e−1)$^+$ | 1.4e−1 (1.7e−2)$^−$ | 5/ 0/ 1 |
| JES | 5.6e+7 (8.2e+5)$^+$ | 1.5e−1 (8.3e−3)$^+$ | 5.0e+7 (2.0e+6)$^+$ | 3.0e−1 (2.4e−3)$^+$ | 7.1e−1 (3.6e−2)$^+$ | 1.5e−1 (8.1e−3)$^−$ | 5/ 0/ 1 |
| SPMO | **5.8e+7 (1.7e+6)** | **1.7e−1 (5.3e−3)** | **5.9e+7 (3.5e+6)** | **3.1e−1 (6.4e−6)** | **7.9e−1 (5.1e−3)** | 1.2e−1 (7.5e−3) | |

| Method | DTLZ1 (10) Mean (Std) | DTLZ2 (10) Mean (Std) | Inverted DTLZ1 (10) Mean (Std) | Inverted DTLZ2 (10) Mean (Std) | Convex DTLZ2 (10) Mean (Std) | Scaled DTLZ2 (10) Mean (Std) | Sum up +/∼/− |
|---|---|---|---|---|---|---|---|
| Sobol | 8.7e+25 (3.8e+24)$^+$ | 4.7e−2 (2.5e−2)$^+$ | 2.4e+25 (9.4e+24)$^+$ | 5.9e−11 (3.2e−10)$^+$ | 7.7e−1 (2.4e−1)$^+$ | 5.0e−2 (2.1e−2)$^+$ | 6/ 0/ 0 |
| ParEGO | 9.2e+25 (3.3e+24)$^+$ | 1.2e−1 (9.6e−2)$^+$ | 7.8e+25 (1.1e+25)$^∼$ | 2.2e−5 (1.4e−5)$^+$ | 1.9e+0 (1.0e−1)$^+$ | 1.3e−1 (8.3e−2)$^+$ | 5/ 1/ 0 |
| TS-TCH | 8.8e+25 (5.1e+24)$^+$ | 4.3e−2 (4.0e−2)$^+$ | 2.2e+25 (1.3e+25)$^+$ | 1.6e−8 (2.6e−8)$^+$ | 1.0e+0 (2.5e−1)$^+$ | 4.1e−2 (2.5e−2)$^+$ | 6/ 0/ 0 |
| C-EHVI | 9.2e+25 (1.0e+24)$^+$ | 2.6e−1 (3.2e−2)$^+$ | 7.0e+25 (1.3e+25)$^+$ | 3.4e−5 (1.4e−5)$^+$ | 9.1e−1 (3.3e−1)$^+$ | 2.0e−1 (9.0e−2)$^+$ | 6/ 0/ 0 |
| JES | 9.3e+25 (1.9e+24)$^+$ | 9.9e−2 (7.2e−2)$^+$ | 4.9e+25 (1.8e+25)$^+$ | 2.1e−5 (1.1e−5)$^+$ | 1.8e+0 (1.3e−1)$^+$ | 1.3e−1 (9.7e−2)$^+$ | 6/ 0/ 0 |
| SPMO | **9.7e+25 (3.6e+24)** | **2.9e−1 (3.4e−2)** | **8.2e+25 (9.3e+24)** | **1.3e−4 (5.1e−5)** | **2.3e+0 (3.5e−2)** | **2.6e−1 (4.6e−2)** | |

Table 7: The HV of all the solutions obtained by the proposed SPMO and the peer methods on the noiseless problems with 3 objectives (**top**) and 10 objectives (**bottom**) on 30 independent runs, respectively. The method with the best mean is highlighted in bold. The symbols "+", "∼", and "−" indicate that a method is statistically worse than, equivalent to, and better than SPMO, respectively.

| Method | DTLZ1 (3) Mean (Std) | DTLZ2 (3) Mean (Std) | Inverted DTLZ1 (3) Mean (Std) | Inverted DTLZ2 (3) Mean (Std) | Convex DTLZ2 (3) Mean (Std) | Scaled DTLZ2 (3) Mean (Std) | Sum up +/∼/− |
|---|---|---|---|---|---|---|---|
| Sobol | 6.3e+7 (2.8e+5)∼ | 4.9e−2 (2.7e−2)+ | 5.2e+7 (2.9e+6)+ | 2.1e−1 (2.6e−2)+ | 2.5e−1 (1.3e−1)+ | 6.2e−2 (2.5e−2)+ | 5/ 1/ 0 |
| ParEGO | 6.3e+7 (1.1e+6)− | 5.4e−1 (5.2e−2)− | 5.8e+7 (2.6e+6)+ | 6.7e−1 (1.1e−2)− | 1.0e+0 (9.0e−2)∼ | **5.4e−1 (7.5e−2)**− | 1/ 1/ 4 |
| TS-TCH | 6.2e+7 (7.8e+5)∼ | 7.5e−2 (4.2e−2)+ | 5.2e+7 (3.3e+6)+ | 4.1e−1 (1.2e−2)− | 5.4e−1 (1.1e−1)+ | 8.8e−2 (4.6e−2)+ | 4/ 1/ 1 |
| EHVI | **6.4e+7 (1.0e+5)**− | **6.4e−1 (2.2e−2)**− | 5.8e+7 (2.4e+6)+ | **7.0e−1 (3.9e−3)**− | **1.0e+0 (6.9e−2)**∼ | 2.6e−1 (7.7e−2)− | 1/ 1/ 4 |
| C-EHVI | 6.3e+7 (6.8e+5)− | 3.3e−1 (5.4e−2)+ | 5.7e+7 (3.2e+6)+ | 3.2e−1 (3.5e−2)+ | 5.7e−1 (2.0e−1)+ | 3.3e−1 (6.5e−2)− | 4/ 0/ 2 |
| JES | **6.4e+7 (5.0e+4)**− | 5.6e−1 (5.3e−2)− | 6.1e+7 (1.6e+6)∼ | 6.6e−1 (1.1e−2)− | 1.0e+0 (6.3e−2)∼ | 5.0e−1 (8.5e−2)− | 0/ 2/ 4 |
| SPMO | 6.2e+7 (1.0e+6) | 4.6e−1 (6.3e−2) | **6.1e+7 (1.5e+6)** | 3.7e−1 (1.5e−2) | 1.0e+0 (4.2e−2) | 1.6e−1 (5.2e−2) | |

| Method | DTLZ1 (10) Mean (Std) | DTLZ2 (10) Mean (Std) | Inverted DTLZ1 (10) Mean (Std) | Inverted DTLZ2 (10) Mean (Std) | Convex DTLZ2 (10) Mean (Std) | Scaled DTLZ2 (10) Mean (Std) | Sum up +/∼/− |
|---|---|---|---|---|---|---|---|
| Sobol | 2.4e+28 (1.3e+28)− | 3.6e−1 (1.7e−1)+ | 5.3e+25 (5.9e+25)+ | 5.9e−11 (3.2e−10)+ | 5.2e+0 (3.2e+0)+ | 3.6e−1 (1.6e−1)+ | 5/ 0/ 1 |
| ParEGO | 9.0e+26 (9.8e+26)+ | 1.7e+0 (3.5e+0)+ | 3.6e+27 (1.3e+28)+ | 7.2e−4 (4.9e−4)+ | 2.7e+2 (2.2e+2)+ | 6.8e−1 (7.6e−1)+ | 6/ 0/ 0 |
| TS-TCH | **2.9e+28 (2.5e+28)**− | 1.8e−1 (2.0e−1)+ | 4.7e+25 (4.4e+25)+ | 1.9e−8 (3.3e−8)+ | 2.8e+1 (2.4e+1)+ | 1.5e−1 (1.0e−1)+ | 5/ 0/ 1 |
| C-EHVI | 1.0e+26 (1.6e+24)+ | 5.2e−1 (1.2e−1)+ | 7.1e+25 (1.3e+25)+ | 6.6e−5 (2.6e−5)+ | 1.3e+0 (4.0e−1)+ | 3.8e−1 (2.2e−1)+ | 6/ 0/ 0 |
| JES | 8.8e+26 (1.3e+27)+ | 6.3e−1 (1.1e+0)+ | 5.7e+27 (1.9e+28)+ | 7.5e−4 (3.9e−4)+ | 2.8e+2 (1.9e+2)+ | 1.5e+0 (2.2e+0)+ | 6/ 0/ 0 |
| SPMO | 1.2e+28 (1.5e+28) | **5.2e+1 (3.3e+1)** | **2.3e+28 (5.3e+28)** | **1.4e−1 (1.5e−1)** | **6.3e+3 (8.0e+3)** | **5.2e+1 (6.5e+1)** | |

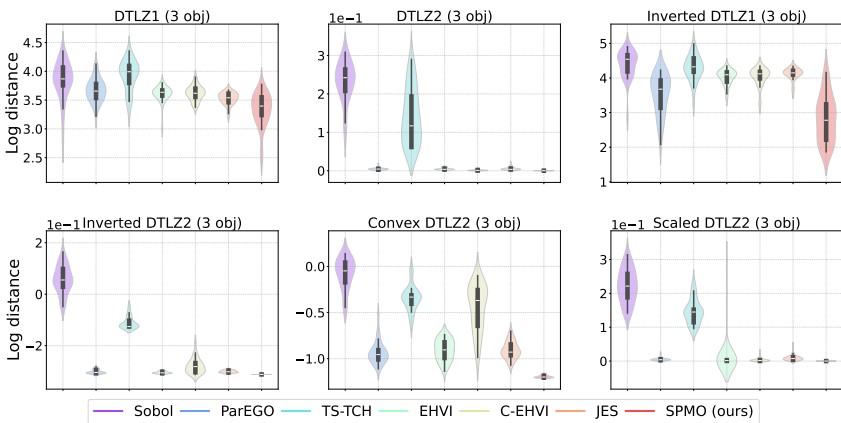

(a) Violin plots of the distance-based metric (log distance) obtained by each method on problems with 3 objectives.

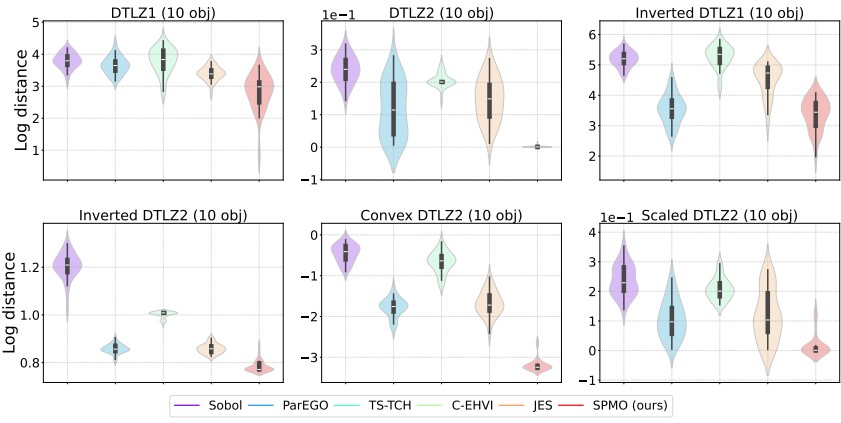

(b) Violin plots of the distance-based metric (log distance) obtained by each method on problems with 10 objectives.

Figure 6: Violin plots of the distance-based metric (log distance) obtained by the proposed SPMO and the peer methods on the noiseless problems with 3 and 10 objectives. Each violin represents the distribution of the distance-based metric obtained by a method over 30 independent runs.

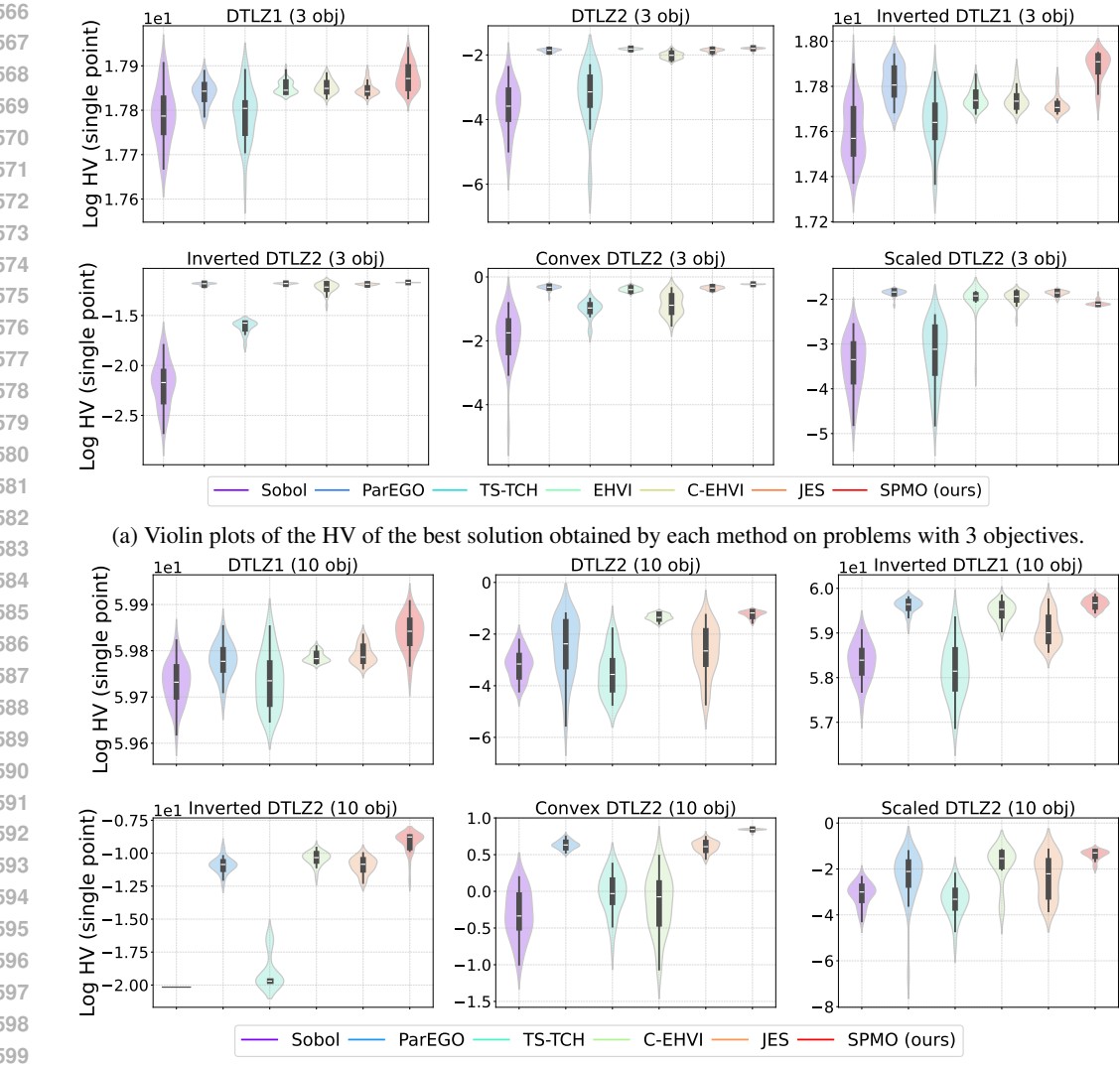

(a) Violin plots of the HV of the best solution obtained by each method on problems with 3 objectives.

(b) Violin plots of the HV of the best solution obtained by each method on problems with 10 objectives.

Figure 7: Violin plots of the HV of the best solution (in terms of its HV value) obtained by the proposed SPMO and the peer methods on the noiseless problems with 3 and 10 objectives. Each violin represents the distribution of HV values obtained by a method over 30 independent runs.

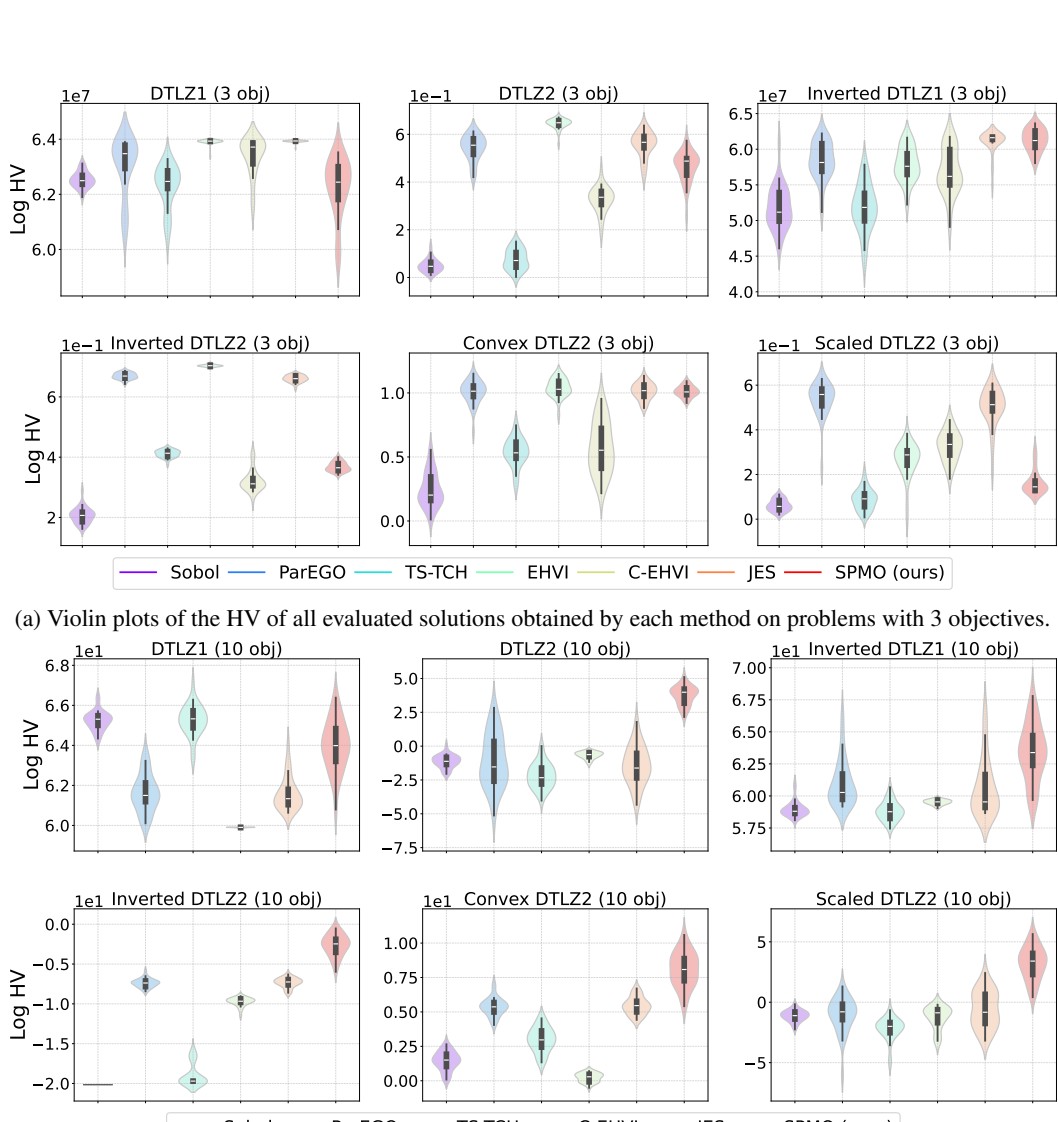

(a) Violin plots of the HV of all evaluated solutions obtained by each method on problems with 3 objectives.

(b) Violin plots of the HV of all evaluated solutions obtained by each method on problems with 10 objectives.

Figure 8: Violin plots of the HV of all evaluated solutions obtained by the proposed SPMO and the peer methods on the noiseless problems with 3 objectives (**top**) and 10 objectives (**bottom**), respectively. Each violin represents the distribution of HV values obtained by a method over 30 independent runs.

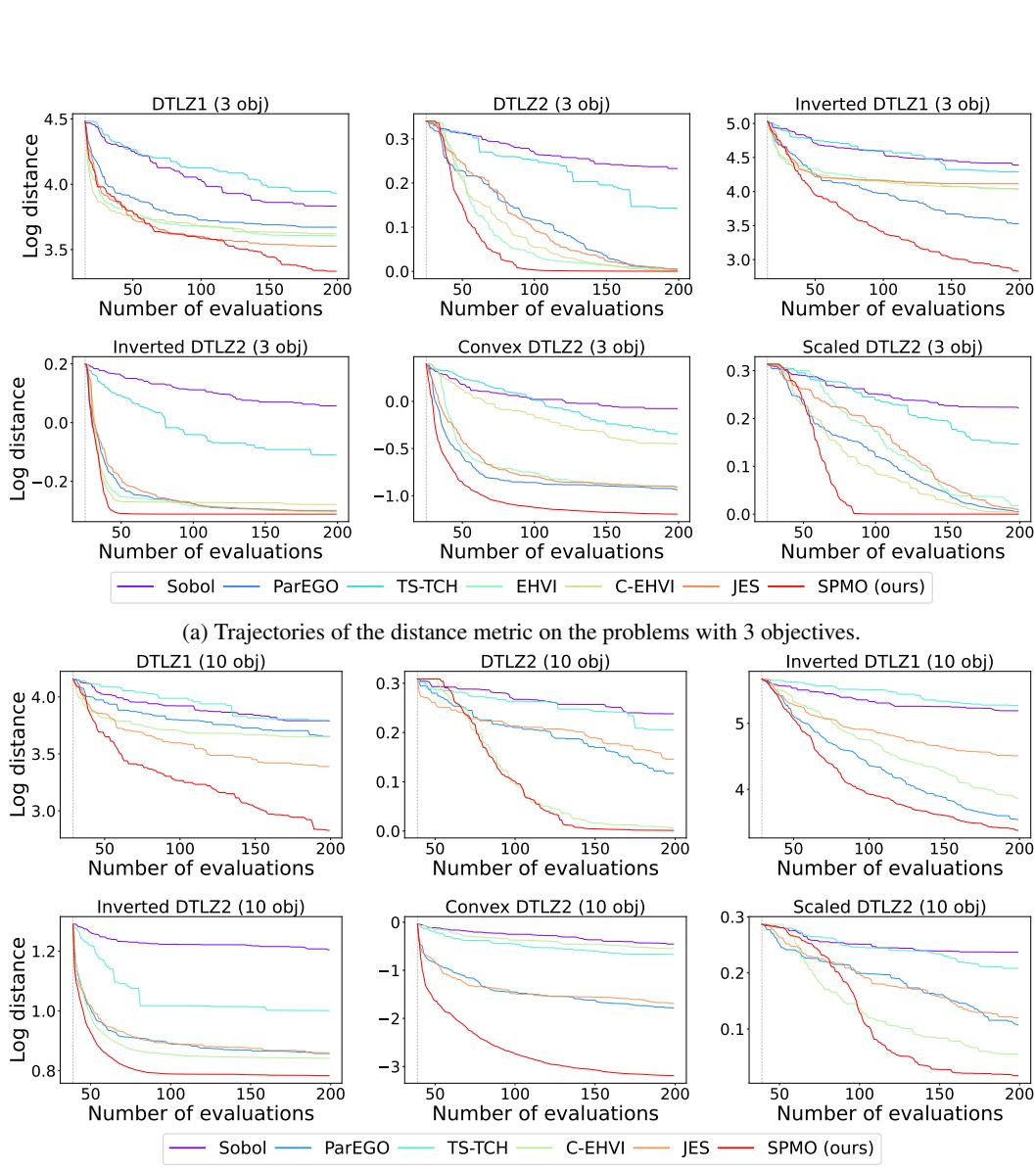

(a) Trajectories of the distance metric on the problems with 3 objectives.

(b) Trajectories of the distance metric on the problems with 10 objectives.

Figure 9: Trajectories of the distance metric obtained by the SPMO and the peer methods on the noiseless problems with 3 and 10 objectives. Each coloured line represents the mean distance of the closest solution to the utopian point on 30 independent runs (after the initial Sobol samples, represented by the dashed grey line).

Table 8: Results of the distance-based metric (log distance) obtained by obtained by the SPMO and the peer methods on DTLZ3–DTLZ7 with 5 objectives on 30 independent runs. The method with the best mean is highlighted in bold. The symbols "+", "∼" and "−" indicate that the method is statistically worse than, equivalent to and better than our SPMO, respectively.

| Method | DTLZ3 Mean (Std) | DTLZ4 Mean (Std) | DTLZ5 Mean (Std) | DTLZ6 Mean (Std) | DTLZ7 Mean (Std) | Sum up +/∼/− |
|---|---|---|---|---|---|---|
| Sobol | 6.3e+0 (1.6e–1)$^+$ | 2.7e–1 (6.0e–2)$^+$ | 2.5e–1 (4.1e–2)$^+$ | 2.2e+0 (2.1e–2)$^+$ | 3.1e+0 (5.0e–2)$^+$ | **5/ 0/ 0** |
| ParEGO | 5.5e+0 (1.1e–1)$^\sim$ | 2.3e–1 (7.1e–2)$^+$ | 4.0e–2 (3.1e–2)$^+$ | 4.3e–2 (1.6e–1)$^+$ | 2.2e+0 (8.0e–2)$^+$ | **4/ 1/ 0** |
| TS-TCH | 6.3e+0 (1.6e–1)$^+$ | 2.6e–1 (5.0e–2)$^+$ | 2.3e–1 (5.9e–2)$^+$ | 2.2e+0 (1.6e–2)$^+$ | 3.1e+0 (4.8e–2)$^+$ | **5/ 0/ 0** |
| EHVI | **5.4e+0 (4.6e–2)**$^-$ | 8.7e–2 (9.4e–2)$^+$ | 3.0e–1 (6.0e–2)$^+$ | -9.4e–7 (8.4e–7)$^+$ | 1.7e+0 (8.5e–2)$^+$ | **4/ 0/ 1** |
| JES | 5.5e+0 (5.0e–2)$^\sim$ | 2.3e–1 (8.1e–2)$^+$ | 5.2e–2 (5.8e–2)$^+$ | 2.1e–2 (1.1e–1)$^+$ | 2.2e+0 (5.8e–2)$^+$ | **4/ 1/ 0** |
| SPMO | 5.5e+0 (1.8e–1) | **1.2e–2 (1.9e–2)** | **3.2e–3 (5.0e–3)** | **-1.9e–6 (5.6e–7)** | **1.7e+0 (1.1e–1)** | |

Table 9: The HV of the best solution (in terms of its HV value) obtained by SPMO and the peer methods on the DTLZ3–DTLZ7 with 5 objectives on 30 independent runs. The method with the best mean is highlighted in bold. The symbols "+", "∼" and "−" indicate that the method is statistically worse than, equivalent to and better than our SPMO, respectively.

| Method | DTLZ3 Mean (Std) | DTLZ4 Mean (Std) | DTLZ5 Mean (Std) | DTLZ6 Mean (Std) | DTLZ7 Mean (Std) | Sum up +/∼/− |
|---|---|---|---|---|---|---|
| Sobol | 9.2e+19 (1.1e+18)$^+$ | 2.2e–2 (1.4e–2)$^+$ | 8.4e+4 (9.5e+2)$^\sim$ | 8.7e+3 (8.2e+2)$^+$ | -0.0e+0 (0.0e+0)$^+$ | **4/ 1/ 0** |
| ParEGO | 9.7e+19 (5.2e+17)$^\sim$ | 2.5e–2 (3.0e–2)$^+$ | **8.9e+4 (1.4e+3)** | 8.8e+4 (3.8e+3)$^+$ | 2.8e+5 (2.7e+4)$^+$ | **2/ 1/ 2** |
| TS-TCH | 9.3e+19 (1.0e+18)$^+$ | 1.3e–2 (1.1e–2)$^+$ | 8.6e+4 (1.1e+3)$^-$ | 9.3e+3 (9.9e+2)$^+$ | -0.0e+0 (0.0e+0)$^+$ | **4/ 0/ 1** |
| EHVI | **9.8e+19 (1.1e+17)**$^-$ | 1.3e–1 (5.6e–2)$^+$ | 8.3e+4 (1.6e+3)$^+$ | **9.0e+4 (1.3e–2)**$^-$ | 3.8e+5 (2.2e+4)$^\sim$ | **2/ 1/ 2** |
| JES | 9.8e+19 (1.4e+17)$^-$ | 3.6e–2 (3.5e–2)$^+$ | 8.9e+4 (1.5e+3)$^-$ | 8.8e+4 (3.5e+3)$^-$ | 2.8e+5 (2.0e+4)$^+$ | **2/ 0/ 3** |
| SPMO | 9.7e+19 (1.0e+18) | **1.6e–1 (2.5e–2)** | 8.5e+4 (2.6e+3) | 8.3e+4 (2.9e+3) | **3.8e+5 (2.8e+4)** | |

Table 10: The HV of all evaluated solutions obtained by SPMO and the peer methods on DTLZ3–DTLZ7 with 5 objectives on 30 independent runs. The method with the best mean is highlighted in bold. The symbols "+", "∼" and "−" indicate that the method is statistically worse than, equivalent to and better than our SPMO, respectively.

| Method | DTLZ3 Mean (Std) | DTLZ4 Mean (Std) | DTLZ5 Mean (Std) | DTLZ6 Mean (Std) | DTLZ7 Mean (Std) | Sum up +/∼/− |
|---|---|---|---|---|---|---|
| Sobol | 3.9e+20 (1.7e+20)$^\sim$ | 4.9e–2 (2.4e–2)$^+$ | 2.1e+5 (1.1e+5)$^\sim$ | 1.4e+5 (1.7e+4)$^+$ | -0.0e+0 (0.0e+0)$^+$ | **3/ 2/ 0** |
| ParEGO | 2.1e+20 (2.1e+20)$^\sim$ | 5.3e–2 (7.4e–2)$^+$ | 2.4e+5 (8.7e+4)$^\sim$ | 3.9e+5 (1.6e+5)$^+$ | 9.1e+5 (4.1e+5)$^+$ | **2/ 2/ 1** |
| TS-TCH | **5.0e+20 (3.0e+20)**$^-$ | 2.1e–2 (1.8e–2)$^+$ | 1.8e+5 (8.2e+4)$^+$ | 1.8e+5 (4.2e+4)$^+$ | -0.0e+0 (0.0e+0)$^+$ | **4/ 0/ 1** |
| EHVI | 1.7e+20 (1.5e+20)$^\sim$ | 9.3e–1 (7.1e–1)$^\sim$ | 3.5e+5 (1.8e+5)$^-$ | **4.2e+5 (1.2e+5)**$^-$ | 1.9e+6 (5.3e+5)$^\sim$ | **0/ 3/ 2** |
| JES | 2.7e+20 (2.7e+20)$^\sim$ | 8.8e–2 (1.3e–1)$^+$ | **3.6e+5 (3.5e+5)**$^\sim$ | 4.0e+5 (1.6e+5)$^-$ | 1.0e+6 (4.3e+5)$^+$ | **2/ 2/ 1** |
| SPMO | 4.2e+20 (7.0e+20) | **1.1e+0 (9.3e–1)** | 2.5e+5 (1.4e+5) | 3.1e+5 (1.9e+5) | **2.3e+6 (1.3e+6)** | |

F.2 NOISY CASES

In this section, we present the results on the noisy problems. Tables 11, 12 and 13 show the distance-based metric (log distance), the HV of the best solution (in terms of its HV value) and the HV of all evaluated solutions obtained by the SPMO and the peer methods, respectively. Figures 10, 11 and 12 present the violin plots, illustrating the distributions of the corresponding results reported in Tables 11, 12 and 13, respectively.

Table 11: Results of the distance-based metric (log distance) obtained by the SPMO and the peer methods on the noisy problems with 5 objectives on 30 independent runs. The method with the best mean is highlighted in bold. The symbols "+", "∼" and "−" indicate that the method is statistically worse than, equivalent to and better than our SPMO, respectively.

| Method | DTLZ1 Mean (Std) | DTLZ2 Mean (Std) | Inverted DTLZ1 Mean (Std) | Inverted DTLZ2 Mean (Std) | Convex DTLZ2 Mean (Std) | Scaled DTLZ2 Mean (Std) | Sum up +/∼/− |
|---|---|---|---|---|---|---|---|
| Sobol | $3.7e+0\ (4.2e-1)^+$ | $1.9e-1\ (5.6e-2)^+$ | $4.7e+0\ (3.4e-1)^+$ | $6.1e-1\ (6.1e-2)^+$ | $-2.9e-1\ (2.2e-1)^+$ | $2.2e-1\ (4.4e-2)^+$ | **6/ 0/ 0** |
| NParEGO | $3.6e+0\ (3.1e-1)^+$ | $1.0e-1\ (8.2e-2)^+$ | $3.1e+0\ (3.8e-1)^+$ | $1.8e-1\ (5.0e-2)^+$ | $-1.3e+0\ (2.7e-1)^+$ | $5.2e-2\ (5.9e-2)^+$ | **6/ 0/ 0** |
| TS-TCH | $3.8e+0\ (3.5e-1)^+$ | $2.0e-1\ (4.2e-2)^+$ | $4.9e+0\ (3.2e-1)^+$ | $4.6e-1\ (5.5e-2)^+$ | $-4.7e-1\ (2.3e-1)^+$ | $2.1e-1\ (5.0e-2)^+$ | **6/ 0/ 0** |
| NEHVI | $3.5e+0\ (1.4e-1)^+$ | $-1.1e-1\ (8.1e-2)^+$ | $4.0e+0\ (5.8e-1)^+$ | $1.5e-1\ (3.3e-2)^+$ | $-1.3e+0\ (3.0e-1)^+$ | $2.9e-1\ (9.5e-2)^+$ | **6/ 0/ 0** |
| JES | $3.4e+0\ (1.2e-1)^+$ | $1.3e-1\ (1.2e-1)^+$ | $4.5e+0\ (1.1e-1)^+$ | $1.9e-1\ (6.5e-2)^+$ | $-6.9e-1\ (3.5e-1)^+$ | $1.0e-1\ (8.7e-2)^+$ | **6/ 0/ 0** |
| SPMO | **3.0e+0 (6.0e-1)** | **-1.8e-1 (5.8e-2)** | **2.9e+0 (4.8e-1)** | **5.8e-2 (6.8e-2)** | **-2.1e+0 (3.3e-1)** | **-1.9e-1 (1.8e-1)** | |

Table 12: The HV of the best solution (in terms of its HV value) obtained by SPMO and the peer methods on the noisy problems with 5 objectives on 30 independent runs. The method with the best mean is highlighted in bold. The symbols "+", "∼" and "−" indicate that the method is statistically worse than, equivalent to and better than our SPMO, respectively.

| Method | DTLZ1 Mean (Std) | DTLZ2 Mean (Std) | Inverted DTLZ1 Mean (Std) | Inverted DTLZ2 Mean (Std) | Convex DTLZ2 Mean (Std) | Scaled DTLZ2 Mean (Std) | Sum up +/∼/− |
|---|---|---|---|---|---|---|---|
| Sobol | $8.6e+12\ (5.2e+11)^+$ | $5.1e-2\ (3.0e-2)^+$ | $5.4e+12\ (1.2e+12)^+$ | $8.2e-4\ (1.5e-3)^+$ | $4.0e-1\ (1.8e-1)^+$ | $3.8e-2\ (1.9e-2)^+$ | **6/ 0/ 0** |
| NParEGO | $9.1e+12\ (2.5e+11)^+$ | $1.0e-1\ (4.9e-2)^+$ | $9.0e+12\ (4.1e+11)^+$ | $5.8e-2\ (1.6e-2)^+$ | $1.2e+0\ (2.0e-1)^+$ | $1.3e-1\ (4.0e-2)^+$ | **6/ 0/ 0** |
| TS-TCH | $8.5e+12\ (5.1e+11)^+$ | $5.0e-2\ (1.4e-2)^+$ | $4.8e+12\ (1.2e+12)^+$ | $8.1e-3\ (5.4e-3)^+$ | $5.3e-1\ (1.7e-1)^+$ | $3.1e-2\ (1.6e-2)^+$ | **6/ 0/ 0** |
| NEHVI | $9.1e+12\ (1.5e+11)^+$ | $2.8e-1\ (7.6e-2)^+$ | $7.4e+12\ (1.2e+12)^+$ | $6.7e-2\ (1.0e-2)^+$ | $1.2e+0\ (2.1e-1)^+$ | $1.2e-2\ (1.0e-2)^+$ | **6/ 0/ 0** |
| JES | $9.0e+12\ (1.3e+11)^+$ | $8.7e-2\ (7.2e-2)^+$ | $6.1e+12\ (3.2e+11)^+$ | $5.8e-2\ (2.0e-2)^+$ | $7.8e-1\ (3.1e-1)^+$ | $1.0e-1\ (5.7e-2)^+$ | **6/ 0/ 0** |
| SPMO | **9.4e+12 (4.4e+11)** | **3.2e-1 (4.9e-2)** | **9.2e+12 (4.2e+11)** | **1.0e-1 (2.7e-2)** | **1.8e+0 (2.2e-1)** | **3.8e-1 (1.6e-1)** | |

Table 13: The HV of all the solutions obtained by the proposed SPMO and the peer methods on the noisy problems with five objectives on 30 independent runs. The method with the best mean is highlighted in bold. The symbols "+", "∼", and "−" indicate that a method is statistically worse than, equivalent to, and better than SPMO, respectively.

| Method | DTLZ1 Mean (Std) | DTLZ2 Mean (Std) | Inverted DTLZ1 Mean (Std) | Inverted DTLZ2 Mean (Std) | Convex DTLZ2 Mean (Std) | Scaled DTLZ2 Mean (Std) | Sum up +/∼/− |
|---|---|---|---|---|---|---|---|
| Sobol | $4.5e+13\ (1.4e+13)^+$ | $2.1e-1\ (8.4e-2)^+$ | $8.2e+12\ (4.7e+12)^+$ | $8.5e-4\ (1.5e-3)^+$ | $9.3e-1\ (5.1e-1)^+$ | $1.6e-1\ (5.7e-2)^+$ | **6/ 0/ 0** |
| NParEGO | $5.8e+13\ (2.3e+13)^\sim$ | $4.3e-1\ (3.9e-1)^+$ | $1.7e+13\ (9.6e+12)^+$ | $5.1e-1\ (1.3e-1)^+$ | $1.1e+1\ (4.8e+0)^+$ | $7.7e-1\ (6.5e-1)^+$ | **5/ 1/ 0** |
| TS-TCH | $5.1e+13\ (1.3e+13)^\sim$ | $1.9e-1\ (8.3e-2)^+$ | $6.9e+12\ (2.6e+12)^+$ | $3.2e-2\ (1.8e-2)^+$ | $2.2e+0\ (9.8e-1)^+$ | $9.0e-2\ (6.5e-2)^+$ | **5/ 1/ 0** |
| NEHVI | $7.2e+13\ (2.1e+13)^\sim$ | **3.1e+0 (8.9e-1)**$^-$ | $1.5e+13\ (1.7e+13)^+$ | $7.7e-1\ (2.0e-1)^+$ | $9.5e+0\ (3.8e+0)^+$ | $1.5e-2\ (1.3e-2)^+$ | **4/ 1/ 1** |
| JES | **7.5e+13 (2.3e+13)**$^\sim$ | $4.3e-1\ (5.0e-1)^+$ | $3.5e+13\ (2.0e+13)^\sim$ | $5.7e-1\ (1.5e-1)^+$ | $4.6e+0\ (2.8e+0)^+$ | $4.3e-1\ (3.3e-1)^+$ | **4/ 2/ 0** |
| SPMO | $6.2e+13\ (2.9e+13)$ | $2.0e+0\ (5.6e-1)$ | **4.4e+13 (3.6e+13)** | **9.6e-1 (3.1e-1)** | **1.4e+1 (4.4e+0)** | **2.9e+0 (1.8e+0)** | |

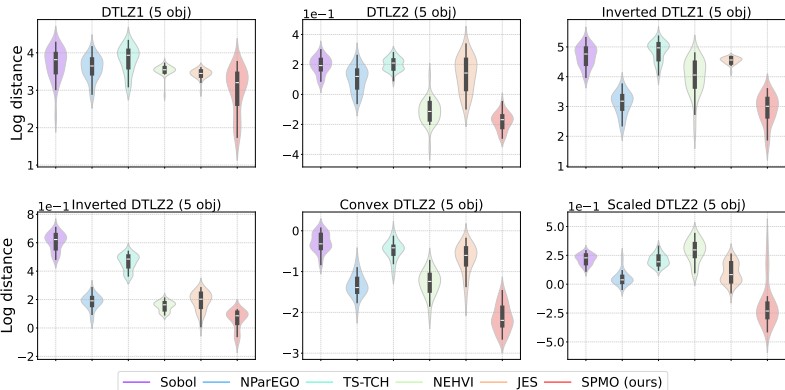

Figure 10: Violin plots of the distance-based metric (log distance) obtained by the six methods on the noisy problems with five objectives. Each violin represents the distribution of the distance-based metric obtained by a method over 30 independent runs.

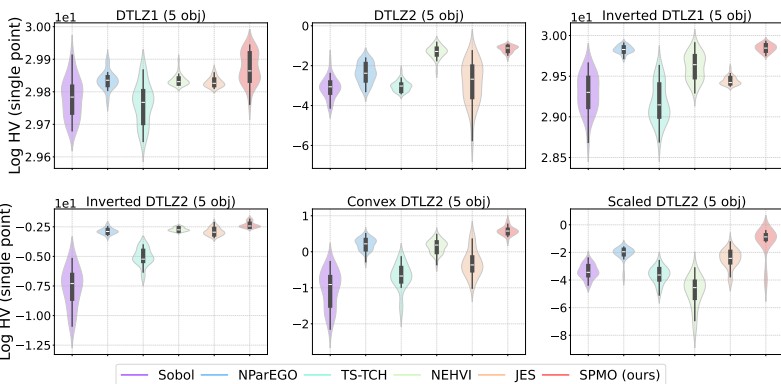

Figure 11: Violin plots of the HV of the best solution (in terms of its HV value) obtained by the six methods on the noisy problems with five objectives. Each violin represents the distribution of maximum HV values obtained by a method over 30 independent runs.

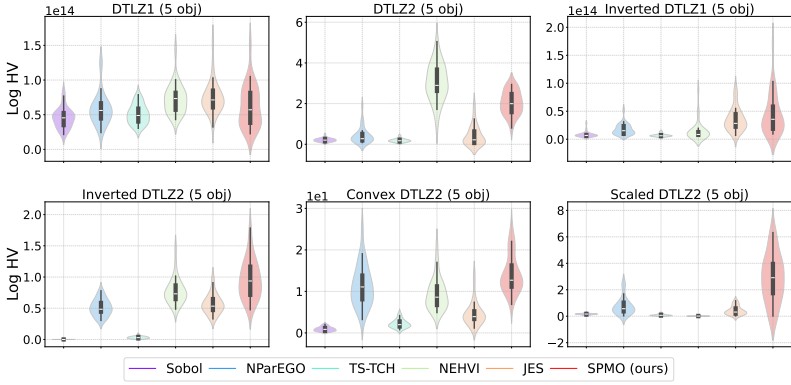

Figure 12: Violin plots of the HV of all evaluated solutions obtained by the six methods on the noisy problems with five objectives. Each violin represents the distribution of maximum HV values obtained by a method over 30 independent runs.

## F.3 Batch Setting

We compare the proposed SPMO with the peer methods in the batch setting where the batch size $q$ is set to 5 (a commonly used value Lin et al. (2022)). Tables 14, 15 and 16 show the distance-based metric (log distance), the HV of the best solution (in terms of its HV value) and the HV of all evaluated solutions obtained by the SPMO and the peer methods, respectively. Figures 13, 14 and 15 present the violin plots, illustrating the distributions of the corresponding results reported in Tables 14, 15 and 16, respectively.

Table 14: Results of the distance-based metric (log distance) obtained by the SPMO and the five peer methods with a batch size $q = 5$ on the problems with 5 objectives on 30 independent runs. The method with the best mean is highlighted in bold. The symbols "+", "∼" and "−" indicate that the method is statistically worse than, equivalent to and better than our SPMO, respectively.

| Method | DTLZ1 Mean (Std) | DTLZ2 Mean (Std) | Inverted DTLZ1 Mean (Std) | Inverted DTLZ2 Mean (Std) | Convex DTLZ2 Mean (Std) | Scaled DTLZ2 Mean (Std) | Sum up +/∼/− |
|---|---|---|---|---|---|---|---|
| Sobol | 3.7e+0 (3.0e−1)$^+$ | 2.4e−1 (4.6e−2)$^+$ | 4.8e+0 (3.1e−1)$^+$ | 6.0e−1 (5.7e−2)$^+$ | −3.1e−1 (2.5e−1)$^+$ | 2.3e−1 (5.1e−2)$^+$ | **6/ 0/ 0** |
| ParEGO | 3.4e+0 (1.9e−1)$^+$ | 6.7e−2 (8.1e−2)$^+$ | 3.2e+0 (5.4e−1)$^+$ | 2.5e−1 (1.3e−2)$^+$ | −1.7e+0 (2.0e−1)$^+$ | 1.3e−1 (8.9e−2)$^+$ | **6/ 0/ 0** |
| TS-TCH | 3.9e+0 (2.2e−1)$^+$ | 2.0e−1 (4.1e−2)$^+$ | 4.8e+0 (3.2e−1)$^+$ | 4.6e−1 (1.2e−2)$^+$ | −7.1e−1 (2.1e−1)$^+$ | 2.6e−1 (3.8e−2)$^+$ | **6/ 0/ 0** |
| EHVI | 3.5e+0 (9.6e−2)$^+$ | 9.3e−3 (3.5e−3)$^+$ | 4.0e+0 (4.4e−1)$^+$ | 2.3e−1 (5.5e−3)$^+$ | −1.4e+0 (2.1e−1)$^+$ | 3.1e−1 (6.0e−2)$^+$ | **6/ 0/ 0** |
| JES | 3.4e+0 (1.3e−1)$^+$ | 1.1e−1 (8.0e−2)$^+$ | 4.5e+0 (1.7e−1)$^+$ | 2.6e−1 (2.0e−2)$^+$ | −1.0e+0 (4.5e−1)$^+$ | 8.8e−2 (9.3e−2)$^+$ | **6/ 0/ 0** |
| SPMO | **3.1e+0 (3.0e−1)** | **9.0e−4 (8.3e−4)** | **2.9e+0 (4.9e−1)** | **2.1e−1 (5.0e−5)** | **−2.1e+0 (7.7e−3)** | **1.7e−4 (1.2e−4)** | |

Table 15: The HV of the best solution (in terms of its HV value) obtained by SPMO and the five peer methods with a batch size $q = 5$ on the problems with 5 objectives on 30 independent runs. The method with the best mean is highlighted in bold. The symbols "+", "∼" and "−" indicate that the method is statistically worse than, equivalent to and better than our SPMO, respectively.

| Method | DTLZ1 Mean (Std) | DTLZ2 Mean (Std) | Inverted DTLZ1 Mean (Std) | Inverted DTLZ2 Mean (Std) | Convex DTLZ2 Mean (Std) | Scaled DTLZ2 Mean (Std) | Sum up +/∼/− |
|---|---|---|---|---|---|---|---|
| Sobol | 8.7e+12 (3.7e+11)$^+$ | 3.5e−2 (1.5e−2)$^+$ | 5.1e+12 (1.1e+12)$^+$ | 8.7e−4 (1.5e−3)$^+$ | 3.8e−1 (1.7e−1)$^+$ | 3.8e−2 (2.1e−2)$^+$ | **6/ 0/ 0** |
| ParEGO | 9.1e+12 (1.8e+11)$^+$ | 1.3e−1 (5.6e−2)$^+$ | 8.8e+12 (7.0e+11)$^+$ | 4.0e−2 (2.9e−3)$^+$ | 1.1e+0 (6.2e−2)$^+$ | 8.7e−2 (5.3e−2)$^+$ | **6/ 0/ 0** |
| TS-TCH | 8.4e+12 (3.8e+11)$^+$ | 3.2e−2 (1.3e−2)$^+$ | 4.9e+12 (1.2e+12)$^+$ | 7.1e−3 (1.1e−3)$^+$ | 6.5e−1 (1.4e−1)$^+$ | 2.8e−2 (1.5e−2)$^+$ | **6/ 0/ 0** |
| EHVI | 9.0e+12 (9.5e+10)$^+$ | **1.9e−1 (7.2e−3)**$^\sim$ | 7.5e+12 (9.7e+11)$^+$ | 4.5e−2 (1.4e−3)$^+$ | 1.0e+0 (9.7e−2)$^+$ | 1.5e−2 (1.4e−2)$^+$ | **5/ 1/ 0** |
| JES | 9.0e+12 (1.2e+11)$^+$ | 1.0e−1 (3.9e−2)$^+$ | 6.2e+12 (5.0e+11)$^+$ | 3.9e−2 (4.5e−3)$^+$ | 8.5e−1 (2.4e−1)$^+$ | 1.1e−1 (5.5e−2)$^+$ | **6/ 0/ 0** |
| SPMO | **9.3e+12 (2.8e+11)** | 1.9e−1 (1.4e−2) | **9.2e+12 (4.8e+11)** | **4.9e−2 (1.2e−5)** | **1.3e+0 (5.0e−3)** | **1.6e−1 (1.6e−2)** | |

Table 16: The HV of all the solutions obtained by the six methods on the problems with five objectives on 30 independent runs. The method with the best mean is highlighted in bold. The symbols "+", "∼", and "−" indicate that a method is statistically worse than, equivalent to, and better than SPMO, respectively.

| Method | DTLZ1 Mean (Std) | DTLZ2 Mean (Std) | Inverted DTLZ1 Mean (Std) | Inverted DTLZ2 Mean (Std) | Convex DTLZ2 Mean (Std) | Scaled DTLZ2 Mean (Std) | Sum up +/∼/− |
|---|---|---|---|---|---|---|---|
| Sobol | 4.4e+13 (1.4e+13)$^\sim$ | 1.4e−1 (5.6e−2)$^+$ | 7.7e+12 (2.9e+12)$^+$ | 9.5e−4 (1.6e−3)$^+$ | 9.5e−1 (4.6e−1)$^+$ | 1.6e−1 (6.6e−2)$^+$ | **5/ 1/ 0** |
| ParEGO | 2.9e+13 (1.2e+13)$^\sim$ | 6.5e−1 (5.3e−1)$^+$ | 1.5e+13 (7.8e+12)$^+$ | 5.0e−1 (9.6e−2)$^\sim$ | 9.6e+0 (4.8e+0)$^+$ | 4.2e−1 (3.9e−1)$^+$ | **4/ 2/ 0** |
| TS-TCH | **4.5e+13 (1.3e+13)**$^\sim$ | 8.9e−2 (5.3e−2)$^+$ | 6.8e+12 (1.8e+12)$^+$ | 5.0e−2 (1.7e−2)$^+$ | 3.2e+0 (1.7e+0)$^+$ | 7.4e−2 (5.2e−2)$^+$ | **5/ 1/ 0** |
| EHVI | 2.9e+13 (1.8e+13)$^\sim$ | **2.1e+0 (5.9e−1)**$^-$ | 1.9e+13 (1.7e+13)$^+$ | **6.1e−1 (1.5e−1)**$^\sim$ | 7.4e+0 (2.2e+0)$^+$ | 2.0e−2 (1.7e−2)$^+$ | **3/ 2/ 1** |
| JES | 3.1e+13 (2.8e+13)$^\sim$ | 6.9e−1 (5.2e−1)$^+$ | **7.4e+13 (9.1e+13)**$^-$ | 4.7e−1 (1.1e−1)$^\sim$ | 5.7e+0 (4.6e+0)$^+$ | 7.5e−1 (6.3e−1)$^+$ | **3/ 2/ 1** |
| SPMO | 3.7e+13 (2.2e+13) | 1.5e+0 (7.0e−1) | 3.7e+13 (3.5e+13) | 5.9e−1 (3.3e−1) | **2.6e+1 (1.7e+1)** | **1.9e+0 (1.6e+0)** | |

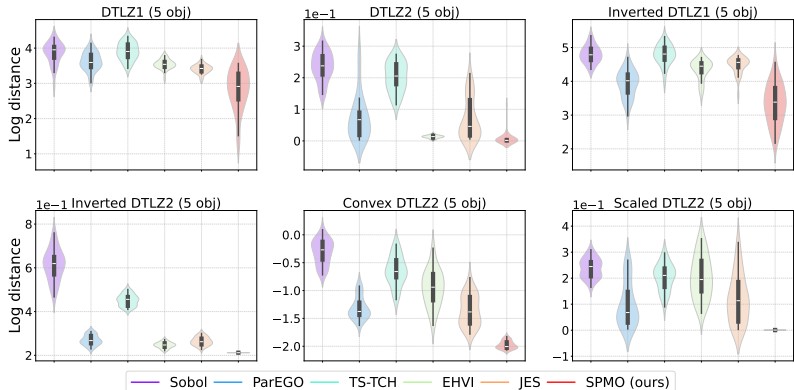

Figure 13: Violin plots of the distance-based metric (log distance) obtained by the proposed SPMO and the peer methods on the problems with five objectives. Each violin represents the distribution of the distance-based metric obtained by a method over 30 independent runs.

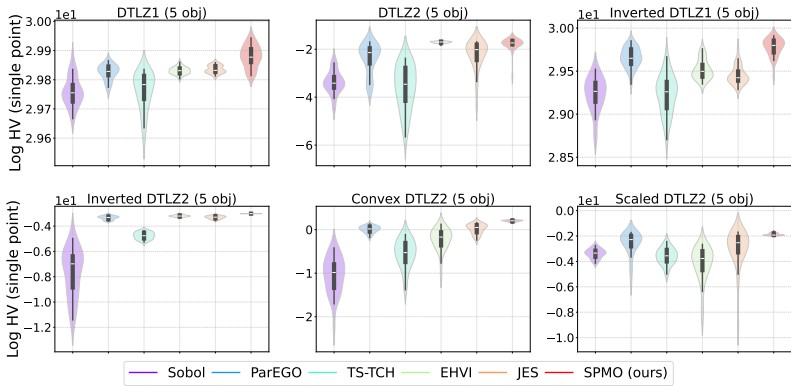

Figure 14: Violin plots of the HV of the best solution (in terms of its HV value) obtained by the six methods with a batch size $q = 5$ on the problems with five objectives. Each violin represents the distribution of maximum HV values obtained by a method over 30 independent runs.

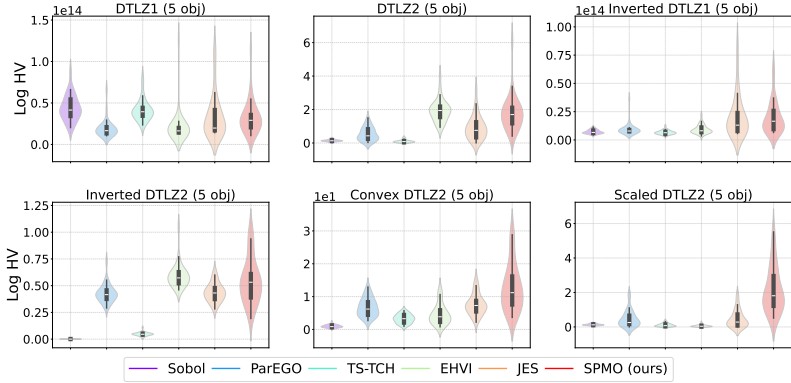

Figure 15: Violin plots of the HV of all evaluated solutions obtained by the six methods on the problems with five objectives. Each violin represents the distribution of maximum HV values obtained by a method over 30 independent runs.

## F.4 SENSITIVITY ANALYSIS

In this section, we conduct a sensitivity analysis to assess the effect of different utopian points. We consider three different settings. The first one is slightly better than the ideal point, i.e., with a difference of 0.01, the second is fairly better than the ideal point (i.e. 0.1), and the last one is significantly better than the ideal point (i.e. 1.0). Tables 17, 18 and 19 show the distance-based metric (log distance), the HV of the best solution (in terms of its HV value), and the HV of all evaluated solutions obtained by the SPMO with four different utopian points, respectively. Figures 16, 17 and 18 present the violin plots, illustrating the distributions of the corresponding results reported in Tables 17, 18 and 19, respectively.

Table 17: Results of the distance-based metric (log distance) obtained by the SPMO with four different utopian points on the problems with five objectives on 30 independent runs. The method with the best mean is highlighted in bold. The symbols "+", "∼" and "−" indicate that the method is statistically worse than, equivalent to and better than SPMO (current), respectively.

| Method | DTLZ1 Mean (Std) | DTLZ2 Mean (Std) | Inverted DTLZ1 Mean (Std) | Inverted DTLZ2 Mean (Std) | Convex DTLZ2 Mean (Std) | Scaled DTLZ2 Mean (Std) | Sum up +/∼/− |
|---|---|---|---|---|---|---|---|
| SPMO_0.01 | 3.1e+0 (4.9e–1)$^\sim$ | 6.2e–4 (1.4e–3)$^-$ | **2.8e+0 (4.2e–1)**$^\sim$ | 2.1e–1 (3.6e–5)$^\sim$ | -2.1e+0 (1.6e–2)$^\sim$ | 6.4e–5 (3.9e–5)$^-$ | 0/ 4/ 2 |
| SPMO_0.1 | 2.9e+0 (3.6e–1)$^-$ | **3.0e–4 (2.3e–4)**$^-$ | 3.0e+0 (5.3e–1)$^\sim$ | 2.1e–1 (4.6e–5)$^\sim$ | -2.1e+0 (2.6e–2)$^\sim$ | **5.3e–5 (2.4e–5)**$^-$ | 0/ 3/ 3 |
| SPMO_1.0 | **2.8e+0 (6.1e–1)**$^\sim$ | 3.3e–4 (2.0e–4)$^-$ | 2.9e+0 (5.0e–1)$^\sim$ | **2.1e–1 (3.4e–5)**$^\sim$ | -2.1e+0 (1.6e–2)$^\sim$ | 5.7e–5 (2.4e–5)$^-$ | 0/ 4/ 2 |
| SPMO | 3.1e+0 (3.0e–1) | 9.0e–4 (8.3e–4) | 2.9e+0 (4.9e–1) | 2.1e–1 (5.0e–5) | **-2.1e+0 (7.7e–3)** | 1.7e–4 (1.2e–4) | |

Table 18: The HV of the best solution (in terms of its HV value) obtained by the proposed SPMO with four different utopian points on the problems with five objectives on 30 independent runs. The method exhibiting the best mean is highlighted in bold. The symbols "+", "∼", and "−" denote that a method is statistically worse than, equivalent to, or better than SPMO (current), respectively.

| Method | DTLZ1 Mean (Std) | DTLZ2 Mean (Std) | Inverted DTLZ1 Mean (Std) | Inverted DTLZ2 Mean (Std) | Convex DTLZ2 Mean (Std) | Scaled DTLZ2 Mean (Std) | Sum up +/∼/− |
|---|---|---|---|---|---|---|---|
| SPMO_0.01 | 9.4e+12 (3.7e+11)$^\sim$ | **1.9e–1 (1.0e–2)**$^\sim$ | **9.3e+12 (3.8e+11)**$^\sim$ | 4.9e–2 (9.0e–6)$^\sim$ | **1.3e+0 (7.8e–3)**$^\sim$ | 1.6e–1 (1.4e–2)$^\sim$ | 0/ 6/ 0 |
| SPMO_0.1 | 9.5e+12 (2.7e+11)$^-$ | 1.9e–1 (1.6e–2)$^\sim$ | 9.0e+12 (5.6e+11)$^\sim$ | 4.9e–2 (1.1e–5)$^\sim$ | 1.2e+0 (1.1e–2)$^\sim$ | 1.6e–1 (1.0e–2)$^\sim$ | 0/ 5/ 1 |
| SPMO_1.0 | **9.5e+12 (3.8e+11)**$^-$ | 1.9e–1 (1.3e–2)$^\sim$ | 9.1e+12 (4.2e+11)$^\sim$ | **4.9e–2 (8.5e–6)**$^\sim$ | 1.3e+0 (7.8e–3)$^\sim$ | 1.6e–1 (1.4e–2)$^\sim$ | 0/ 5/ 1 |
| SPMO | 9.3e+12 (2.8e+11) | 1.9e–1 (1.4e–2) | 9.2e+12 (4.8e+11) | 4.9e–2 (1.2e–5) | 1.3e+0 (5.0e–3) | **1.6e–1 (1.6e–2)** | |

Table 19: The HV of all the solutions obtained by the SPMO with four different utopian points on the problems with five objectives on 30 independent runs. The method with the best mean is highlighted in bold. The symbols "+", "∼", and "−" indicate that a method is statistically worse than, equivalent to, and better than SPMO (current), respectively.

| Method | DTLZ1 Mean (Std) | DTLZ2 Mean (Std) | Inverted DTLZ1 Mean (Std) | Inverted DTLZ2 Mean (Std) | Convex DTLZ2 Mean (Std) | Scaled DTLZ2 Mean (Std) | Sum up +/∼/− |
|---|---|---|---|---|---|---|---|
| SPMO_0.01 | 3.1e+13 (1.7e+13)$^\sim$ | 2.7e+0 (1.2e+0)$^-$ | 3.5e+13 (3.7e+13)$^\sim$ | 6.5e–1 (7.0e–1)$^\sim$ | 2.5e+1 (1.8e+1)$^\sim$ | **2.2e+0 (1.5e+0)**$^\sim$ | 0/ 5/ 1 |
| SPMO_0.1 | 3.5e+13 (1.5e+13)$^\sim$ | 2.7e+0 (1.2e+0)$^-$ | 2.6e+13 (1.7e+13)$^\sim$ | 6.1e–1 (4.5e–1)$^\sim$ | 2.7e+1 (1.7e+1)$^\sim$ | 1.8e+0 (1.0e+0)$^\sim$ | 0/ 5/ 1 |
| SPMO_1.0 | 3.3e+13 (1.4e+13)$^\sim$ | **3.0e+0 (1.6e+0)**$^-$ | 3.2e+13 (1.9e+13)$^\sim$ | **7.1e–1 (6.3e–1)**$^\sim$ | **2.9e+1 (2.2e+1)**$^\sim$ | 2.1e+0 (1.5e+0)$^\sim$ | 0/ 5/ 1 |
| SPMO | **3.7e+13 (2.2e+13)** | 1.5e+0 (7.0e–1) | **3.7e+13 (3.5e+13)** | 5.9e–1 (3.3e–1) | 2.6e+1 (1.7e+1) | 1.9e+0 (1.6e+0) | |

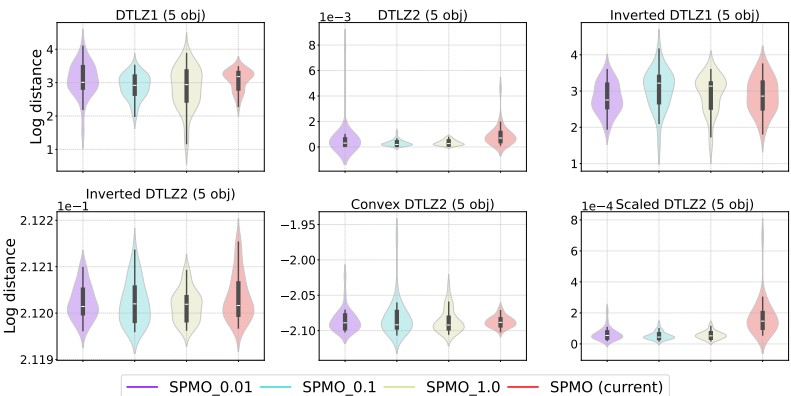

Figure 16: Violin plots of the distance-based metric (log distance) obtained by the four methods on the problems with five objectives. Each violin represents the distribution of the distance-based metric obtained by a method over 30 independent runs.

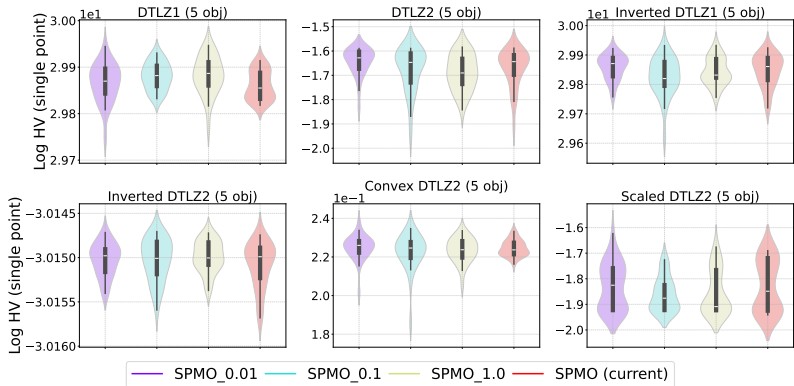

Figure 17: Violin plots of the HV of the best solution (in terms of its HV value) obtained by the four methods on the problems with five objectives. Each violin represents the distribution of maximum HV values obtained by a method over 30 independent runs.

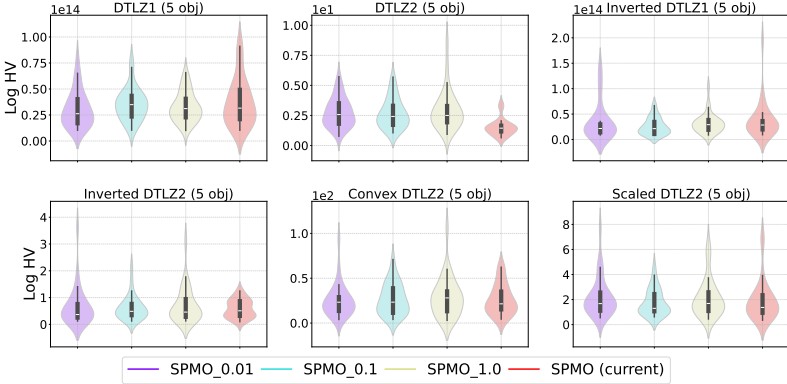

Figure 18: Violin plots of the HV of all evaluated solutions obtained by the four methods the problems with five objectives. Each violin represents the distribution of maximum HV values obtained by a method over 30 independent runs.

## F.5 COMPARISON OF SINGLE-POINT METRICS WITHIN SPMO

In the proposed SPMO framework, we employ a distance metric (i.e., the distance of a solution to the utopian point). However, different metrics can be adopted provided that it can reflect the quality of a solution in achieving a good trade-off between objectives, such as the weighted sum and Tchebycheff scalarisation with the same weights $(\frac{1}{m}, \dots, \frac{1}{m})$, where $m$ denotes the number of objectives. Here, we compare these three variants of SPMO, i.e., $\text{SPMO}_{dist}$, $\text{SPMO}_{Tch}$, and $\text{SPMO}_{ws}$. Tables 20, 21 and 22 show the distance-based metric (log distance), the HV of the best solution (in terms of its HV value) and the HV of all evaluated solutions obtained by the SPMO with three different single-point metrics, respectively. Figures 19, 20 and 21 present the violin plots, illustrating the distributions of the corresponding results reported in Tables 20, 21 and 22, respectively.

Table 20: Results of the distance-based metric (log distance) obtained by the SPMO using three different single-point metrics on the problems with five objectives on 30 independent runs. The method with the best mean is highlighted in bold. The symbols "+", "∼" and "−" indicate that the method is statistically worse than, equivalent to and better than our SPMO (i.e., $\text{SPMO}_{dist}$), respectively.

| Method | DTLZ1 Mean (Std) | DTLZ2 Mean (Std) | Inverted DTLZ1 Mean (Std) | Inverted DTLZ2 Mean (Std) | Convex DTLZ2 Mean (Std) | Scaled DTLZ2 Mean (Std) | Sum up +/∼/− |
|---|---|---|---|---|---|---|---|
| $\text{SPMO}_{Tch}$ | 3.5e+0 (3.2e–1)$^+$ | **3.5e–5 (4.3e–5)**$^-$ | 2.9e+0 (6.5e–1)$^\sim$ | 3.1e–1 (3.3e–2)$^+$ | -1.8e+0 (1.7e–1)$^+$ | **2.9e–5 (1.9e–5)**$^-$ | 3/ 1/ 2 |
| $\text{SPMO}_{ws}$ | 3.6e+0 (1.3e–1)$^+$ | 2.1e–4 (1.5e–4)$^-$ | 3.1e+0 (5.4e–1)$^\sim$ | 2.2e–1 (2.7e–3)$^+$ | -1.2e+0 (2.7e–1)$^+$ | 3.2e–4 (3.4e–4)$^+$ | 4/ 1/ 1 |
| **SPMO** | **3.1e+0 (3.0e–1)** | 9.0e–4 (8.3e–4) | **2.9e+0 (4.9e–1)** | **2.1e–1 (5.0e–5)** | **-2.1e+0 (7.7e–3)** | 1.7e–4 (1.2e–4) | |

Table 21: The HV of the best solution (in terms of its HV value) obtained by the proposed SPMO using three different single-point metrics on the problems with five objectives on 30 independent runs. The method exhibiting the best mean is highlighted in bold. The symbols "+", "∼", and "−" denote that a method is statistically worse than, equivalent to, or better than SPMO (i.e., $\text{SPMO}_{dist}$), respectively.

| Method | DTLZ1 Mean (Std) | DTLZ2 Mean (Std) | Inverted DTLZ1 Mean (Std) | Inverted DTLZ2 Mean (Std) | Convex DTLZ2 Mean (Std) | Scaled DTLZ2 Mean (Std) | Sum up +/∼/− |
|---|---|---|---|---|---|---|---|
| $\text{SPMO}_{Tch}$ | 9.1e+12 (3.0e+11)$^+$ | 1.6e–1 (1.9e–2)$^+$ | 9.1e+12 (5.7e+11)$^\sim$ | 2.7e–2 (5.9e–3)$^+$ | 1.2e+0 (4.2e–2)$^+$ | 1.6e–1 (2.2e–2)$^\sim$ | 4/ 2/ 0 |
| $\text{SPMO}_{ws}$ | 9.1e+12 (1.3e+11)$^+$ | 1.7e–1 (1.4e–2)$^+$ | 9.0e+12 (6.5e+11)$^\sim$ | 4.8e–2 (6.7e–4)$^+$ | 9.7e–1 (1.2e–1)$^+$ | **1.7e–1 (1.5e–2)**$^\sim$ | 4/ 2/ 0 |
| **SPMO** | **9.3e+12 (2.8e+11)** | **1.9e–1 (1.4e–2)** | **9.2e+12 (4.8e+11)** | **4.9e–2 (1.2e–5)** | **1.3e+0 (5.0e–3)** | 1.6e–1 (1.6e–2) | |

Table 22: The HV of all the solutions obtained by the SPMO using three different single-point metrics on the problems with five objectives on 30 independent runs. The method with the best mean is highlighted in bold. The symbols "+", "∼", and "−" indicate that a method is statistically worse than, equivalent to, and better than SPMO (i.e., $\text{SPMO}_{dist}$), respectively.

| Method | DTLZ1 Mean (Std) | DTLZ2 Mean (Std) | Inverted DTLZ1 Mean (Std) | Inverted DTLZ2 Mean (Std) | Convex DTLZ2 Mean (Std) | Scaled DTLZ2 Mean (Std) | Sum up +/∼/− |
|---|---|---|---|---|---|---|---|
| $\text{SPMO}_{Tch}$ | 1.5e+13 (6.2e+12)$^+$ | **4.9e+0 (7.6e+0)**$^-$ | 2.1e+13 (1.5e+13)$^+$ | **2.1e+0 (2.6e+0)**$^-$ | 1.3e+1 (7.4e+0)$^+$ | **3.3e+0 (3.4e+0)**$^-$ | 3/ 0/ 3 |
| $\text{SPMO}_{ws}$ | 1.3e+13 (3.1e+12)$^+$ | 1.6e+0 (7.5e–1)$^\sim$ | 2.4e+13 (1.8e+13)$^+$ | 7.4e–1 (7.5e–1)$^\sim$ | 6.4e+0 (3.8e+0)$^+$ | 1.2e+0 (3.5e–1)$^\sim$ | 3/ 3/ 0 |
| **SPMO** | **3.7e+13 (2.2e+13)** | 1.5e+0 (7.0e–1) | **3.7e+13 (3.5e+13)** | 5.9e–1 (3.3e–1) | **2.6e+1 (1.7e+1)** | 1.9e+0 (1.6e+0) | |

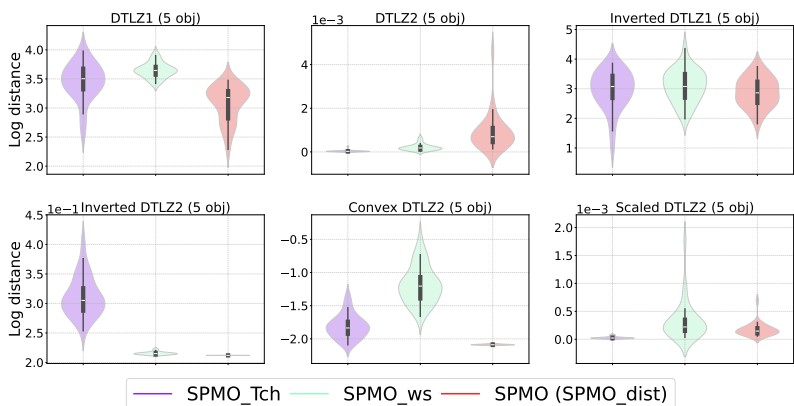

Figure 19: Violin plots of the distance-based metric (log distance) obtained by the SPMO using three different single-point metrics on the problems with five objectives. Each violin represents the distribution of the distance-based metric obtained by a method over 30 independent runs.

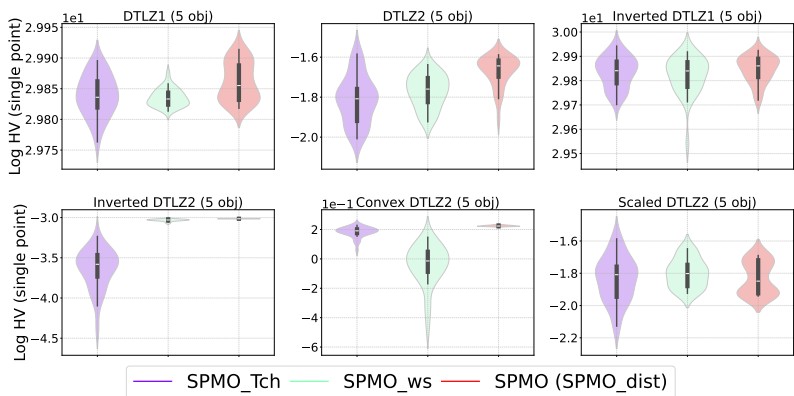

Figure 20: Violin plots of the HV of the best solution (in terms of its HV value) obtained by the SPMO using three different single-point metrics on the problems with 5 objectives. Each violin represents the distribution of the single-point HV obtained by a method over 30 independent runs.

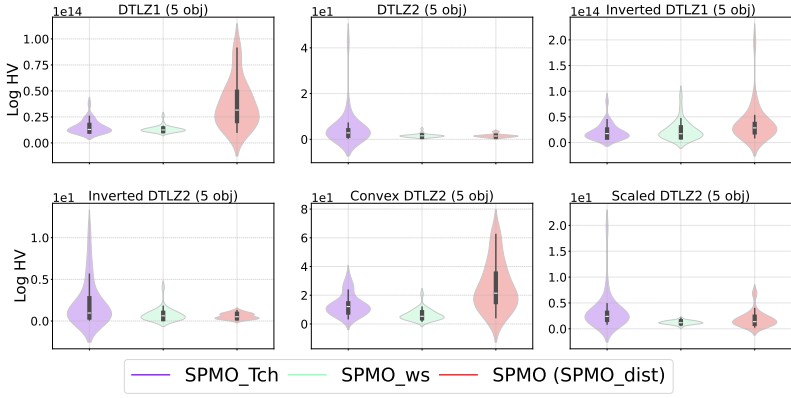

Figure 21: Violin plots of the HV of all evaluated solutions obtained by the SPMO using three different single-point metrics on the problems with five objectives. Each violin represents the distribution of maximum HV values obtained by a method over 30 independent runs.

## F.6 ACQUISITION WALL TIME

Table 23 presents the mean acquisition optimisation wall time of the eight methods. As shown, when the number of objectives is 3 or 5, the time of all the methods is acceptable with a maximum of 98 seconds. As the number of objectives increases to 10, hypervolume-based methods (i.e., EHVI and NEHVI) become very expensive (taking about half an hour and more than 3 hours, respectively). The proposed SPMO method shows high computational efficiency, achieving the lowest time requirement in four out of the six instances.

Table 23: Mean acquisition optimisation wall time in seconds based on the $2(d+1)$ initial Sobol samples on DTLZ1 problems with $m = 3, 5, 10$ objectives, where $d = m + 4$, over 30 runs. Experiments are conducted using a CPU (Intel Xeon CPU Platinum 8360Y @ 2.40 GHz) and a GPU (NVIDIA A100). Note that N/A means the wall time of NEHVI on DTLZ1 with 10 objectives exceeds 3 hours.

| Device\Method | ParEGO | NParEGO | TS-TCH | EHVI | NEHVI | C-EHVI | JES | SPMO (ours) |
|---|---|---|---|---|---|---|---|---|
| **CPU (3 obj)** | 3.46 | 3.49 | 12.29 | 4.23 | 6.15 | 10.23 | 9.24 | 2.40 |
| **GPU (3 obj)** | 5.19 | 3.94 | 14.74 | 3.05 | 5.73 | 9.48 | 14.02 | 2.24 |
| **CPU (5 obj)** | 2.59 | 2.14 | 28.32 | 36.03 | 97.98 | 29.89 | 25.16 | 2.58 |
| **GPU (5 obj)** | 9.02 | 10.33 | 22.29 | 10.32 | 63.90 | 23.89 | 25.80 | 5.31 |
| **CPU (10 obj)** | 8.76 | 5.97 | 87.44 | 1134.35 | N/A | 77.94 | 388.37 | 6.95 |
| **GPU (10 obj)** | 41.33 | 30.30 | 64.78 | 2426.04 | N/A | 86.45 | 676.31 | 24.05 |

## G APPLICABILITY OF THE PROPOSED SPMO

In conventional multi-objective optimisation, algorithms are designed to approximate the entire Pareto front, so that a decision-maker can later select a preferred solution based on their own preferences. This is the ideal situation, as a well-represented Pareto front provides the most comprehensive view of the possible trade-offs. However, under tight evaluation budgets - especially when many objectives are involved - it is often unrealistic, if not impossible, to obtain a good approximation of the Pareto front. In such settings, the decision-maker may benefit more from receiving a well-balanced solution that is close to the Pareto front, rather than a well-distributed solution set far from the front. The proposed SPMO framework is designed for this purpose: instead of spreading search effort across the whole front, it directs the optimisation towards a well-balanced solution. With a focused search effort, the obtained solution is often closer to the front, and thus has a higher likelihood of being selected by the decision-maker.

It is worth pointing out that if the optimisation problem under consideration is extremely simple (e.g., smooth, unimodal landscape with a very limited search space), on which finding a good representation of the entire Pareto front is possible under tight budgets, then our approach may not be desirable. Moreover, exploring the Pareto front can help decision-makers better understand the optimisation problem and facilitate the elicitation or refinement of their preferences. In such cases, our framework is not applicable.

## H EXTENSIONS

The preceding discussion has addressed multi-objective optimisation problems in which evaluations are performed either sequentially or in batches, under both noiseless and noisy scenarios. However, not all multi-objective settings conform to these scenarios. To accommodate a broader class of problems, we propose several extensions that enable the methodology to handle more optimisation scenarios.

**High-Dimensional Bayesian Optimisation (HDBO).** High-dimensional black-box optimisation problems are highly challenging and frequently encountered in a wide range of applications. The dimensionality may range from tens to a billion González-Duque et al. (2024); Papenmeier et al. (2023); Santoni et al. (2024); Wang et al. (2016). To tackle such optimisation problems, various HDBO methods have been proposed Binois & Wycoff (2022); Chen et al. (2024); Nayebi et al. (2019);

Wang et al. (2018; 2016); Xu et al. (2025). They can loosely be categorised into four classes Santoni et al. (2024), i.e., variable selection Eriksson & Jankowiak (2021), additive models Delbridge et al. (2020); Han et al. (2021); Wang et al. (2018); Ziomek & Ammar (2023), embeddings Antonov et al. (2022); Letham et al. (2020); Raponi et al. (2020), and trust regions Daulton et al. (2022b); Diouane et al. (2023); Eriksson et al. (2019). However, recent studies show that standard Gaussian processes without the above techniques can perform well in high-dimensional spaces Hvarfner et al. (2024); Papenmeier et al. (2025); Xu et al. (2025) and suggest that the main issue in high-dimensional BO is the gradient vanishing. Our work can be naturally extended to the high-dimensional setting by mitigating the gradient vanishing issue Papenmeier et al. (2025); Xu et al. (2025).

**Multi-Fidelity Bayesian Optimisation (MFBO).** In many real-world optimisation scenarios, the evaluation is often available at multiple fidelity levels, where increasing fidelity typically leads to improved accuracy at the expense of higher computational cost. Many MFBO methods have been proposed to tackle such optimisation problems Belakaria et al. (2020a); Kandasamy et al. (2017); Li et al. (2020); Moss et al. (2021); Song et al. (2019); Takeno et al. (2020); Wu et al. (2020); Zhang et al. (2017). Our proposed SPMO can be potentially extended to the multi-fidelity setting by integrating prior techniques, e.g., building multiple surrogate models of different levels of fidelity.

# I LLM USAGE

We used large language models (LLMs) solely for language polishing.

