# OpenReview forum: "Do We Really Need to Approach the Entire Pareto Front in Many-Objective Bayesian Optimisation?"
_ICLR.cc/2026/Conference — Submitted to ICLR 2026_

### Official Review · Reviewer_Jbtm · 2025-10-27

**Soundness:** 3
**Presentation:** 3
**Contribution:** 2
**Rating:** 4
**Confidence:** 4

**Summary:**

The paper introduces a single point-based multi-objective search framework, SPMO, with the goal of improving the trade-off between solutions along the Pareto frontier by optimizing for a single point objective. The idea behind this is that emphasizing a single solution of high quality provides a better solution than exploring the entire Pareto front. The methodology uses a new acquisition function definition that emphasizes a single direction, given an oracle value, allowing BO to approach a point that might be closer to the user's ultimate preference. The authors define the method for both noisy and noiseless functions. The method shows improved results compared to baselines on the author’s single point metric and some improvement on hypervolume calculations across the Pareto front. This shows promise for an improved methodology for many-objective optimization.

**Strengths:**

- The paper is clearly written and provides an insightful overview of the intuition behind the method and the methodology used. The definition of the acquisition is clear and the methodology used for evaluation is well done.
- The paper shows SPMO outperforms baselines over various benchmarks across noisy and noiseless tests. In particular, the distance-based metric results show a clear superiority of SPMO over baselines. The methodology is defined clearly and tested on both the noisy and noiseless domains, which expands its usefulness.
- The runtime of SPMO compared to EHVI is superior and represents an important improvement over this competitive method. In particular, the method is much more efficient for many-objective (ex: 10 objectives) scenarios. The authors demonstrate that the method is flexible to extend to a batch mode as well.

**Weaknesses:**

- The method utilizes a known oracle value in its methodology, which may limit the applicability of the method to many real-world circumstances where little information may be known. The authors show an ablation over various scales applied to this oracle, but still rely on this knowledge, which makes the comparison to other methods more difficult to infer.
- While the method does not optimize for hypervolume of all solutions, EHVI is competitive/beats SPMO on this benchmark. For users who prefer this metric, they may not prefer SPMO. Further justifying the distance-based metric or best hypervolume as better metrics would be beneficial.
- The results obtained on DTLZ 3-7 are less convincing, particularly for the hypervolume of all solutions. This is concerning that the results from the main paper are not consistent with these tests. While good benchmarks for these methods are difficult to find, further explanation or exploration of diverse test benchmarks would improve the paper.

**Questions:**

- How does the method perform without any knowledge of a known oracle? For example, inferring a possible oracle value based on previously evaluated points? Does an error in oracle setting (ex: a much lower oracle value for a specific parameter than is realistic) degrade results?
- Have you identified any problems/scenarios where SPMO fails compared to existing methods?
Can you justify why the results on DTLZ 3-7 are worse, particularly for the hypervolume of all solutions. Why do these differ from the results presented in the main paper and why is this not discussed further?
- How can a user’s preferences be incorporated into the method? Many users may select from the Pareto front based on differing desired objectives, but this may limit this possibility.

---

> ### Author Response · Authors · 2025-11-27
> **Response [1/2]**
>
> **W1\&Q1: The method needs a known oracle value (i.e., the ideal point of multi-objective problems). Does an error in oracle setting (e.g., a much lower oracle value than realistic) degrade results?**
>
> Yes, the proposed method needs to set a reference point (the utopian point) for a given problem. Our sensitivity analysis (Section *Sensitivity Analysis* and Appendix F.4) shows that the method works reliably on a wide range of utopian point settings: from the ideal point to a very optimistic estimate (even commensurable to the entire objective space range for functions like DTLZ2). An intuitive explanation for this robustness is that the ranking of solutions based on their distances to the utopian point typically does not change when the utopian point is shifted from the ideal point to a more optimistic position. For example, consider two bi-objective solutions, $(0.4, 0.4)$ and $(0, 1)$. The first can be viewed as the better trade-off. When the utopian point is $(0, 0)$, the first solution has a smaller distance $(0.57 < 1)$. When the utopian point is shifted to $(-1, -1)$, it still has a smaller distance $(1.98 < 2.24)$. Even when a more extreme utopian point, such as $(-10, -10)$, the ranking remains unchanged $(14.71 < 14.87)$.
>
> In real-world settings, the ideal point is generally unknown before optimisation begins. However, in many cases, the ideal value of an objective can be estimated optimistically using domain knowledge or trivial bounds. For example, energy loss, fuel consumption, delay time, error, number of defects, or cost can often be idealised as zero; material usage is trivially known to be at least the minimum required volume; cycle time must not be less than the machine’s processing time; and so on.
>
> In addition, it is worth noting that the main contribution of this paper is to propose a novel framework for many-objective BO that focuses on identifying a single high-quality solution, rather than approximating the entire Pareto front as is commonly done in the literature. Within the framework, any metric that captures the quality of a solution in terms of achieving a good trade-off among objectives can be adopted, not only necessarily the one that uses a utopian point.
>
>
> **W2\&Q2, part1: EHVI is competitive/beats SPMO with respect to hypervolume for all solutions. Have you identified any problems/scenarios where SPMO fails compared to existing methods?**
>
> On the hypervolume metric for all solutions, EHVI is better on four problems whereas the proposed SPMO is better on three problems (Table 3). This is expected since the goal of SPMO is to identify one high-quality solution, rather than to find a good representation of the entire Pareto front which is equivalent to maximising the problem’s hypervolume [Zitzler et al., 2003]. Notably, a solution set with a higher hypervolume does not necessarily contain solutions preferred by the decision-maker (see the counterexample in Figure 1 of the paper), since in the end the decision-maker will not choose multiple solutions but rather only one solution to deploy.
>
> In fact, for the other two metrics that evaluate the quality of a single point, the proposed SPMO significantly outperforms EHVI. It achieves a significantly better value on 13 out of the 14 cases, with one case being statistically equivalent (Tables 1 and 2).
>
> Lastly, we would like to note that, ideally, the decision-maker would benefit most from having access to a well-represented Pareto front, enabling them to select a preferred solution based on their specific preferences. This is feasible when the problem is relatively simple and optimisation algorithms can approximate the Pareto front well, which is the situation in which the proposed SPMO would not be suitable compared with existing methods. However, in many-objective optimisation (especially under a limited evaluation budget), obtaining a good approximation of the Pareto front is highly unrealistic, if not impossible, as can be seen in our experiments.
>
> We will incorporate the above discussion into the revision to make this issue clearer – thank you for this helpful comment.
>
>
> **W3\&Q2, part2: The results obtained on DTLZ3-7 (in Appendix) are less convincing, particularly for the hypervolume of all solutions.**
>
> As explained in the reply to the last comment, our method SPMO is expected to perform poorly on the hypervolume for all solutions. In fact, on this metric, SPMO can still obtain the best hypervolume in 2 out of the 5 problems on DTLZ3--7 (Table 10). For the other metrics which measure the quality of a single solution, SPMO clearly outperforms its competitors, obtaining significantly better results on 36 out of 50 comparisons (Tables 8 and 9).

---

> > ### Author Response · Authors · 2025-11-27
> > **Response [2/2]**
> >
> > **Q3: How can a user’s preferences be incorporated into the proposed method?**
> >
> > Since the proposed idea involves a conversion from multiple objectives of a solution to a single scalar metric that reflects its quality, user preferences can be naturally incorporated into this process (e.g., through the use of weights). This applies not only to scalarisation-based metrics (e.g., the weighted sum and Tchebycheff), but also to distance-based and hypervolume-based metrics [Zitzler et al., 2007]. For example, with the distance-based metric used in this work, if a user prioritises one objective over the others, they can assign it a larger weight, resulting in a weighted distance. We will discuss this point in the revised paper.
> >
> >
> >
> > **References**
> >
> > [Zitzler et al., 2003] Zitzler, E., Thiele, L., Laumanns, M., Fonseca, C. M., and Da Fonseca, V. G., 2003. Performance assessment of multiobjective optimizers: An analysis and review. IEEE Transactions on Evolutionary Computation, 2003.
> >
> > [Zitzler et al., 2007] Zitzler, E., Brockhoff, D., and Thiele, L., 2007. The hypervolume indicator revisited: On the design of Pareto-compliant indicators via weighted integration. EMO, 2007.

---

### Official Review · Reviewer_T62o · 2025-10-31

**Soundness:** 4
**Presentation:** 3
**Contribution:** 2
**Rating:** 4
**Confidence:** 4

**Summary:**

The paper considers many-objective Bayesian optimization. Rather than aiming to find a representative subset of Pareto optimal solutions distributed over the whole Pareto front, which can become computationally intractable in BO (where the objective is expensive), the algorithm presents instead finds a single solution that lies closest (in the Euclidean distance sense) to a utopian point (presumably) suggested by the user, with the underlying motivation (assumption) that the user wants a single solution conforming to a pre-defined preference/bias.

**Strengths:**

1. The paper is easy to follow, covering relevant points.
2. The paper appears mathematically sound, and the convergence proof appears reasonable (though I have only skimmed the proof).
3. The experimental validation is thorough.

**Weaknesses:**

While the proposed solution in the paper is mathematically elegant, I do wonder about how user-friendly it would be in practice.  For example, suppose $f_1, f_2 \in [-1,0]$ and the user selects a utopian point $z^\star = (-1.1,-1,1)$. To me, this would indicate that the user wants (at least roughly speaking) an equal balance between the two objectives (perhaps I am misinterpreting here, in which case please correct me), but if if the Pareto front is concave this may well not happen.

For example, suppose for arguments sake that the Pareto front in our 2-d example is well approximated by the lines $f_1 = -0.5$, $-1 \leq f_2 \leq -0.5$ and $-1 \leq f_1 \leq -0.5$, $f_2 = -0.5$. In this case the closest (Euclidean) points on the front to the utopian point will be $f_1 = -1, f_2 = -0.5$ and $f_1 = -0.5, f_2 = -1$, which do not (at least as I understand it) reflect the desires of the user at all (who would presumably prefer the Pareto-optimal solution $f_1 = f_2 = 0.5$). And even if we assume a convex Pareto front there may still be problems - for example suppose the Pareto front is well approximated by the line $f_1 = -0.5$, $-1 \leq f_2 \leq 0$ - again, the closest point to the utopian point is $f_1 = -0.5$, $f_2 = -1$, which again does not appear to reflect the user's wishes.

Nor do I see an easy way to fix this problem by, for example, considering a non-Euclidean distance, as the choice of a "correct" norm (perhaps the $\epsilon$-norm (pseudo-norm) in the concave example above) is strongly dependent on the (a-priori unknown, and presumably unknowable in a reasonable time in the many-objective case if experiments are expensive) shape of the Pareto front.

Perhaps the solution is a fast way to find a solution, but in this case I must wonder if it would be better to use a scalarization scheme, which should be faster almost by definition.

**Questions:**

See weaknesses - basically, can you provide a more thorough discussion on (a) how the utopian point is selected and (b) how realistic it is to assume a connection between the preferences (presumably) reflected by the utopian point and under what conditions these preferences may be reflected in the solution found.

---

> ### Author Response · Authors · 2025-11-27
> **Response**
>
> **W1\&Q1(b): The proposed method is dependent on the shape of the Pareto front. How user-friendly would it be in practice?**
>
> This is an insightful comment. We would like to take this opportunity to explain why the proposed method can address this issue well, following the examples provided by the reviewer.
>
> Under minimisation, when the Pareto front can be well approximated by the two lines $f\_1 = -0.5, -1 \leq f\_2 \leq -0.5$ and $f\_2 = -0.5, -1 \leq f\_1 \leq -0.5$, it means the front is highly concave. In other words, the middle region of the front is very close to the point $(-0.5, -0.5)$. Suppose the point $(-0.51, -0.51)$ is one such solution on the Pareto front. Let us say the decision-maker desires a good trade-off between the objectives (e.g., both objectives are considered equally important). When comparing the two solutions $(-0.51, -0.51)$ and $(-1, -0.5)$, they would very likely prefer $(-1, -0.5)$ as it is significantly better on the first objective and only slightly worse on the second. In other words, they are unlikely to hesitate to trade a 0.01 loss on one objective for a 0.49 gain on the other.
>
> Similarly, for the case where the Pareto front can be well approximated by the line $f\_1 = -0.5, -1 \le f\_2 \le 0$, the decision-maker would be highly likely to choose solutions near $(-0.5, -1)$ rather than those near $(-0.5, -0.5)$ on the Pareto front. In fact, the aim of the proposed method is to choose, among various nondominated solutions, a good trade-off, knee-like point. Note that a good trade-off does not necessarily lie in the middle of the Pareto front – it means that sacrificing a little on one objective can lead to a substantial gain on the others.
>
> We will incorporate the above discussion into the revision to clarify this issue – thank you again for these thoughtful comments and examples.
>
>
> **Q1(a): Provide a more thorough discussion on how the utopian point is selected.**
>
> In the sensitivity analysis experiment (Section *Sensitivity Analysis* and Appendix F.4), we show the robustness of the proposed method to the choice of the utopian point. Specifically, we examine three settings: slightly better than the ideal point (offset 0.01), moderately better (offset 0.1), and significantly better (offset 1.0). The results show that all the three optimistic utopian points lead to performance that is better than (or at least equivalent to) using the original ideal point. This improvement is particularly notable for the moderately and significantly optimistic settings. These findings suggest that 1) the exact ideal point is not necessarily the optimal choice and using a more optimistic utopian point can even lead to better performance, and 2) the algorithm is robust to the utopian point specification, and loose estimates can achieve good results. Note that for some problems like DTLZ2, an offset of 1.0 represents a very loose estimate, given that the objective space ranges approximately in $[0,3]$ for each objective.
>
>
> This robustness behaviour of the proposed method can be intuitively explained by the fact that moving the utopian point from the ideal point to a more optimistic estimate typically does not change how solutions are ranked. For example, consider two bi-objective solutions, $(0.4,0.4)$ and $(0, 1)$. The first can be viewed as the better trade-off. When the utopian point is $(0, 0)$, the first solution has a smaller distance ($0.57 < 1$). When the utopian point is shifted to $(-1, -1)$, it still has a smaller distance $(1.98 < 2.24)$. Even with a more extreme utopian point, such as $(-10, -10)$, the ranking remains unchanged $(14.71 < 14.87)$.
>
> In real-world settings, the ideal point is generally unknown before optimisation begins. However, in many cases, the ideal value of an objective can be estimated optimistically using domain knowledge or trivial bounds. For example, energy loss, fuel consumption, delay time, error, number of defects, or cost can often be idealised as zero; material usage is trivially known to be at least the minimum required volume; cycle time must be no less than the machine’s processing time; and so on. Given the robustness of the proposed method, such optimistic loose estimates can be reliably used.
>
> We will include a detailed discussion about this point in the revised paper.

---

### Official Review · Reviewer_Bjky · 2025-10-31

**Soundness:** 2
**Presentation:** 2
**Contribution:** 2
**Rating:** 2
**Confidence:** 5

**Summary:**

The paper argues that multiobjective optimization gets challenging as the number of objectives increase (because the surface area of the Pareto frontier grows exponentially with objectives). It further argues that identifying the Pareto frontier is wasteful since practitioners still need only one solution in the end. To overcome this, the paper proposes identifying a single solution in a multiobjective setting that provides the "best tradeoff" between objectives. They propose a "single point expected improvement" based on the Euclidean distance to a utiopian point $z^*$.

**Strengths:**

The paper identifies a relevant practical problem in multiobjective optimization -- scaling with number of objectives. However, I am not sure if what they propose is an appropriate solution to that challenge.

**Weaknesses:**

I have several fundamental concerns about this work.
1. They define $\geq 3$ objectives as "many-objective" optimization, while $\geq 2$ is called "multiobjective" optimization -- this seems quite arbitrary (and pointless) to me.
2. They argue that having too many objectives is particularly hard in Bayesian optimization because BO operates under a limited budget. Actually, it is quite the contrary -- BO is known for its sample efficiency and works well when the budget is limited, but is in no way restricted to a limited budget.
3. The authors argue that practitioners typically use only one solution from the Pareto frontier and hence finding the entire Pareto frontier is wasteful -- I disagree. The Pareto frontier, by definition, is a set of solutions that are nondominated (offer the best pragmatic tradeoff). Therefore, Pareto optimal solutions are, theoretically speaking, equally __good__. If a practitioner has to pick one out of this set, it would be based on domain knowledge specific to the application. Without finding the entire Pareto set, it would be impossible for practitioners to pick a single solution.
4. The proposed expected single point improvement (ESPI) essentially tries to find a solution that is closes to the utopian point $z*$ in the Euclidean sense. There is no justification as to why this would be the best choice.
5. The utopian point requires finding independent minimizers of each objective -- I find this to be nontrivial, and there is not sufficient explanation as to how they would address this in high (output) dimensions.
6.  The proposed theoretical results are nothing new or unique to this paper -- they are general guarantees when you have a stochastic acquisition function.
7. The experiments (Fig 2) show log distance (to $z^*$ I believe). It is not clear how did they compute this for other methods, where the entire Pareto frontier is obtained. Did they find the shortest Euclidean distance to the frontier?
8. The performance metrics used in the experiments are questionable. Finding the hypervolume using 1 solution can significantly under/overpredict the hypervolume. It is not clear what the authors are doing to address this.
9. The experiments do not include some recent developments in MOBO
      - qPOTS which finds the Pareto frontier via Thompson sampling and without any hypervolume computation- https://proceedings.mlr.press/v258/renganathan25a.html
      - MORBO which uses trust-regions for MOBO - https://proceedings.mlr.press/v180/daulton22a.html

**Questions:**

I have covered questions within the "Weaknesses" section.

---

> ### Author Response · Authors · 2025-11-27
> **Response [1/4]**
>
> **W1: Definitions of "many-objective" ($\geq 3$) and "multi-objective" ($\geq 2$) sound quite arbitrary.**
>
> We follow the conventional definitions in multi-objective optimisation, where a problem with more than 3 objectives (notably, not including the case of 3 objectives) is referred to as a many-objective problem [Li et al., 2015]. One of the reasons for this specific number is that when the number of objectives exceeds 3, the objective space can no longer be easily visualised.
>
>
>
> **W2: BO is not restricted to a limited budget; it is quite the contrary that having too many objectives is particularly hard in BO.**
>
> Yes, you are right. BO is not necessarily restricted to a limited budget; recent studies have shown that BO can work well with a relatively large number of evaluations (e.g., scalable Gaussian processes [Liu et al., 2020] and trust regions-based methods [Eriksson et al., 2019; Daulton et al., 2022]. However, in general, BO may be less competitive than methods whose computational cost per iteration is independent of the amount of accumulated data (i.e., the number of solutions evaluated), such as evolutionary algorithms, which are widely used for many-objective problems [Li et al. 2015].
>
> It is also worth pointing out that the number of solutions required to well represent the Pareto front increases exponentially with the number of objectives. For example, in a 10-objective problem with a standard simplex-shaped Pareto front, more than 2,000 ($\binom{14}{5}$) Pareto optimal solutions are needed even when only 5 representative points per objective are desired. As such, identifying a good approximation of the entire Pareto front in high-dimensional objective spaces remains challenging for BO.
>
>
> **W3: Theoretically speaking, Pareto optimal solutions are equally good. Without finding the entire Pareto set, it would be impossible for practitioners to pick a single solution.**
>
> We agree that Pareto optimal solutions are equally good in terms of the Pareto dominance relation (as they are mutually incomparable). However, we argue that they are not equally preferable from a decision-maker's perspective. For example, consider two bi-objective nondominated solutions (under minimisation): (1, 1) and (100, 0.99). The first solution is substantially better than the second solution on the first objective and only slightly worse on the second objective, and would almost certainly be preferred by a decision-maker.
>
> In fact, helping practitioners select a single solution from several ones is precisely the aim of multi-criteria decision-making methods, which do not require the full Pareto set and, in some cases, do not even need any preference information from the decision-maker [Lahdelma and Salminen, 2001].

---

> > ### Author Response · Authors · 2025-11-27
> > **Response [2/4]**
> >
> > **W4: The proposed expected single point improvement essentially tries to find a solution that is close to the utopian point in the Euclidean sense. There is no justification as to why this would be the best choice.**
> >
> > The reason for considering a metric that measures the Euclidean distance to the utopian point is not that we believe it would be the best choice, but rather because of its simplicity and commonality. The main contribution of this paper is to propose a novel framework SPMO (single point-based multi-objective optimisation) for BO when dealing with many-objective problems, i.e., a framework that focuses on identifying a single high-quality solution, rather than approximating the entire Pareto front, as is commonly done in the literature. Under our proposed framework, we instantiate SPMO with three simple, well-known metrics: the Euclidean distance to the utopian point, the weighted sum with equal weights (similar to the Manhattan distance), and Tchebycheff scalarisation functions. Among these, we found that the Euclidean distance-based metric performs in general better than the other two (Section *Comparison of Single-Point Metrics within SPMO* and Appendix E.5).
> >
> > The reason why SPMO with the metric based on the Euclidean distance (SPMO$\_{L2norm}$) performs well (e.g., compared to the Manhattan distance SPMO$\_{L1norm}$) is that it tends to prefer more well-balanced solutions. SPMO$\_{L1norm}$ evaluates solutions based on a hyperplane (simplex), treating all solutions lying on that hyperplane as equally good. In contrast, SPMO$\_{L2norm}$ ranks solutions by their Euclidean distance to the utopian point, thus favouring solutions that are more balanced across objectives. For example, consider a bi-objective minimisation problem with three candidate solutions: (0.5, 0.5), (0.3, 0.7), and (0.8, 0.2), and the utopian point is (0, 0). SPMO$\_{L1norm}$ treats all three as equally good ($0.5+0.5=0.3+0.7=0.8+0.2$). In contrast, SPMO$\_{L2norm}$ ranks (0.5, 0.5) as the best, followed by (0.3, 0.7), and then lastly (0.8, 0.2), based on their distances to the utopian point ($0.5^2+0.5^2=0.5 < 0.3^2+0.7^2=0.58 < 0.8^2+0.2^2=0.68$).
> >
> > That said, there may certainly exist other metrics that yield better results than SPMO$\_{L2norm}$, for example, metrics that maximise the hypervolume of a single point (provided the reference point is set properly). In short, within the SPMO framework, any metric that captures the quality of a solution in terms of achieving a good trade-off among objectives can be adopted. In the revised paper, we will add the above discussions to make this issue clearer.
> >
> >
> > **W5: The utopian point requires finding independent minimisers of each objective; there is not sufficient explanation as to how they would address this in high (output) dimensions.**
> >
> > We should have been clearer. The proposed method does not require finding minimisers of each objective – only a loose optimistic estimate is sufficient. In real-world problems, the minimiser of each objective (i.e., the problem’s ideal point) is generally unknown. However, in many cases it can be approximated optimistically using domain knowledge or trivial bounds. For example, objectives like energy loss, fuel consumption, delay time, error, number of defects, or cost can often be idealised as zero; objectives like material usage are trivially known to be at least the minimum required volume; objectives like cycle time must be no less than the machine’s processing time; and so on.
> >
> >
> >
> > In our sensitivity analysis experiment (Section *Sensitivity Analysis* and Appendix F.4), it shows that the proposed method with a wide range of the utopian point settings can work consistently well. Notably, the method with more optimistic estimates than the ideal point performs even better in general. This suggests that 1) using the exact ideal point in the proposed method may not be the optimal choice and a more optimistic one can even yield better performance, and 2) the algorithm is robust to the choice of the utopian point and a loose estimate that is significantly superior to the ideal point can achieve good results. In the revised paper, we will make this issue clearer.

---

> > > ### Author Response · Authors · 2025-11-27
> > > **Response [3/4]**
> > >
> > > **W6: The proposed theoretical results are general guarantees when you have a stochastic acquisition function.**
> > >
> > >
> > > We would like to clarify that our theoretical results are not general guarantees that hold for all stochastic acquisition functions, which is pointed out in [Balandat et al., 2020]. The theoretical results rely on several regularity assumptions (e.g., Lipschitz continuity of the stochastic acquisition function).
> > >
> > > These theoretical results are necessary when optimising stochastic acquisition functions. Particularly, the SAA convergence results indicate that one is able to optimise the MC estimator of an acquisition function to obtain a solution that converges almost surely to the optimal solution of the original function. This approach yields a deterministic acquisition function, which enables using (quasi-) higher-order optimisation methods to obtain fast convergence rates for acquisition optimisation. In fact, this is an established practice for showing theoretical guarantees of a stochastic acquisition function in the area (e.g., in qEHVI and qNEHVI [Daulton et al., 2020; 2021]).
> > >
> > >
> > >
> > >
> > > **W7: For other methods where the entire Pareto front is obtained, is the shortest Euclidean distance to the Pareto front calculated (Figure 2)?**
> > >
> > > For every method considered, the metric is computed based not on the Euclidean distance to the Pareto front, but on the distance to the utopian point. Specifically, for a peer method that produces an approximation of the Pareto front, we identify the solution with the shortest distance to the utopian point and report that distance. Note that using the distance to the Pareto front can fail to identify good trade-off solutions. This is because Pareto optimal solutions can differ greatly in their likelihood of being chosen by the decision-maker, and the shortest distance to the front does not necessarily reflect the true quality of a solution. For example, suppose the Pareto front consists of two points: (1, 1) and (10, 0). Now consider two candidate solutions, (10, 2) and (5, 2). The former is closer to the Pareto front (since $2 < \sqrt{17}$), even though it is dominated by the latter.
> > >
> > >
> > >
> > > Admittedly, when using the metric that the proposed method SPMO directly optimises, we expect its clear advantage. Indeed, as observed, SPMO significantly outperforms its peers on all the problems (Table 1). To provide a complementary comparison, we also consider the hypervolume metric, which SPMO does not optimise. Even under this metric, SPMO still demonstrates a clear statistically significant advantage, achieving higher hypervolume in 40 out of the 42 pairwise comparisons (Table 2).
> > >
> > >
> > > **W8: Finding the hypervolume using 1 solution can significantly under/overpredict the hypervolume.**
> > >
> > > Hypervolume calculates the Lebesgue measure enclosed by a solution set (which may contain only a single solution) and a reference point. A solution that offers a better trade-off across the objectives will receive a higher hypervolume value than another (provided that the reference point is set properly), even when the two solutions are nondominated to each other [Li and Yao, 2019].
> > >
> > >
> > > In this work, we consider two ways of using hypervolume to compare algorithms in the experiment. First, we compare algorithms based on the hypervolume of their single best point, where the proposed SPMO significantly outperforms all competitors (Table 2). Second, we compare algorithms based on the hypervolume of all solutions they produce. Note that in this case, SPMO is expected to perform poorly, since its goal is to identify one high-quality solution, rather than to find a good representation of the entire Pareto front, which is equivalent to maximising the problem’s hypervolume [Zitzler et al., 2003]. However, interestingly, SPMO is still fairly competitive, achieving higher hypervolume in 26 out of 42 pairwise comparisons (with statistical significance). One explanation is that within a tight budget and many objectives, improving convergence of solutions tends to contribute more to hypervolume than enhancing their diversity (as they are still not close to the Pareto front).
> > >
> > >
> > > Lastly, it is worth mentioning that a solution set with a higher hypervolume does not necessarily contain solutions preferred by the decision-maker (see the counterexample in Figure 1 of the paper), since in the end, the decision-maker will not choose multiple solutions but rather only one solution to deploy. As such, in optimisation scenarios involving many objectives and tight budgets, it is likely more valuable to focus the search on identifying one solution of the highest possible quality, rather than expending resources to approximate the entire Pareto front.
> > >
> > > In the revision, we will incorporate the above discussion to make this issue clearer.

---

> > > > ### Author Response · Authors · 2025-11-27
> > > > **Response [4/4]**
> > > >
> > > > **W9: Not including some recent developments in MOBO (qPOTS and MORBO).**
> > > >
> > > >
> > > > In this work, we aim to identify a single balanced solution by quickly searching along one direction towards the Pareto front, rather than seeking to approximate the entire Pareto front. To evaluate the proposed method, we consider six representative MOBO methods. These include one baseline method (Sobol), one method targeting a specific region of the Pareto front (i.e., the central area), and four established methods that aim to approximate the entire Pareto front. Among the methods aiming to approximate the entire front, our selection covers three representative classes of MOBO, i.e., scalarisation-based (EI-based ParEGO and TS-based TS-TCH), HV-based (EHVI), and information-based (JES). Like these methods, most modern state-of-the-art methods (e.g., qPOTS [Renganathan and Carlson, 2025] and MORBO [Daulton et al., 2022]) are also designed with the same goal of approximating the entire front. It is expected that they perform well or better in terms of the hypervolume of all generated solutions, but are unlikely to work well when the goal is to identify a single best trade-off solution, and thus are not the particular focus of our study. We will discuss this in the revised paper.
> > > >
> > > >
> > > >
> > > > **References**
> > > >
> > > > [Li et al., 2015] Li, B., Li, J., Tang, K. and Yao, X., 2015. Many-objective evolutionary algorithms: A survey. ACM Computing Surveys, 2015.
> > > >
> > > >
> > > > [Liu et al., 2020] Liu, H., Ong, Y.S., Shen, X. and Cai, J., 2020. When Gaussian process meets big data: A review of scalable GPs. IEEE Transactions on Neural Networks and Learning Systems, 2020.
> > > >
> > > > [Eriksson et al., 2019] Eriksson, D., Pearce, M., Gardner, J., Turner, R.D. and Poloczek, M., 2019. Scalable global optimization via local Bayesian optimization. NeurIPS, 2019.
> > > >
> > > > [Daulton et al., 2022] Daulton, S., Eriksson, D., Balandat, M. and Bakshy, E., 2022. Multi-objective Bayesian optimization over high-dimensional search spaces. UAI, 2022.
> > > >
> > > > [Lahdelma and Salminen, 2001] Lahdelma, R. and Salminen, P., 2001. SMAA-2: Stochastic multicriteria acceptability analysis for group decision making. Operations Research, 2001.
> > > >
> > > >
> > > > [Balandat et al., 2020] Balandat, M., Karrer, B., Jiang, D., Daulton, S., Letham, B., Wilson, A.G. and Bakshy, E., 2020. BoTorch: A framework for efficient Monte-Carlo Bayesian optimization. NeurIPS, 2020.
> > > >
> > > > [Daulton et al., 2020] Daulton, S., Balandat, M. and Bakshy, E., 2020. Differentiable expected hypervolume improvement for parallel multi-objective Bayesian optimization. NeurIPS, 2020.
> > > >
> > > > [Daulton et al., 2021] Daulton, S., Balandat, M. and Bakshy, E., 2021. Parallel Bayesian optimization of multiple noisy objectives with expected hypervolume improvement. NeurIPS, 2021.
> > > >
> > > > [Li and Yao, 2019] Li, M., and Yao, X., 2019. Quality evaluation of solution sets in multiobjective optimisation: A survey. ACM Computing Surveys, 2019.
> > > >
> > > > [Zitzler et al., 2003] Zitzler, E., Thiele, L., Laumanns, M., Fonseca, C. M., and Da Fonseca, V. G., 2003. Performance assessment of multiobjective optimizers: An analysis and review. IEEE Transactions on Evolutionary Computation, 2003.
> > > >
> > > > [Renganathan and Carlson, 2025] Renganathan, A. and Carlson, K., 2025. qPOTS: Efficient batch multiobjective Bayesian optimization via Pareto optimal Thompson sampling. AISTATS, 2025.

---

### Official Review · Reviewer_cB9x · 2025-11-01

**Soundness:** 4
**Presentation:** 4
**Contribution:** 3
**Rating:** 6
**Confidence:** 3

**Summary:**

The primary problem with MOBO the authors address is the exponential increase in points on the Pareto front of a problem given a constant number of divisions of each objective. The authors claim that most Pareto-optimal solutions are not used in practice, so solving for many points on the Pareto front during MOBO is wasteful. The authors present an alternative to optimizing multiple points on the Pareto front of the MOBO problem, instead presenting a method to optimize in a single direction toward a selected utopian point using a novel gradient based method.

**Strengths:**

The approach is novel and tackles a salient problem; fully sampling the Pareto front is not scalable as the number of objectives increases. The authors demonstrate good performance on a variety of benchmarks, usually pareto-dominating other methods. The method is shown to be relatively robust toward the choice of utopian point. The paper demonstrates that the wall clock time is shorter than the other methods.

**Weaknesses:**

The authors should evaluate the method on a wider range of synthetic benchmarks widely used in the literature such as ZDT1-3, ZDT1-3, and VLMOP2-3. The authors should evaluate on some real-world benchmarks to prove that the method extends to objectives beyond those that can be parameterized by simple polynomial equations such as those presented in “An easy-to-use real-world multi-objective optimization problem suite” by Ryoji Tababe et al.

There are recent high-performance MOBO methods, such as MESMO and DGEMO, that the authors have not compared against.

**Questions:**

The authors show in their utopian point sensitivity analysis that selecting a suboptimal utopian point can lead to a better result. This is quite unintuitive, can the authors explain in more detail when cases like this actually happen? The authors should provide guidance as to how to actually select an optimal utopian point in the paper.

---

> ### Author Response · Authors · 2025-11-27
> **Response [1/2]**
>
> **W1: Evaluation on a wider range of synthetic benchmarks (e.g., ZDT1-3 and VLMOP2-3) and real-world ones beyond those that can be parameterised by simple polynomial equations.**
>
> In this work, we consider an extended suite of DTLZ benchmarks with various characteristics (e.g., concave, convex, inverted, degenerate and/or disconnected Pareto fronts, numerous local optimal solutions, biased mappings from decision to objective spaces, and objectives with different scales). We consider their 5-objective versions in the main text as the proposed work is focused on many-objective optimisation. We thus did not include benchmarks having only 2 or 3 objectives like ZDT (2 objectives) and VLMOP (2 or 3 objectives). For real-world optimisation scenarios, we consider two expensive real-world problems, car side impact design and car cab design (Tanabe and Ishibuchi, 2020), which are parameterised by polynomial equations based on surrogate models that are fit to data collected from a simulator.
> The former is a 4-objective problem without noise, and the latter is a 9-objective problem with noise.
>
> Taking the reviewer's suggestion, we now add a real-world expensive black-box problem from LCBench [Zimmer et al., 2021], accessed via YAHPO Gym [Pfisterer et al., 2022]. The LCBench benchmark consists of 34 instances, each with 8 decision variables and 6 objectives, and we used the default instance (i.e., Instance 3945) in our experiments. The results (means and standard deviations) are reported in the table below.
>
> Table: Results of the distance-based metric (log distance), the HV of the best solution (in terms of its HV value) and the HV of all the solutions obtained by the SPMO and the peer methods on the LCBench problem. Each entry shows the mean and standard deviation in the format "Mean (Std)". The method with the best mean is highlighted in bold. The symbols "+", "∼" and "−" indicate that the method is statistically worse than, equivalent to and better than our SPMO, respectively.
>
> | Method         | Distance-based metric | HV of the best solution | HV of all solutions |
> |----------------|---------------------|--------------------|----------------------|
> | Sobol          | -7.81e-1 (8.9e-2)+  | 6.95e-1 (7.5e-2)+ | 7.88e--1 (4.9e-2)+    |
> | ParEGO         | -1.95e+0 (5.3e-1)+  | 1.49e+0 (2.9e-1)+ | 1.57e+0 (2.9e-1)+     |
> | TS-TCH         | -1.34e+0 (2.2e-1)+  | 1.08e+0 (1.4e-1)+ | 1.15e+0 (1.2e-1)+     |
> | EHVI           | -2.28e+0 (1.4e-1)+  | 1.55e+0 (8.5e-2)+ | **1.69e+0 (1.0e-1)-** |
> | C-EHVI         | -1.86e+0 (5.9e-1)+  | 1.46e+0 (3.2e-1)+ | 1.56e+0 (3.2e-1)+     |
> | JES            | -2.06e+0 (5.2e-1)+  | 1.45e+0 (2.8e-1)+ | 1.52e+0 (2.8e-1)+     |
> | SPMO (Ours)    | **-2.32e+0 (2.2e-1)** | **1.57e+0 (1.3e-1)** | 1.66e+0 (1.2e-1)  |
>
> As we can see from the table, the results have the same pattern as those observed on the other problems considered in the paper. The proposed SPMO consistently performs best on the two metrics that evaluate the quality of a single solution. For the metric that evaluates approximation quality to the entire Pareto front, SPMO remains competitive, ranking second (behind EHVI) and significantly outperforming the other five methods.
>
> We will include the above experimental results in the revised paper.
>
>
>
>
> **W2: Comparison with recent high-performance MOBO methods (e.g., MESMO and DGEMO).**
>
> In this work, we aim to identify a single balanced solution by quickly searching along one direction towards the Pareto front, rather than seeking to approximate the entire Pareto front. To evaluate the proposed method, we consider six representative MOBO methods. These include one baseline method (Sobol), one method targeting a specific region of the Pareto front (i.e., the central area), and four established methods that aim to approximate the entire Pareto front. Among the methods aiming to approximate the entire front, our selection covers three representative classes of MOBO, i.e., scalarisation-based (EI-based ParEGO and TS-based TS-TCH), HV-based (EHVI), and information-based (JES). Like these methods, most state-of-the-art methods (e.g., MESMO [Belakaria et al., 2019; 2021] and DGEMO [Konaković Luković et al., 2020]) are also designed with the same goal of approximating the entire front. It is expected that they perform well or better in terms of the hypervolume of all generated solutions, but are unlikely to work well when the goal is to identify a single best trade-off solution, and thus are not the particular focus of our study. We will discuss this in the revised paper.

---

> > ### Author Response · Authors · 2025-11-27
> > **Response [2/2]**
> >
> > **Q1: In the utopian point sensitivity analysis, why can selecting a suboptimal ideal point lead to a better result? Guidance as to how to select a utopian point.**
> >
> > We should have been clearer – we do not consider a point inferior to the ideal point, instead we consider points that are strictly better than the ideal point. In the sensitivity analysis experiment (Section *Sensitivity Analysis* and Appendix F.4), we examine three such settings. The first setting uses a point slightly better than the ideal point, with an offset of 0.01; for example, if the ideal point is (0, 0) in a bi-objective minimisation case, the utopian point we choose is (-0.01, -0.01). The second setting uses a moderately better point (offset 0.1), and the third one uses a significantly better point (offset 1.0).
> >
> > The experimental results show that all the three optimistic utopian points lead to performance that is better than (or at least equivalent to) using the original ideal point. This improvement is particularly notable for the moderately and significantly optimistic settings. These findings suggest that 1) the exact ideal point is not necessarily the optimal choice and using a more optimistic utopian point can even lead to better performance, and 2) the algorithm is robust to the utopian point specification, and loose estimates can achieve good results. Note that for some problems like DTLZ2, an offset of 1.0 represents a very loose estimate, given that the objective space ranges approximately in $[0,3]$ for each objective.
> >
> > This robustness behaviour of the proposed method can be intuitively explained by the fact that moving the utopian point from the ideal point to a more optimistic estimate typically does not change how solutions are ranked. For example, consider two bi-objective solutions, $(0.4,0.4)$ and $(0, 1)$. The first can be viewed as the better trade-off. When the utopian point is $(0, 0)$, the first solution has a smaller distance ($0.57 < 1$). When the utopian point is shifted to $(-1, -1)$, it still has a smaller distance $(1.98 < 2.24)$. Even with a more extreme utopian point, such as $(-10, -10)$, the ranking remains unchanged $(14.71 < 14.87)$.
> >
> > In real-world settings, the ideal point is generally unknown before optimisation begins. However, in many cases, the ideal value of an objective can be estimated optimistically using domain knowledge or trivial bounds. For example, energy loss, fuel consumption, delay time, error, number of defects, or cost can often be idealised as zero; material usage is trivially known to be at least the minimum required volume; cycle time must be no less than the machine’s processing time; and so on. Given the robustness of the proposed method, such optimistic loose estimates can be reliably used.
> >
> > In the revised paper, we will incorporate the above discussions and provide practical guidance on choosing the utopian point.
> >
> > **References**
> >
> > [Zimmer et al., 2021] Zimmer, L., Lindauer, M. and Hutter, F., 2021. Auto-pytorch: Multi-fidelity metalearning for efficient and robust autodl. TPAMI, 2021.
> >
> > [Pfisterer et al., 2022] Pfisterer, F., Schneider, L., Moosbauer, J., Binder, M. and Bischl, B., 2022, September. YAHPO Gym - an efficient multi-objective multi-fidelity benchmark for hyperparameter optimization. AutoML, 2022.
> >
> > [Belakaria et al., 2019] Belakaria, S., Deshwal, A. and Doppa, J.R., 2019. Max-value entropy search for multi-objective Bayesian optimization. NeurIPS, 2019.
> >
> > [Belakaria et al., 2021] Belakaria, S., Deshwal, A. and Doppa, J.R., 2021. Output space entropy search framework for multi-objective Bayesian optimization. Journal of Artificial Intelligence Research, 2021.
> >
> > [Konaković Luković et al., 2020] Konaković Luković, M., Tian, Y. and Matusik, W., 2020. Diversity-guided multi-objective Bayesian optimization with batch evaluations. NeurIPS, 2020.

---

### Author Response · Authors · 2025-12-04

We thank all reviewers for their thoughtful feedback and area chairs for overseeing the review process. To facilitate the evaluation of our work, we summarise below the motivation and contributions of the paper, major concerns raised by the reviewers, and how we address them. Detailed responses to each reviewer's comments can be found in the point-by-point replies.


**Motivation and contributions**

In the area of multi-objective Bayesian optimisation, particularly for problems with many objectives ($\geq 4$), we have observed that existing methods predominantly aim to approximate the entire Pareto front. We argue this may not be the most appropriate strategy for several reasons:

(1) representing the entire Pareto front typically requires an exponentially growing number of solutions as the number of objectives increases;

(2) in Bayesian optimisation, evaluation budgets are usually very limited (often only a few hundred solutions generated); and

(3) in practice, only a single solution is ultimately selected for deployment.

Motivated by these considerations, we propose an algorithmic framework that focuses on finding a single solution with highest possible quality, rather than generating a diverse set of non-dominated solutions approximating the entire Pareto front. We theoretically show the convergence guarantees of the proposed method (under the Sample Average Approximation) and empirically demonstrate its effectiveness by comparing it with a range of baseline and state-of-the-art multi-objective Bayesian optimisation methods.




**Reviewers' major concerns and how we address them**


- **The setting of the utopian point (Reviewers cB9x, Bjky, T62o, Jbtm)**.

In the original manuscript, we have a section on sensitivity analysis to examine the robustness of the proposed method to the setting of the utopian point. The results show that a wide range of settings (from values slightly better than the ideal point to much looser estimates) yield performance that is better than (or at least equivalent to) that obtained using the problem's ideal point (adopted in our paper). This indicates that 1) the exact ideal point is not necessarily the optimal choice, and using a more optimistic utopian point can even lead to better performance, and 2) the algorithm is robust to the utopian point specification, and coarse estimates can achieve good results. This robustness behaviour can be attributed to the fact that shifting the utopian point from the ideal point to a more optimistic estimate typically does not change the relative ranking of solutions.

In real-world settings, the ideal point is generally unknown before optimisation begins. However, in many cases, the ideal value of an objective can be estimated optimistically using domain knowledge or trivial bounds. For example, energy loss, fuel consumption, latency, error, number of defects, or cost can often be idealised as zero; material usage is trivially known to be at least the minimum required volume; cycle time must be no less than the machine’s processing time; and so on. Given the robustness demonstrated by the proposed method, such optimistic and loosely estimated bounds can be used reliably in practice.

Lastly, it is worth noting that the main contribution of this paper is the introduction of a novel framework for many-objective BO that focuses on identifying a single high-quality solution, rather than approximating the entire Pareto front as is commonly done in the literature. The setting of the utopian point is required only for the distance-based metric, which serves as a simple instantiation of the proposed framework. In fact, any metric that reflects the quality of a solution can be adopted, including those already considered in the paper, such as the weighted-sum and Tchebycheff metrics.

Specific responses to the related comments are provided in **cB9x: Q1; Bjky: W5; T62o: W1\&Q1; Jbtm: W1\&Q1**.



- **Not including some recent MOBO algorithms (Reviewers cB9x and Bjky)**.

These methods were not included because their objectives differ from ours. As explained in the motivation and contributions above, our goal is to identify a single balanced solution by efficiently searching along one direction toward the Pareto front, whereas the recent MOBO methods mentioned by the reviewers are designed to approximate the entire front. For the latter objective, we have in fact included four representative methods in our experimental study, two scalarisation-based methods (EI-based ParEGO and TS-based TS-TCH), one hypervolume-based method (EHVI), and one information-based method (JES). Detailed responses to the related comments are provided in **cB9x: W2; Bjky: W9**.

---

### Meta-Review · Area_Chair_q4Ax · 2026-01-06

**Summary:**

The paper proposes a many-objective Bayesian Optimization framework that focuses on identifying a single high-quality solution (using a distance-based metric to a Utopian point) rather than approximating the entire Pareto front. The authors provided a rebuttal arguing that comparisons with recent state-of-the-art (SOTA) MOBO methods were unnecessary due to differing objectives (single solution vs. full front). However. it must be empirically shown to outperform general MOBO baselines in finding the target solution, which was not adequately established.  The decision is Reject.

**Reviewer Concerns:**

Despite the authors' defense regarding "engineering bounds," the reliance on a specific Utopian point for the distance metric restricts the method's generality compared to oracle-free approaches. This remains a significant constraint for black-box optimization scenarios where such bounds are unknown or misleading.

**Reviewer Scores:**

The refusal to compare with relevant SOTA methods based on a "different goal" argument is a fatal flaw in the experimental design. The paper is rejected but encouraged to be resubmitted with a more rigorous comparative study.

---

### Decision · Program_Chairs · 2026-01-26

Reject